# Molecular basis of accessible plasma membrane cholesterol recognition by the GRAM domain of GRAMD1b

Bilge Ercan[1,†] iD, Tomoki Naito[1,†] iD, Dylan Hong Zheng Koh[1] iD, Dennis Dharmawan[1] & Yasunori Saheki[1,2,*] iD

## Abstract

Cholesterol is essential for cell physiology. Transport of the "accessible" pool of cholesterol from the plasma membrane (PM) to the endoplasmic reticulum (ER) by ER-localized GRAMD1 proteins (GRAMD1a/1b/1c) contributes to cholesterol homeostasis. However, how cells detect accessible cholesterol within the PM remains unclear. We show that the GRAM domain of GRAMD1b, a coincidence detector for anionic lipids, including phosphatidylserine (PS), and cholesterol, possesses distinct but synergistic sites for sensing accessible cholesterol and anionic lipids. We find that a mutation within the GRAM domain of GRAMD1b that is associated with intellectual disability in humans specifically impairs cholesterol sensing. In addition, we identified another point mutation within this domain that enhances cholesterol sensitivity without altering its PS sensitivity. Cell-free reconstitution and cell-based assays revealed that the ability of the GRAM domain to sense accessible cholesterol regulates membrane tethering and determines the rate of cholesterol transport by GRAMD1b. Thus, cells detect the codistribution of accessible cholesterol and anionic lipids in the PM and fine-tune the non-vesicular transport of PM cholesterol to the ER via GRAMD1s.

**Keywords** cholesterol; GRAM domain; lipid sensor; membrane contact sites; plasma membrane

**Subject Categories** Membranes & Trafficking; Metabolism

**The EMBO Journal (2021) 40: e106524**

## Introduction

Sterol serves as a major building block for cellular membranes in eukaryotes. In metazoans, cholesterol represents approximately 20% of total cellular lipids and is therefore critical for the structural integrity of cellular membranes and for cell physiology, including the regulation of signaling pathways (van Meer *et al*, 2008; Vance, 2015). Cells either synthesize cholesterol *de novo* in the endoplasmic reticulum (ER) or acquire it from external sources, primarily through receptor-mediated endocytosis of low-density lipoproteins (LDLs) (Goldstein & Brown, 2015). Up to 90% of total cellular cholesterol is concentrated in the plasma membrane (PM), where it represents almost half of the total lipids in this bilayer (Ray *et al*, 1969; de Duve, 1971; Lange *et al*, 1989; Maxfield & Mondal, 2006; Ikonen, 2008; van Meer *et al*, 2008). Thus, cells must constantly monitor the levels of cholesterol in their PM and adjust levels of cholesterol biosynthesis or uptake to maintain homeostasis. This is mediated, at least in part, by the regulated transport of cholesterol between the PM and the ER, where the sterol regulatory element binding proteins (SREBPs) regulatory network that controls cholesterol biosynthesis and rates of LDL uptake is located (Goldstein & Brown, 1990; Brown *et al*, 2018).

Sterols, including cholesterol, are transported between various cellular membranes primarily via non-vesicular transport, a process that is independent of membrane trafficking (Urbani & Simoni, 1990; Heino *et al*, 2000; Hao *et al*, 2002; Baumann *et al*, 2005; Ikonen, 2008). Normally, the majority of PM cholesterol forms complexes with other membrane lipids, including sphingomyelin and phospholipids, and is therefore sequestered and inaccessible for intracellular transport (also known as "chemically inactive") (Radhakrishnan & McConnell, 2000; Ohvo-Rekila *et al*, 2002; McConnell & Radhakrishnan, 2003; Lange *et al*, 2004; Sokolov & Radhakrishnan, 2010; Lange *et al*, 2013; Das *et al*, 2014; Gay *et al*, 2015; Chakrabarti *et al*, 2017). However, a small fraction of PM cholesterol (~15% of PM lipids) remains unsequestered and accessible for extraction and transport (also known as "chemically active") (Das *et al*, 2014). Increases in the level of PM cholesterol beyond a certain threshold (or liberation of PM cholesterol from sequestration) result in transient expansions in the accessible pool of PM cholesterol. Cells respond by rapidly transporting the newly expanded pool of accessible cholesterol to the ER to suppress the activities of SREBPs, shutting down cholesterol biosynthesis and

1 Lee Kong Chian School of Medicine, Nanyang Technological University, Singapore
2 Institute of Resource Development and Analysis, Kumamoto University, Kumamoto, Japan
*Corresponding author. Tel: +65 6592 3996; Fax: +65 6515 0417; E-mail: yasunori.saheki@ntu.edu.sg
†These authors contributed equally to this work

uptake, thereby avoiding cholesterol overaccumulation and maintaining PM cholesterol levels (Slotte & Bierman, 1988; Lange & Steck, 1997; Scheek et al, 1997; Das et al, 2014; Lange et al, 2014; Infante & Radhakrishnan, 2017). Artificially trapping accessible cholesterol in the PM results in constitutive activation of SREBPs and dysregulated cholesterol metabolism (Infante & Radhakrishnan, 2017; Johnson et al, 2019). Despite its critical importance, how cells sense the accessibility of PM cholesterol and regulate cholesterol transport from the PM to the ER is still poorly understood. Identifying the molecular basis of PM cholesterol sensing will provide important insights into the regulation of cholesterol metabolism.

The ER extends throughout the cytoplasm, forming physical contacts with virtually all other cellular compartments, including the PM (Phillips & Voeltz, 2016; Wu et al, 2018). These membrane contact sites play critical roles in many aspects of cell physiology, including the non-vesicular exchange/delivery of specific lipids (e.g., cholesterol) by lipid transfer proteins (Elbaz & Schuldiner, 2011; Lev, 2012; Drin, 2014; Holthuis & Menon, 2014; Gatta et al, 2015; Murley et al, 2015; Saheki et al, 2016; Saheki & De Camilli, 2017a,b; Antonny et al, 2018; Luo et al, 2018; Jeyasimman & Saheki, 2019; Nishimura & Stefan, 2019; Petrungaro & Kornmann, 2019; Wong et al, 2019; Lees & Reinisch, 2020; Meng et al, 2020; Prinz et al, 2020). We and others recently found that a family of evolutionarily conserved and ER-anchored sterol transfer proteins, the GRAMD1s/Asters (GRAMD1a/Aster-A, GRAMD1b/Aster-B, and GRAMD1c/Aster-C) (the Lam/Ltc proteins in yeast), mediate non-vesicular sterol transport from the PM to the ER at ER-PM contact sites, thereby contributing to sterol homeostasis (Gatta et al, 2015; Sandhu et al, 2018; Naito et al, 2019; Marek et al, 2020). GRAMD1s possess an N-terminal GRAM domain, an evolutionarily conserved PH-like domain (Begley et al, 2003; Tong et al, 2018), a cholesterol-harboring StART-like domain, and a C-terminal transmembrane domain that anchors the protein to the ER. Importantly, the GRAM domain of GRAMD1s acts as a unique coincidence detector, detecting the presence of both accessible cholesterol and anionic lipids (Naito et al, 2019), which include phosphatidylserine (PS) (a major acidic phospholipid of the PM) (Leventis & Grinstein, 2010). Further, the GRAM domain facilitates the cholesterol-dependent recruitment of GRAMD1s to ER-PM contact sites when the accessible pool of PM cholesterol transiently expands beyond a certain threshold (Naito et al, 2019). GRAMD1s then extract accessible PM cholesterol for transport to the ER via the StART-like domain, which can directly capture and transport sterol across the cytoplasm (Gatta et al, 2018; Horenkamp et al, 2018; Jentsch et al, 2018; Sandhu et al, 2018; Tong et al, 2018; Khelashvili et al, 2019; Naito et al, 2019). In the absence of GRAMD1s, accessible cholesterol is inefficiently transported from the PM to the ER, resulting in a chronic expansion of accessible PM cholesterol (Naito et al, 2019; Ferrari et al, 2020).

Interestingly, recent human genetic studies, including genome-wide association studies, have identified links between GRAMD1b and neurodevelopmental disorders such as intellectual disability and schizophrenia (Schizophrenia Working Group of the Psychiatric Genomics C, 2014; Reuter et al, 2017; Santos-Cortez et al, 2018; Thyme et al, 2019). Notably, a missense mutation in the GRAM domain of GRAMD1b (R189W) was identified in a consanguineous family with moderate intellectual disability, indicating the importance of the GRAM domain in GRAMD1b function (Reuter et al, 2017). However, the precise mechanism by which GRAM domains

sense transient expansions in the accessible pool of PM cholesterol and the impact that the intellectual disability-associated missense mutation has on GRAMD1b function both remain unknown.

The GRAM domain was originally identified in membrane-associated proteins, including myotubularin family phosphatases (MTMs) (Doerks et al, 2000). Crystal structures of the GRAM domain of myotubularin-related protein MTMR2 (Begley et al, 2003; Begley et al, 2006) and that of Lam6/Ltc1, a homolog of GRAMD1s in yeast (Tong et al, 2018), together revealed a high degree of structural similarity between the GRAM domain and the PH domain, which often binds anionic lipids, including phosphoinositides, in membranes. The overall structure of the GRAM domain is comprised of a typical PH domain β-sandwich fold, which consists of two seven-stranded antiparallel β-sheets followed by a C-terminal α-helix and an additional short N-terminal α-helix in the case of Lam6/Ltc1 (Begley et al, 2003; Tong et al, 2018). GRAM domains have been reported to bind phosphoinositides, with specificity for PI$(3,5)P_2$ and PI5P, in the case of the myotubularin family of proteins (Berger et al, 2003; Tsujita et al, 2004; Lorenzo et al, 2005), or for PI$(4,5)P_2$ and PI4P, in the case of GRAMD2 (a paralog of the GRAMD1s that does not contain a StART-like domain) (Besprozvannaya et al, 2018). However, sites within the GRAM domain that bind these anionic lipids are currently unknown. It also remains unclear how GRAMD1 GRAM domains bind most efficiently to membranes that contain both cholesterol and anionic lipids. Because GRAMD1s are evolutionarily conserved, elucidating this process will provide key insights into the intracellular mechanisms by which sterol is sensed and transported in eukaryotes.

In this study, we explore how the GRAM domain of GRAMD1b detects accessible cholesterol in the PM and regulates GRAMD1-mediated PM to ER cholesterol transport. Our results show that the GRAM domain of GRAMD1b possesses distinct but physically proximal lipid-sensing sites and that one is dedicated to cholesterol and one is dedicated to anionic lipids. Together, they mediate synergistic binding to membranes that contain both lipids in close proximity, contributing to the detection of the codistribution of accessible cholesterol and anionic lipids, including PS, in the PM when such codistribution becomes most prominent during transient increases in accessible PM cholesterol. Remarkably, a GRAMD1b mutation (R189W), which has been associated with intellectual disability in humans, specifically reduces the ability of the GRAM domain to bind cholesterol without affecting its affinity toward PS, dramatically impairing the ability of GRAMD1b to sense transient expansions in the accessible pool of PM cholesterol. We established a cell-free reconstitution assay using purified near full-length GRAMD1b proteins and artificial membranes and used this assay to demonstrate that GRAMD1b proteins with the R189W mutation failed to tether membranes and transported cholesterol less efficiently compared to wild-type (WT) GRAMD1b proteins. Accordingly, re-expression of WT GRAMD1b, but not the mutant GRAMD1b, in GRAMD1 triple knockout cells restore proper regulation of SREBP-2 and suppress the abnormal accumulation of accessible PM cholesterol. Finally, we performed a mini-mutagenesis screen and found that a single glycine residue at the position 187 (G187) is critical for the sensitivity of the GRAM domain to accessible PM cholesterol. Converting this residue to the more hydrophobic leucine (G187L) made the GRAM domain hypersensitive to cholesterol without altering its affinity toward PS and enhanced GRAMD1b-dependent

membrane tethering and cholesterol transport. Thus, the cholesterol-sensing property of the GRAM domain is critical for determining the rate of GRAMD1-dependent cholesterol transport. Collectively, our results demonstrate that cells monitor the extent to which accessible cholesterol and anionic lipids are codistributed within the inner leaflet of the PM via the GRAM domain of GRAMD1s and use this information to fine-tune the extraction and transport of accessible PM cholesterol to the ER via GRAMD1s.

## Results

### The GRAM domain of GRAMD1b detects the codistribution of accessible cholesterol and PS, which is regulated by sphingomyelin

Cholesterol forms complexes with other lipids, including phospholipids and sphingomyelin. The formation of these complexes contributes to the sequestration of cholesterol, making it inaccessible in membranes. However, the ability of the GRAM domain of GRAMD1b ($GRAM_{1b}$) to detect the presence of both anionic lipids, including PS (a major acidic phospholipid enriched in the inner leaflet of the PM), and accessible cholesterol as a coincidence detector (Naito *et al*, 2019) suggests that at least some cholesterol codistributes with PS without being sequestered in the membrane. Such codistribution may provide a platform for $GRAM_{1b}$ to detect the amount of accessible cholesterol in the inner leaflet of the PM. To confirm this codistribution, we used the Förster resonance energy transfer (FRET) assay. Liposomes containing various amounts of lipids were incubated with ECFP-D4H (an accessible cholesterol biosensor) (Maekawa & Fairn, 2015) and mVenus-Lact-C2 (a PS biosensor) (Yeung *et al*, 2008), and FRET between ECFP and mVenus was measured at 525 nm. Close proximity (1–10 nm) of PS

and accessible cholesterol in liposomes would result in an increase in the FRET signal when ECFP was excited at 430 nm (Fig 1A). Strikingly, strong FRET was observed when both cholesterol (50%) and PS (20%) were simultaneously incorporated into liposomes [i.e. the condition that allows both ECFP-D4H and mVenus-Lact-C2 to robustly interact with membranes (Fig EV1A)] (Fig 1B and C). Additional incorporation of sphingomyelin (25%) into the liposomes reduced the FRET signals and increased ECFP fluorescence (Figs EV1B and 1B and C). As a control, liposomes containing only phosphatidylcholine (PC), only cholesterol (50%), or only PS (20%) were also examined; no FRET was observed in any of these conditions (Fig 1B and C), demonstrating the specificity of the FRET signals. These results are consistent with the codistribution of accessible cholesterol and PS in membranes. Importantly, these results also demonstrate that such codistribution can be modulated by the additional presence of sphingomyelin, which normally sequesters cholesterol into inaccessible pools in the PM.

FRET was decreased by addition of either untagged Lact-C2 or untagged D4H in a concentration-dependent manner (Fig EV1C–F). If codistribution of accessible cholesterol and PS also occurs in the inner leaflet of the PM, the ability of $GRAM_{1b}$ to bind the PM should also be affected by either masking/trapping accessible PM cholesterol via incubation of cells with purified D4H proteins (Infante & Radhakrishnan, 2017; Johnson *et al*, 2019) or masking PS within the inner leaflet of the PM via the overexpression of Lact-C2 (Raghupathy *et al*, 2015).

We previously showed that $GRAM_{1b}$ is recruited to the PM when cholesterol is liberated from sphingomyelin-sequestered pools of PM cholesterol via the hydrolysis of sphingomyelin by sphingomyelinase treatment (Naito *et al*, 2019). Unsequestered cholesterol can spontaneously flip-flop between the outer and inner leaflets of the PM bilayers (Leventis & Silvius, 2001; Steck & Lange, 2018). Thus, despite being enriched in the outer leaflet of the PM, sphingomyelin

---

**Figure 1. The GRAM domain of GRAMD1b detects the codistribution of accessible cholesterol and PS, which is regulated by sphingomyelin.**

A   Schematic of the *in vitro* FRET assay to detect codistribution of phosphatidylserine (PS) and accessible cholesterol in membranes. Liposomes containing PS and cholesterol were mixed with an accessible cholesterol biosensor (ECFP-D4H) and a PS biosensor (mVenus-Lact-C2). ECFP-D4H was excited at 430 nm and emission from mVenus-Lact-C2 due to FRET between ECFP and mVenus was recorded at 525 nm.

B   Representative emission spectra of mixtures containing ECFP-D4H, mVenus-Lact-C2, and liposomes containing indicated compositions of lipids. Note that FRET signal was increased in the presence of both cholesterol (50%) and phosphatidylserine (DOPS) (20%) and that FRET signal was decreased when sphingomyelin (SM) (25%) was additionally incorporated into the liposomes. Asterisks indicate positions of emission maximum of ECFP (*477 nm) and FRET (**525 nm). DOPC, phosphatidylcholine (1,2-dioleoyl-sn-glycero-3-phosphocholine); DOPS, phosphatidylserine (1,2-dioleoyl-sn-glycero-3-phospho-L-serine); Chol., cholesterol; SM, sphingomyelin (N-oleoyl-D-erythro-sphingosylphosphorylcholine).

C   Quantification of the ratio of FRET signal at 525 nm to ECFP emission at 480 nm (FRET/ECFP) (see Materials and Methods) from mixtures as shown in (B) (mean ± SEM, $n = 4$ independent experiments; Dunnett's multiple comparisons test, **$P < 0.0001$).

D   Left: Time course of normalized mCherry signal, as assessed by TIRF microscopy, from GRAMD1 triple knockout (TKO) HeLa cells expressing mCherry-tagged GRAM domain of GRAMD1b (mCherry-$GRAM_{1b}$). Cells were pre-incubated with either control imaging buffer (for control) or imaging buffer containing purified ECFP-D4H proteins (3 μM) (for masking accessible PM cholesterol) for 30 min at 37°C and imaged with the same buffer conditions. Sphingomyelinase (SMase) treatment (100 mU/ml) is indicated. Right: Values of $\Delta F/F_0$ corresponding to the end of the experiment as indicated by the arrow [mean ± SEM, $n = 32$ cells (Control), $n = 35$ cells (Pre-incubation w/ ECFP-D4H); data are pooled from two independent experiments for each condition; two-tailed unpaired Student's *t*-test, **$P < 0.0001$].

E   Left: Time course of normalized EGFP signal, as assessed by TIRF microscopy, from GRAMD1 triple knockout (TKO) HeLa cells expressing EGFP-tagged $GRAM_{1b}$ (EGFP-$GRAM_{1b}$) together with either mCherry for control or mCherry-Lact-C2 for PS masking. SMase treatment (100 mU/ml) is indicated. Right: Values of $\Delta F/F_0$ corresponding to the end of the experiment as indicated by the arrow [mean ± SEM, $n = 53$ cells (mCherry), $n = 32$ cells (mCherry-Lact-C2 OE); data are pooled from four independent experiments for each condition; two-tailed unpaired Student's *t*-test, **$P < 0.0001$].

F   Schematics showing the interaction of the $GRAM_{1b}$ with the plasma membrane (PM) before and after sphingomyelinase (SMase) treatment. Left: At rest, subthreshold levels of accessible PM cholesterol, or limited presence of the codistribution of accessible cholesterol and anionic lipids in the inner leaflet of the PM, are not sufficient to induce interaction of the $GRAM_{1b}$ with the PM. Right: Liberation of the sphingomyelin (SM)-sequestered pool of cholesterol by SMase treatment leads to an increase in accessible PM cholesterol in both inner and outer leaflets (as unsequestered cholesterol can flip-flop between leaflets), resulting in concomitant increase in the codistribution of accessible cholesterol and anionic lipids, including phosphatidylserine (PS), beyond the threshold in the inner leaflet of the PM, and inducing PM recruitment of the $GRAM_{1b}$ as it detects such codistribution in the PM.

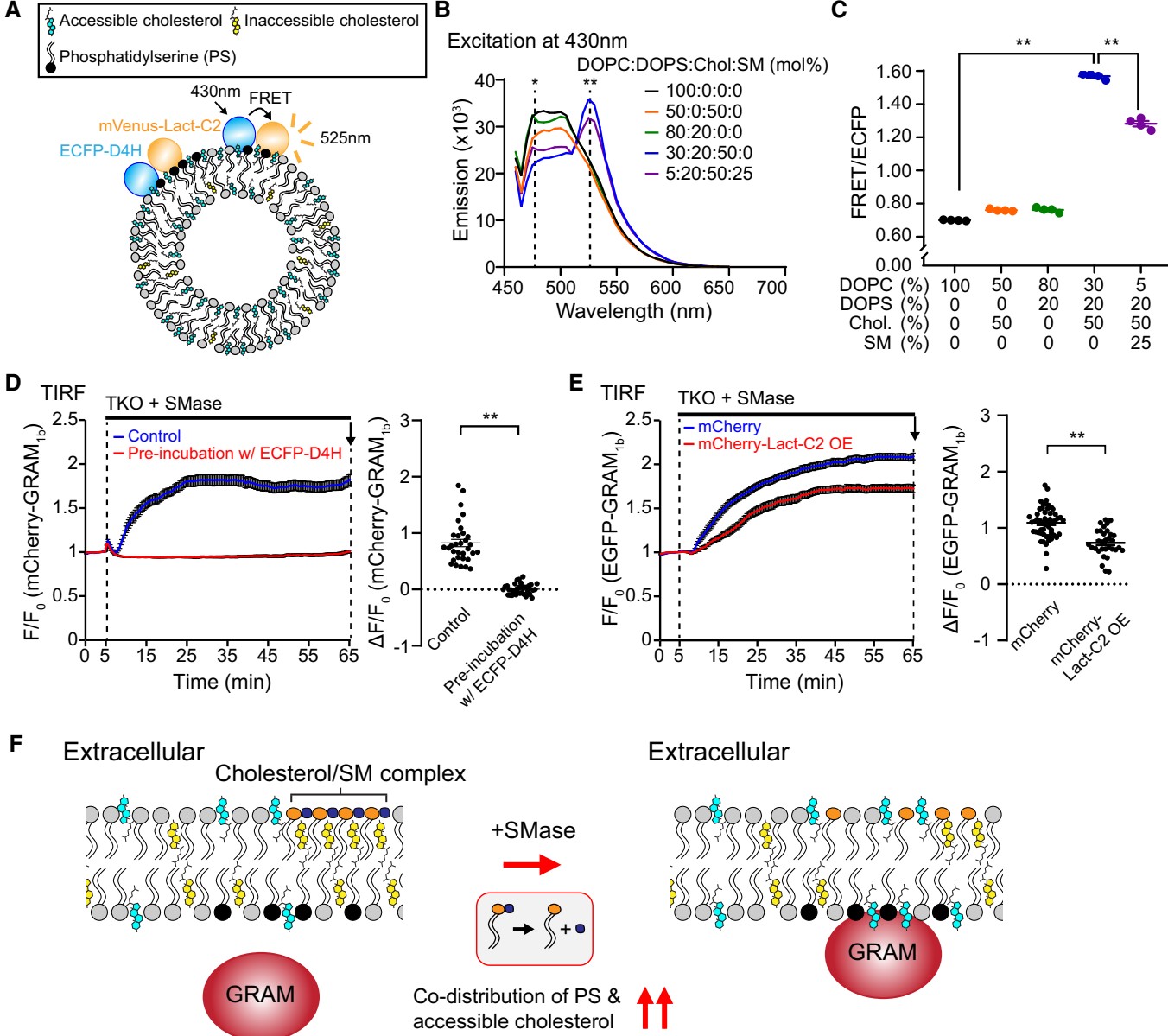

**Figure 1.**

contributes significantly to suppressing the accessibility of cholesterol in the inner leaflet of the PM at steady state. The recruitment of $GRAM_{1b}$ to the PM following sphingomyelinase treatment was significantly enhanced in HeLa cells that lacked all three GRAMD1s (GRAMD1 TKO cells) compared to WT HeLa cells due to exaggerated accumulation of accessible PM cholesterol in these cells (Naito *et al*, 2019). Using GRAMD1 TKO cells as a model system, we examined the requirement of accessible cholesterol and PS within the PM in the binding of the $GRAM_{1b}$ to this bilayer. To examine the specific requirement of accessible PM cholesterol, we pre-incubated GRAMD1 TKO cells that expressed mCherry-tagged $GRAM_{1b}$ ($mCherry$-$GRAM_{1b}$) with either imaging buffer alone (control) or imaging buffer that contained purified ECFP-D4H proteins for 30 min and then treated the cells with sphingomyelinase in the same buffer conditions and measured the recruitment of mCherry-

$GRAM_{1b}$ to the PM via total internal reflection fluorescence (TIRF) microscopy (Fig 1D). To examine the specific requirement of PS in the inner leaflet of the PM, we treated GRAMD1 TKO cells that expressed EGFP-$GRAM_{1b}$ together with mCherry (control) or EGFP-$GRAM_{1b}$ together with mCherry-tagged Lact-C2 (mCherry-Lact-C2) with sphingomyelinase and measured the recruitment of EGFP-$GRAM_{1b}$ to the PM via TIRF microscopy (Fig 1E). In both cases, sphingomyelinase treatment resulted in strong recruitment of $GRAM1_{b}$ ($mCherry$-$GRAM1_{b}$ or EGFP-$GRAM_{1b}$) to the PM in GRAMD1 TKO cells in control conditions (Fig 1D and E). However, PM recruitment of $GRAM_{1b}$ ($mCherry$-$GRAM1_{b}$ or EGFP-$GRAM_{1b}$) was completely abolished by pre-incubation with purified ECFP-D4H proteins (Fig 1D) and significantly attenuated by mCherry-Lact-C2 overexpression (Fig 1E), indicating that the efficient binding of $GRAM_{1b}$ to the PM requires both accessible PM cholesterol and

PS as coligands. The moderate effect of mCherry-Lact-C2 overexpression suggests that the binding of GRAM$_{1b}$ to the PM may additionally depend on other anionic lipids within this bilayer.

Taken together, these results strongly imply that GRAM$_{1b}$ detects the codistribution of accessible cholesterol and anionic lipids, including PS, within the inner leaflet of the PM, which occurs most prominently when the accessible pool of cholesterol in this bilayer expands (Fig 1F). These data further indicate that GRAM$_{1b}$ possesses motifs for sensing anionic lipids and/or accessible cholesterol to detect their codistribution.

## A basic patch within the GRAM domain of GRAMD1b is critical for anionic lipid recognition

We constructed a structural model of GRAM$_{1b}$ using the crystal structure for the GRAM domain of Lam6/Ltc1, a homolog of GRAMD1s in yeast (Fig EV2A and B), as a template (Tong *et al*, 2018). This model predicted the presence of a basic patch on the surface of GRAM$_{1b}$ that may contribute to recognizing the acidic head group of PS and other anionic lipids in membranes (Fig 2A and B). Amino acid sequence analysis indicated that a highly conserved lysine (K) in the 161$^{st}$ amino acid position in the β5-β6 loop and a conserved arginine (R) in the 191$^{st}$ amino acid position in the C-terminal α-helix (α2) of GRAM$_{1b}$ may contribute to the charge distribution of the basic patch (Figs 2A and EV2B–D). To elucidate the importance of K161 and R191 in recognition of PS and other anionic lipids, we generated mutant versions of GRAM$_{1b}$ in which the charges of these residues were individually or simultaneously neutralized (K161A, R191A, and K161A/R191A) (Fig 2B). These GRAM$_{1b}$ mutants were purified and their ability to bind artificial membranes containing PS alone, cholesterol alone, or both lipids was assessed.

WT control GRAM$_{1b}$ bound to liposomes that contained PS alone (0-80%) in a concentration-dependent manner, as we previously reported (Naito *et al*, 2019) (Fig 2C). However, mutant versions of GRAM$_{1b}$ that carried the K161A, R191A, or K161A/R191A mutation bound to liposomes less efficiently than WT controls. For liposomes with 80% PS we measured ~20% binding for K161A and R191A, ~5% binding for K161A/R191A, and ~45% binding for WT GRAM$_{1b}$ (Fig 2C). This suggests a critical role for the basic patch in detecting PS and possibly other anionic lipids in membranes. GRAM$_{1b}$ binds weakly to liposomes that contain only cholesterol (~15–20% binding when liposomes contained 60% cholesterol) (Naito *et al*, 2019) (Fig 2C) (see also Fig 3A and B). In contrast to PS detection, WT and mutant GRAM$_{1b}$ bound to liposomes that contained only cholesterol (60%) to the same extent (~15% binding) (Fig 2C).

Building upon these results, we analyzed the importance of the PS-sensing property of the basic patch of the GRAM$_{1b}$ in coincidence detection of cholesterol and PS (i.e., the detection of cholesterol/PS codistribution in membranes). To this end, liposomes containing a fixed amount of PS (20%) and increasing amounts of cholesterol (0–50%) were generated and the binding of purified GRAM domains to these liposomes was examined. WT control GRAM$_{1b}$ bound strongly to liposomes that contained 40% or 50% cholesterol, consistent with our previous results (Naito *et al*, 2019) (Fig 2D). Mutant GRAM$_{1b}$ (K161A, R191A, or K161A/R191A) bound significantly less to liposomes compared to WT GRAM$_{1b}$. For liposomes that contained 20% PS and 40% cholesterol, we measured ~40%

binding for K161A and R191A, ~20% binding for K161A/R191A, and ~60% binding for WT (Fig 2D). This demonstrates the important role of the basic patch of GRAM$_{1b}$ in detecting the codistribution of PS and cholesterol within membranes.

To further address whether the basic patch of GRAM$_{1b}$ has a specific role in PS sensing or a general role in anionic lipid sensing, liposomes containing a fixed amount of anionic lipids [either PS, phosphatidic acid (PA), PI(4)P, PI(4,5)P$_2$, or phosphatidylinositol (PI)] (10%) and cholesterol (50%) were generated and the binding of purified GRAM domains to these liposomes was examined (Fig 2E). WT control GRAM$_{1b}$ bound liposomes that contained any one of these anionic lipids with some preference toward PS, PA, PI (4)P, and PI(4,5)P$_2$, consistent with our previous results (Naito *et al*, 2019). Mutant GRAM$_{1b}$ (K161A, R191A, or K161A/R191A) bound significantly less to these liposomes compared to WT GRAM$_{1b}$, with K161A/R191A showing the strongest impact. These results support a general role for the basic patch in anionic lipid sensing primarily through electrostatic interaction (Fig 2E).

As major defects in the ability of mutant GRAM$_{1b}$ (e.g. basic patch mutants) to sense anionic lipids did not alter their ability to sense cholesterol, GRAM$_{1b}$ likely possesses distinct sites for recognizing anionic lipids and cholesterol. Thus, these results indicate the presence of another dedicated site for detecting accessible cholesterol that may act independently from the basic patch.

## A GRAMD1b mutation (R189W) associated with intellectual disability specifically impairs binding of the GRAM domain to cholesterol

The GRAM$_{1b}$ binds to liposomes that contain only cholesterol, but this binding requires high levels of cholesterol (Fig 2C). To examine the specificity and selectivity of such binding, we replaced cholesterol with other sterols, namely desmosterol, cholesteryl acetate, epicholesterol, and dehydroergosterol (DHE), and investigated the importance of the hydrocarbon side chain (with desmosterol), hydroxyl head group (with cholesteryl acetate and epicholesterol), and tetracyclic ring structure (with DHE) in binding of GRAM$_{1b}$ to sterol-containing liposomes (60%) (Fig 3A and B). Replacing cholesterol with desmosterol had no effect, but replacing cholesterol with cholesteryl acetate, epicholesterol, or DHE eliminated the binding of GRAM$_{1b}$ to liposomes (Fig 3B). GRAM$_{1b}$ bound equally robustly to liposomes containing 20% PS and either 60% desmosterol or 60% cholesterol (~95% binding) (Appendix Fig S1A). These results demonstrate that the binding of GRAM$_{1b}$ to cholesterol-containing membranes is specific and depends on the stereospecificity of the hydroxyl head group and the tetracyclic ring structure.

To identify amino acid residues that are essential for GRAM$_{1b}$ to recognize cholesterol in membranes, we searched for amino acid residues within GRAM$_{1b}$ that are linked to human diseases. Recent human genetic studies identified several mutations in GRAMD1b that are associated with intellectual disability (Reuter *et al*, 2017; Santos-Cortez *et al*, 2018). One of these mutations is R189W, which was identified in a consanguineous family with intellectual disability (Reuter *et al*, 2017). Interestingly, R189 is a residue that is evolutionarily conserved from yeast to humans (Fig EV2B–D). Structural modeling revealed that residue R189 is located in the β7-α2 loop of GRAM$_{1b}$, near amino acid residues that contribute to PS/anionic lipid-sensing (K161, R191) (Fig 3C). We generated a mutant version

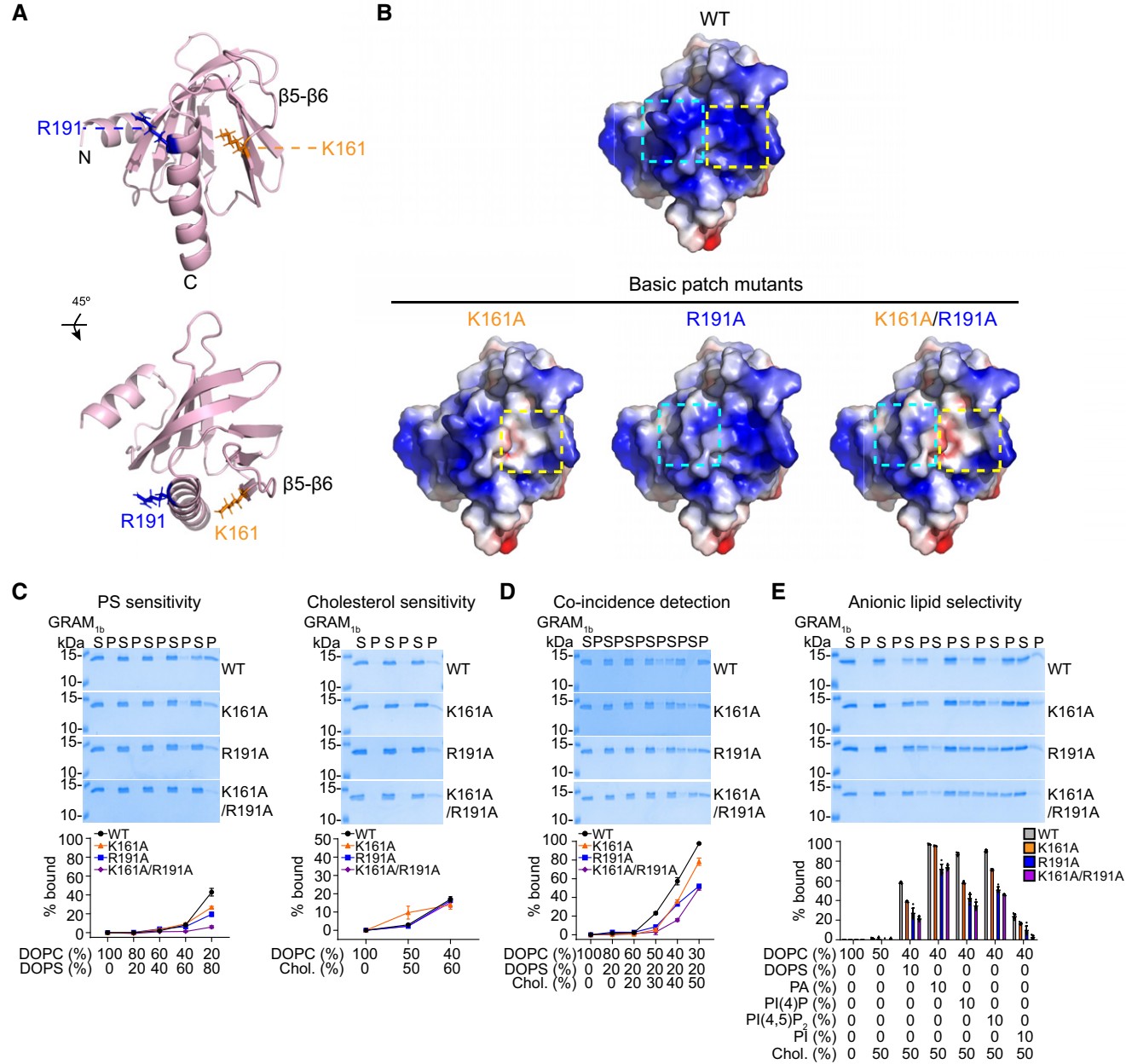

**Figure 2. A basic patch within the GRAM domain of GRAMD1b is critical for anionic lipid recognition.**

A  The ribbon diagram of the modeled GRAM domain of GRAMD1b (GRAM$_{1b}$) (see Materials and Methods). Amino acid side chains that contribute to the basic patch of GRAM$_{1b}$ are indicated. The bottom panel shows the diagram that was rotated 45° about the indicated axis. N, N-terminus; C, C-terminus.

B  Electrostatic surface representations of the modeled wild-type GRAM$_{1b}$ (WT) and mutant versions of GRAM$_{1b}$ carrying mutations that are predicted to alter charge distribution of the basic patch (basic patch mutants: K161A, R191A, and K161A/R191A). Blue indicates positively charged surface, while red indicates negatively charged surface at the level of ± 2 kT/e. The electrostatic surface representations were generated by APBS Electrostatics plugin using PyMoL. The dashed boxes indicate the positions of the mutated amino acid residues.

C–E  Liposome sedimentation assays of wild-type GRAM$_{1b}$ proteins (WT) and mutant versions of GRAM$_{1b}$ proteins carrying indicated mutations (K161A, R191A, and K161A/R191A). Liposomes containing the indicated mole% lipids were incubated with purified GRAM$_{1b}$ proteins as shown. Bound proteins [pellet, (P)] were separated from the unbound proteins [supernatant, (S)], run on SDS–PAGE and visualized by colloidal blue staining (mean ± SEM, *n* = 3 independent experiments for all conditions in phosphatidylserine (PS) sensitivity, coincidence detection, and anionic lipid selectivity assays; *n* = 12 independent experiments for WT, *n* = 3 independent experiments for K161A and R191A, and n = 6 independent experiments for K161A/R191A in the cholesterol sensitivity assay). DOPC, phosphatidylcholine (1,2-dioleoyl-sn-glycero-3-phosphocholine); DOPS, phosphatidylserine (1,2-dioleoyl-sn-glycero-3-phospho-L-serine); Chol., cholesterol; PA, phosphatidic acid (1,2-dioleoyl-sn-glycero-3-phosphate); PI(4)P, phosphatidylinositol 4-phosphate; PI(4,5)P$_2$, phosphatidylinositol 4,5-bisphosphate; PI, L-α-phosphatidylinositol.

Source data are available online for this figure.

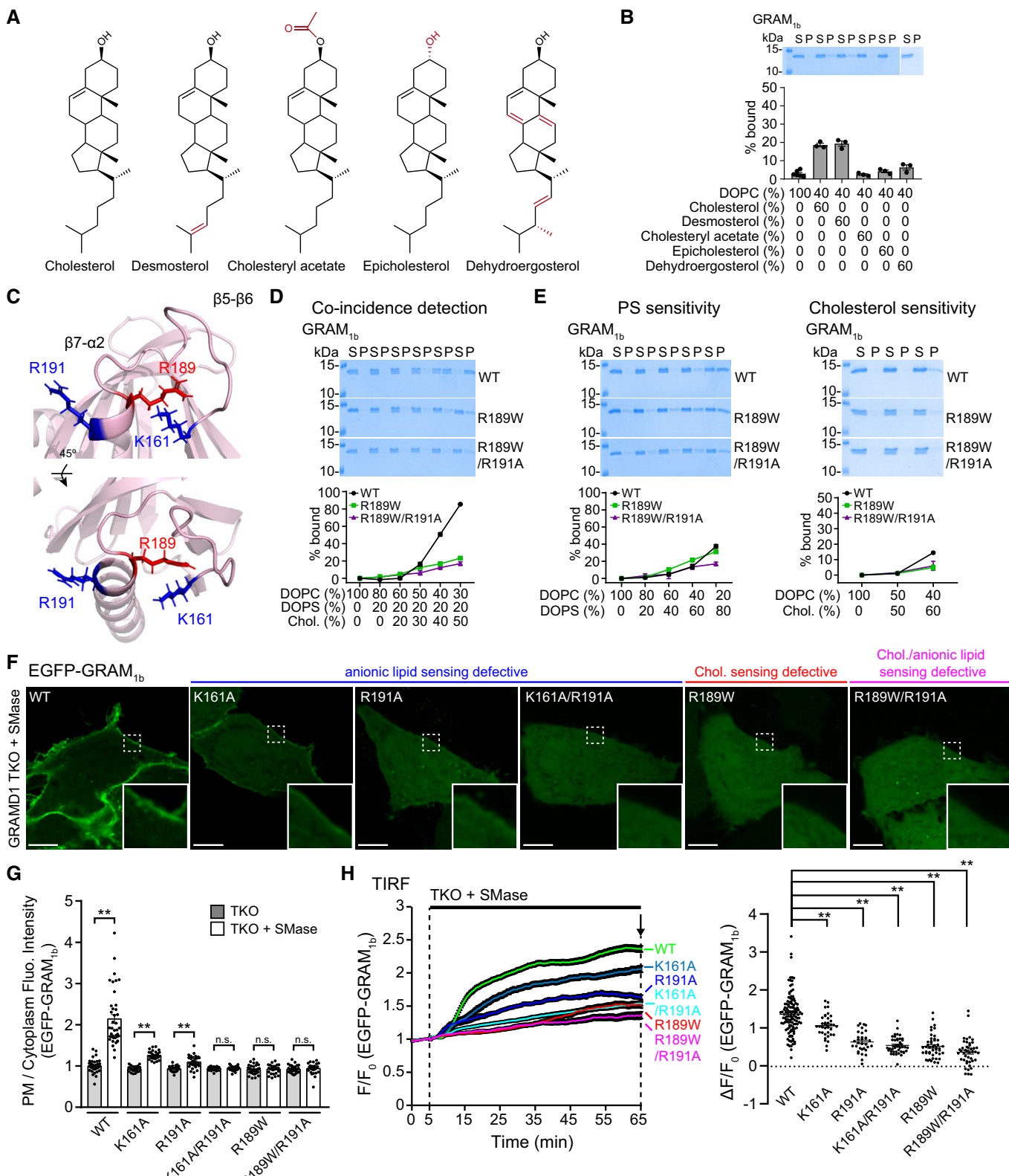

**Figure 3.**

of GRAM$_{1b}$ that carried the R189W mutation (GRAM$_{1b}$–R189W) and examined its ability to bind membranes containing both cholesterol and PS (Fig 3D). Strikingly, GRAM$_{1b}$–R189W bound much less

efficiently to liposomes compared to control GRAM$_{1b}$ (~22% binding compared to ~90% binding for the WT control when liposomes contained 20% PS and 50% cholesterol). This indicates a critical

**Figure 3.  A GRAMD1b mutation (R189W) associated with intellectual disability specifically impairs binding of the GRAM domain to cholesterol.**

A    Chemical structures of the sterols tested in (B).
B    Sterol specificity of the GRAM domain of GRAMD1b ($GRAM_{1b}$) as assessed via liposome sedimentation assays. Liposomes containing the indicated mole% lipids were incubated with purified $GRAM_{1b}$ proteins as indicated. Bound proteins [pellet, (P)] were separated from the unbound proteins [supernatant, (S)], run on SDS–PAGE and visualized by colloidal blue staining (mean ± SEM, $n = 6$ independent experiments for liposomes containing only DOPC, $n = 3$ independent experiments for the rest). DOPC, phosphatidylcholine (1,2-dioleoyl-sn-glycero-3-phosphocholine).
C    Magnified ribbon diagrams of the modeled $GRAM_{1b}$ showing amino acid side chains that contribute to anionic lipid sensitivity (K161, R191) (blue) and an amino acid side chain of R189 (red), whose substitution to tryptophan (W) is associated with intellectual disability in humans. The bottom panel shows the diagram that was rotated 45° around the indicated axis as shown.
D, E   Liposome sedimentation assays of wild-type $GRAM_{1b}$ proteins (WT) and mutant versions of $GRAM_{1b}$ proteins carrying indicated mutations (R189W, R189W/R191A). Liposomes containing the indicated mole% lipids were incubated with purified $GRAM_{1b}$ proteins as shown. Bound proteins [pellet, (P)] were separated from the unbound proteins [supernatant, (S)], run on SDS–PAGE and visualized by colloidal blue staining [mean ± SEM, $n = 3$ independent experiments for all conditions in the coincidence detection assay; $n = 3$ independent experiments for R189W and R189W/R191A, $n = 6$ independent experiments for WT in phosphatidylserine (PS) sensitivity assay; $n = 3$ independent experiments for R189W and R189W/R191A, $n = 5$ independent experiments for WT in cholesterol sensitivity assay]. DOPC, phosphatidylcholine (1,2-dioleoyl-sn-glycero-3-phosphocholine); DOPS, phosphatidylserine (1,2-dioleoyl-sn-glycero-3-phospho-L-serine); Chol., cholesterol.
F    Confocal images of live GRAMD1 TKO HeLa cells expressing either wild-type (WT) or mutant versions of EGFP-tagged GRAM domain of GRAMD1b (EGFP-$GRAM_{1b}$) constructs as indicated. Cells were treated with SMase (100 mU/ml for 1 h at 37°C) before imaging. Insets show at higher magnification the regions indicated by white dashed boxes. Anionic lipid-sensing defective mutants (K161A, R191A, and K161A/R191A), a cholesterol-sensing defective mutant (R189W), and a cholesterol/anionic lipid-sensing defective mutant (R189W/R191A) are shown. Note the absence of (R189W, R189W/R191A, and K161A/R191A) or significantly reduced (K161A, R191A) PM recruitment of the mutant versions of EGFP-$GRAM_{1b}$ compared to the strong PM recruitment of wild-type EGFP-$GRAM_{1b}$ (WT). Scale bars, 10 μm.
G    Quantification of the ratio of PM EGFP-$GRAM_{1b}$ signals to the cytosolic EGFP-$GRAM_{1b}$ signals, as assessed by confocal microscopy and line scan analysis from GRAMD1 TKO HeLa cells expressing either wild-type (WT) or mutant versions of EGFP-$GRAM_{1b}$ with or without SMase treatment (100 mU/ml for 1 h at 37°C) as shown in (F) and Appendix Fig S1B (mean ± SEM, $n = 40$ cells for WT, and $n = 30$ cells for other conditions; data are pooled from four independent experiments for WT and three independent experiments for the rest; two-tailed unpaired Student's $t$-test, $**P < 0.0001$. n.s. denotes not significant).
H    Left: Time course of normalized EGFP signal, as assessed by TIRF microscopy, from GRAMD1 TKO HeLa cells expressing either wild-type (WT) or mutant versions of EGFP-$GRAM_{1b}$ constructs as indicated. SMase treatment (100 mU/ml) is indicated. Right: Values of $\Delta F/F_0$ corresponding to the end of the experiment as indicated by the arrow [mean ± SEM, $n = 126$ cells (WT), $n = 38$ cells (K161A), $n = 33$ cells (R191A), $n = 43$ cells (K161A/R191A), $n = 46$ cells (R189W), $n = 44$ cells (R189W/R191A); data are pooled from six independent experiments for WT and two independent experiments for the rest; Dunnett's multiple comparisons test, $**P = 0.0002$ (WT versus K161A) and $**P < 0.0001$ for the rest]. See also Movie EV1.

Source data are available online for this figure.

role of the R189 residue in the coincidence detection of cholesterol and anionic lipids, including PS, in membranes (Fig 3D).

We then asked whether the R189W mutation specifically affected cholesterol binding, PS binding, or both. We compared binding of WT control $GRAM_{1b}$ and $GRAM_{1b}$–R189W to liposomes containing only PS (0–80%) or only cholesterol (0%, 50%, or 60%). Control and mutant $GRAM_{1b}$ bound similarly to liposomes containing PS (Fig 3E). However, $GRAM_{1b}$–R189W bound less efficiently to liposomes containing 60% cholesterol and no PS (~4% binding compared to ~15% for WT controls) (Fig 3E). Thus, the R189W mutation specifically impaired the ability of $GRAM_{1b}$ to sense cholesterol without affecting its affinity for PS.

Mutations within the $GRAM_{1b}$ basic patch (K161A and R191A) specifically impaired anionic lipid binding (Fig 2C–E), whereas the R189W mutation specifically impaired cholesterol binding (Fig 3D and E). These results are consistent with the notion that anionic lipid and cholesterol recognition are mediated by independent sites. We generated a mutant version of $GRAM_{1b}$ that carried the R191A basic patch mutation (defective in anionic lipid sensing) and the R189W mutation (defective in cholesterol sensing). Purified $GRAM_{1b}$ carrying the R189W/R191A double mutation ($GRAM_{1b}$–R189W/R191A) bound slightly less efficiently to liposomes containing cholesterol and PS than $GRAM_{1b}$–R189W (~17% binding when liposomes contained 20% PS and 50% cholesterol), thereby suggesting that cholesterol-sensing and anionic lipid-sensing properties of the GRAM domain act independently (Fig 3D and E). Binding of $GRAM_{1b}$–R189W/R191A to liposomes containing only PS was significantly reduced compared to $GRAM_{1b}$–R189W or $GRAM_{1b}$, as predicted based on the importance of R191 in anionic lipid recognition. No further reduction in binding to liposomes

containing only cholesterol was observed when the R191A mutation was added (Fig 3E). Taken together, our results reveal the presence of distinct recognition sites for anionic lipids and cholesterol within $GRAM_{1b}$. Further, these sites act synergistically to detect the codistribution of cholesterol and anionic lipids in membranes.

**Both cholesterol- and anionic lipid-sensing properties are required for the GRAM domain of GRAMD1b to sense expansions of the accessible pool of PM cholesterol**

The impact of various GRAM domain mutations on the ability to sense accessible PM cholesterol was further examined in GRAMD1 TKO HeLa cells. GRAMD1 TKO HeLa cells expressing either WT EGFP-$GRAM_{1b}$, anionic lipid-sensing defective EGFP-$GRAM_{1b}$ mutants (K161A, R191A, and K161A/R191A), a cholesterol-sensing defective EGFP-$GRAM_{1b}$ mutant (R189W), or an anionic lipid- and cholesterol-sensing defective EGFP-$GRAM_{1b}$ mutant (R189W/R191A), were treated with sphingomyelinase and then monitored using spinning disk confocal (SDC) microscopy. The recruitment of EGFP-$GRAM_{1b}$ to the PM was assessed by line scan analysis. At steady state, all EGFP-$GRAM_{1b}$ variants were distributed throughout the cytosol (Appendix Fig S1B). Treating GRAMD1 TKO cells with sphingomyelinase for 1 h led to the recruitment of EGFP-$GRAM_{1b}$ to the PM due to the expansion of the accessible pool of PM cholesterol (Fig 3F and G). Anionic lipid-sensing defective EGFP-$GRAM_{1b}$ mutants that exhibit some anionic lipid binding (K161A and R191A) were partially recruited to the PM following sphingomyelinase treatment. In contrast, the same treatment failed to recruit anionic lipid-sensing defective (K161A/R191A), cholesterol-sensing defective

(R189W) or cholesterol- and anionic lipid-sensing defective (R189W/R191A) EGFP-GRAM$_{1b}$ mutants (Fig 3F and G).

To examine the kinetics by which EGFP-GRAM$_{1b}$ variants were recruited to the PM, these cells were treated with sphingomyelinase and imaged using TIRF microscopy. Within 30 min of sphingomyelinase treatment, robust PM recruitment of EGFP-GRAM$_{1b}$ was observed. However, the K161A/R191A, R189W, and R189W/R191A mutants failed to localize to the PM, even after 60 min (Fig 3H, Appendix Fig S1C, Movie EV1). The K161A and R191A mutants were still recruited to the PM, albeit less efficiently compared to control EGFP-GRAM$_{1b}$ (Fig 3H, Appendix Fig S1C). These results demonstrate the importance of both cholesterol-sensing and anionic lipid-sensing properties in the ability of GRAM$_{1b}$ to detect expansions of the accessible pool of PM cholesterol and suggest that the GRAM domain plays a critical role in regulating GRAMD1-mediated transport of accessible cholesterol.

## Mutant versions of GRAMD1b that cannot sense cholesterol exhibit impairment in membrane tethering and cholesterol transport

GRAMD1s sense transient expansions in the accessible pool of PM cholesterol via the GRAM domain and then transport accessible cholesterol from the PM to the ER at ER-PM contacts via the StART-like domain (Naito *et al*, 2019). How interactions between the GRAM domain and accessible PM cholesterol impacts their cholesterol transport functions remain unclear. To examine the importance of the GRAM domain in StART-like domain-dependent cholesterol transport, we reconstituted this process using purified near full-length GRAMD1b proteins and liposomes that mimicked the PM and ER membranes. We prepared: (i) PM-like donor liposomes (L$_{PM}$) that contained PS, cholesterol, Dansyl-PE (DNS-PE), and DHE (a fluorescent analog of cholesterol), and (ii) ER-like acceptor liposomes (L$_{ER}$) that contained Nickel-labeled lipids (DGS-NTA) (see Materials and Methods) (Fig 4A). The near full-length versions of GRAMD1b [GRAMD1b–WT or GRAMD1b carrying GRAM domain mutations (GRAMD1b–R189W, GRAMD1b–R189W/R191A)] consisted of an N-terminal GRAM domain followed by a StART-like domain and a C-terminal His-tag that replaced the ER-anchoring transmembrane domain. We mixed purified GRAMD1b proteins with PM-like liposomes and ER-like liposomes (Fig 4A and Appendix Fig S2A–C). The ER-like liposomes were bound to the GRAMD1b proteins via His-tag-Nickel interactions whereas their GRAM domains facilitated tethering of the ER-like liposomes to the PM-like liposomes (Fig 4A). The transfer of DHE from PM-like to ER-like liposomes was monitored by measuring FRET between DHE and DNS-PE in the PM-like liposomes (Appendix Fig S2D). At the same time, GRAM domain-dependent tethering of PM-like and ER-like liposomes was monitored by measuring turbidity, which reflects the clustering of liposomes. Methyl-β-cyclodextrin (MCD) was used to determine cholesterol equilibrium between liposomes.

The addition of near full-length GRAMD1b protein (GRAMD1b–WT) resulted in an increase in optical turbidity over time, consistent with the progressive tethering of PM-like and ER-like liposomes (Fig 4B). In contrast, the addition of GRAMD1b–R189W (defective in sensing cholesterol) or GRAMD1b–R189W/R191A (defective in sensing both cholesterol and anionic lipids) failed to increase optical turbidity despite their equally robust interaction with ER-like

liposomes compared to GRAMD1b–WT (Appendix Fig S3E), indicating the failure of liposome tethering (Fig 4B). Consistent with the progressive tethering of PM-like and ER-like liposomes, the addition of GRAMD1b–WT, but not GRAMD1b–R189W or GRAMD1b–R189W/R191A, resulted in robust DHE transfer from PM-like to ER-like liposomes over time in a concentration-dependent manner (~1.2 DHE molecules per min) (Fig 4C and Appendix Fig S2E and F). Thus, the R189W mutation, which specifically impairs the cholesterol-sensing property of the GRAM domain and has been associated with intellectual disability in humans, affects the ability of GRAMD1b to transfer cholesterol. These results demonstrate that the cholesterol-sensing property of the GRAM domain plays a critical role in StART-like domain-dependent cholesterol transport via facilitation of membrane tethering.

We also purified near full-length versions of GRAMD1b carrying anionic lipid-sensing defective GRAM domain mutations (GRAMD1b–R191A, GRAMD1b–K161A/R191A) (Appendix Fig S3A and B) and examined their ability to tether membranes and transfer cholesterol *in vitro*. Both of them interacted equally robustly to ER-like liposomes compared to GRAMD1b–WT (Appendix Fig S3E). Surprisingly, the addition of GRAMD1b–R191A or GRAMD1b–K161A/R191A resulted in an increase in optical turbidity and robust DHE transfer from PM-like to ER-like liposomes over time, similar to GRAMD1b–WT (Appendix Fig S3C and D), suggesting efficient membrane tethering and cholesterol transport mediated by these mutant near full-length GRAMD1b proteins *in vitro*. R191A and K161A/R191A mutations strongly impair the property of the isolated GRAM domain to sense anionic lipids *in vitro*; these mutations also impair the ability of the GRAM domain to interact with the PM *in vivo* (Figs 2C and D, and 3F–H, Appendix Fig S1C). Thus, the results from the *in vitro* lipid transport assay suggest that there are other amino acids outside of the GRAM domain that may contribute to the anionic lipid-sensing property of full-length GRAMD1b proteins (see Discussion).

## Cholesterol-sensing property of the GRAM domain is critical for GRAMD1b function

The GRAMD1b–R189W and GRAMD1b–R189W/R191A mutants were unable to tether membranes and could not transfer cholesterol between membranes *in vitro* (Fig 4). Thus, we hypothesized that the cholesterol-sensing property of the GRAM domain is critical for the localization and cellular function of GRAMD1b (and more generally of GRAMD1s). To test this hypothesis, we asked whether GRAM$_{1b}$ mutations that render it insensitive to cholesterol (R189W or R189W/R191A) affect the ability of GRAMD1b to localize to ER-PM contacts and transport cholesterol from the PM to the ER.

Through transduction of lentiviruses that express individual constructs (see Materials and Methods), we generated HeLa cells that stably expressed EGFP, as well as GRAMD1 TKO HeLa cells that stably expressed EGFP, EGFP-tagged GRAMD1b (EGFP-GRAMD1b), EGFP-GRAMD1b–R189W, or EGFP-GRAMD1b–R189W/R191A. Immunoblot analysis of total cell lysates from these newly established cell lines confirmed similar levels of the transduced proteins. As assessed by an antibody against GRAMD1b, levels of the transduced EGFP-GRAMD1b proteins were generally higher than that of endogenous GRAMD1b proteins (Fig 5A). Stably expressed EGFP-GRAMD1b proteins localized to the ER as discrete patches, similar

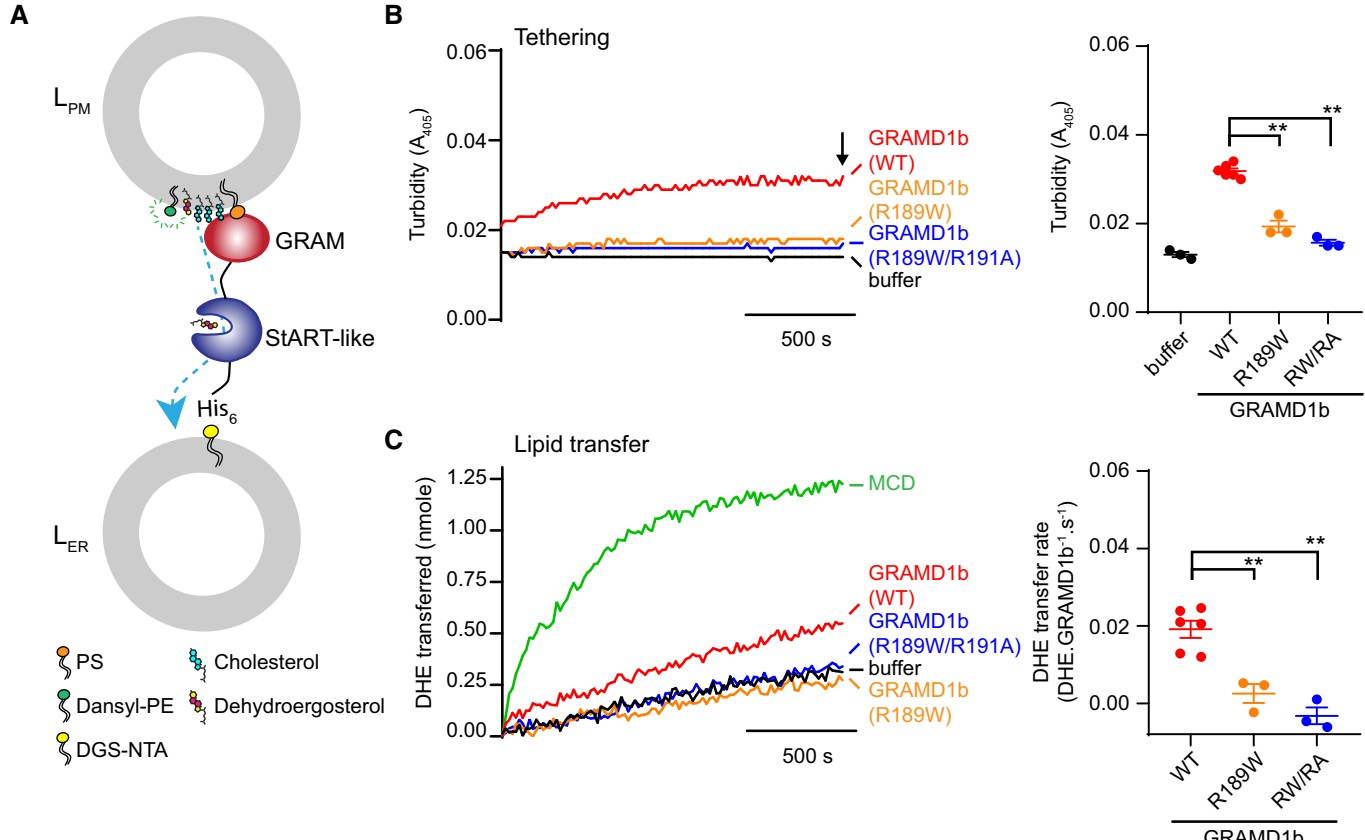

**Figure 4. GRAMD1b carrying an intellectual disability mutation (R189W) is defective in membrane tethering and cholesterol transport *in vitro*.**

A   Schematic of the dehydroergosterol (DHE) transfer assay *in vitro*. Plasma membrane-like liposomes ($L_{PM}$) [10% phosphatidylserine (PS: DOPS), 30% cholesterol, 2.5% Dansyl-phosphatidylethanolamine (DNS-PE), 10% dehydroergosterol (DHE), 47.5% phosphatidylcholine (DOPC)] and endoplasmic reticulum-like liposomes ($L_{ER}$) [15% DGS-NTA (Ni), 20% phosphatidylethanolamine (1-palmitoyl-2-oleoyl-sn-glycero-3-phosphoethanolamine/POPE), 65% DOPC] were incubated with either purified wild-type GRAMD1b proteins (WT) or purified mutant GRAMD1b proteins carrying GRAM domain mutations (1 mM total lipid, 0.2 μM final protein concentrations in 50 μl reaction volume unless otherwise noted). C-terminus of GRAMD1b, including the transmembrane domain, was replaced with a hexahistidine tag (His-tag) to attach purified GRAMD1b proteins to $L_{ER}$ via His-tag DGS-NTA(Ni) interaction. A fluorometer was used to simultaneously monitor tethering of the liposomes in (B) and transfer of DHE from $L_{PM}$ to $L_{ER}$ in (C).

B   Liposome tethering by purified GRAMD1b proteins. Left: Representative time course of liposome tethering, as assessed by turbidity of mixtures containing both $L_{PM}$ and $L_{ER}$ together with either wild-type GRAMD1b proteins (WT), mutant GRAMD1b R189W proteins (R189W), mutant GRAMD1b R189W/R191A proteins (RW/RA), or buffer alone as indicated. Turbidity, as assessed by measuring absorbance values at 405 nm, reflects clustering of liposomes due to tethering of $L_{PM}$ and $L_{ER}$. Right: Values of turbidity corresponding to the end of the experiment as indicated by the arrow (mean ± SEM, n = 6 independent experiments for WT, n = 3 independent experiments for buffer alone, R189W, and RW/RA; Dunnett's multiple comparisons test, **P < 0.0001).

C   DHE transfer between liposomes by purified GRAMD1b proteins. Left: Representative time course of DHE transfer from $L_{PM}$ to $L_{ER}$ mediated by either wild-type GRAMD1b proteins (WT), mutant GRAMD1b R189W proteins (R189W), mutant GRAMD1b R189W/R191A proteins (RW/RA), or buffer alone as indicated. Methyl-β-cyclodextrin (MCD) at 1 mM was used to determine DHE equilibration. Transfer of DHE from $L_{PM}$ to $L_{ER}$ resulted in a decrease in fluorescence resonance energy transfer (FRET) between DHE and DNS-PE in $L_{PM}$. A series of liposomes with different DHE mol% were prepared to plot a calibration curve to convert FRET signals to DHE molecules transferred (see Appendix Fig S2D and Materials and Methods). Right: Values of DHE transfer rate of GRAMD1b (WT), GRAMD1b (R189W), and GRAMD1b (RW/RA), as estimated by the FRET-based lipid transfer assay [mean ± SEM, n = 6 independent experiments for WT, n = 3 independent experiments for R189W and RW/RA; Dunnett's multiple comparisons test, **P = 0.0018 (WT versus R189W), **P = 0.0002 (WT versus RW/RA)].

to previously reported localization of transiently transfected and functional EGFP-GRAMD1b (Naito *et al*, 2019), allowing us to compare the activities of various versions of EGFP-GRAMD1b proteins at similar levels of expression in GRAMD1 TKO cells (Fig 5B). Cholesterol-dependent recruitment of EGFP-GRAMD1b to ER-PM contacts was analyzed using TIRF microscopy by incubating cells with a cholesterol/methyl-β-cyclodextrin complex (i.e., cholesterol loading). Robust PM recruitment of EGFP-GRAMD1b was observed within 20 min of cholesterol loading (Fig EV3A, Movie

EV2). In contrast, mutant versions of EGFP-GRAMD1b (R189W and R189W/R191A) were recruited less efficiently to the PM, consistent with the essential role of cholesterol-sensing properties of the GRAM domain in mediating cholesterol-dependent PM recruitment of GRAMD1b (Fig EV3A, Movie EV2).

Sphingomyelinase treatment greatly expands the accessible pool of PM cholesterol in GRAMD1 TKO cells due to inefficient transport of newly liberated accessible cholesterol from the PM to the ER (Naito *et al*, 2019). We transiently expressed an accessible PM

cholesterol biosensor, mCherry-tagged GRAM$_{1b}$ (mCherry-GRAM$_{1b}$), in GRAMD1 TKO cells and control HeLa cells, and visualized accessible cholesterol accumulation in the PM following 1 h of sphingomyelinase treatment. mCherry-GRAM$_{1b}$ was more strongly recruited to the PM in GRAMD1 TKO cells compared to control HeLa cells (Fig EV3B). PM recruitment of mCherry-GRAM$_{1b}$ following the treatment of GRAMD1 TKO cells with sphingomyelinase was suppressed by the re-expression of EGFP-GRAMD1b, but not by the re-expression of EGFP-GRAMD1b–R189W or EGFP-GRAMD1b–R189W/R191A, as assessed by SDC microscopy (Fig 5B and C) or TIRF microscopy over time (Fig 5D). We previously showed that PM recruitment of the GRAM$_{1b}$ cholesterol biosensor following the treatment of GRAMD1 TKO cells with sphingomyelinase was suppressed by the re-expression of GRAMD1b, but not by the re-expression of GRAMD1b carrying mutations in the StART-like

domain that cannot transport cholesterol (despite robust recruitment of this mutant protein to the PM upon sphingomyelinase treatment) (Naito *et al*, 2019), showing the specificity of this assay in examining cholesterol transporting property of GRAMD1b. Thus, these results suggest important roles of the cholesterol-sensing properties of GRAM$_{1b}$ in regulating GRAMD1b-mediated transport of accessible cholesterol in cells.

Using SREBP-2 cleavage to estimate the efficiency of accessible cholesterol transport from the PM to the ER, we previously showed that levels of SREBP-2 cleavage are higher in GRAMD1 TKO cells than control cells following sphingomyelinase treatment (Naito *et al*, 2019). Here we examined whether such defects in accessible cholesterol transport are rescued by mutant versions of EGFP-GRAMD1b. We first depleted most of the accessible cholesterol from cells by treating them with a combination of lipoprotein-deficient

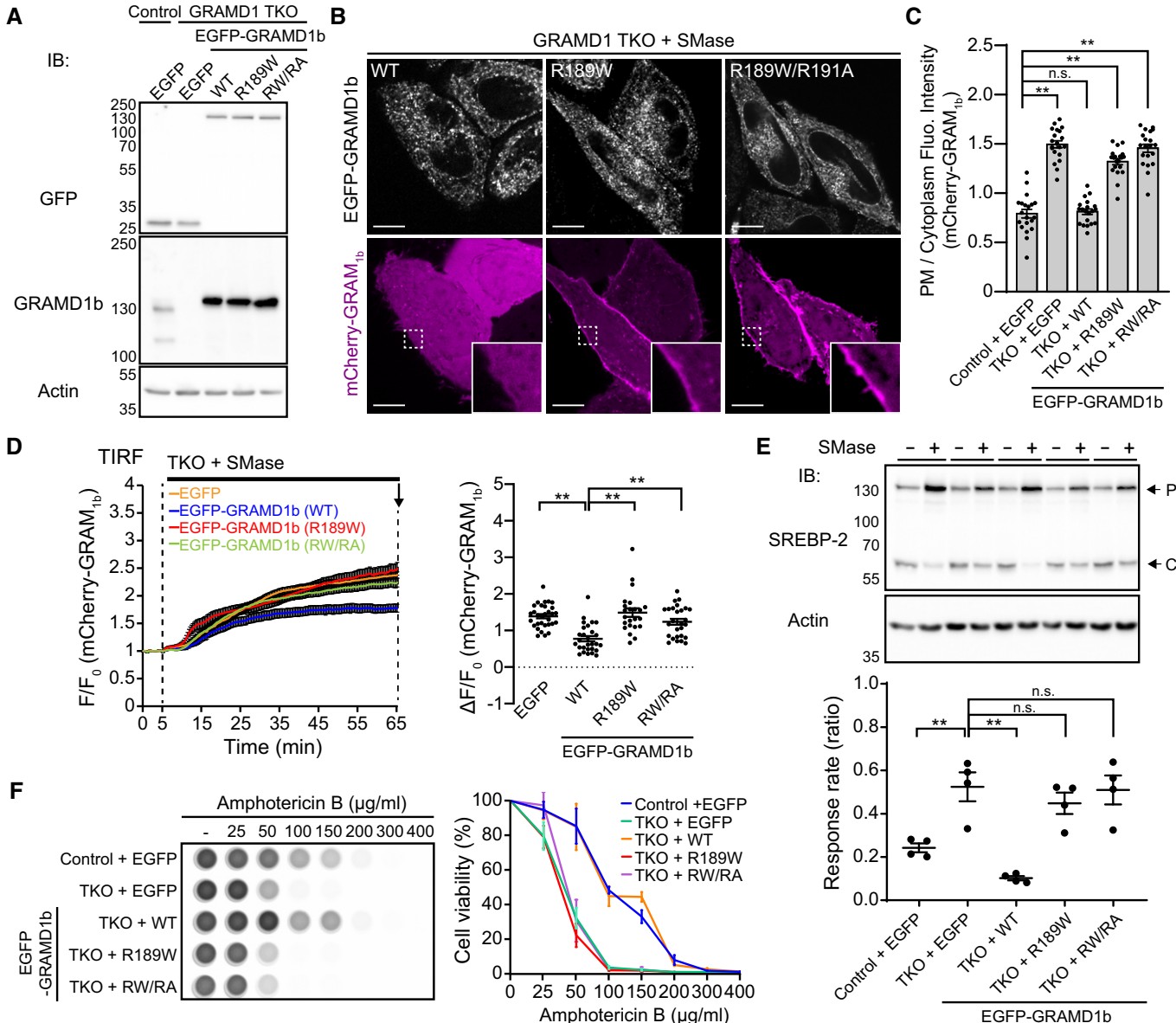

**Figure 5.**

◀

**Figure 5. Cholesterol-sensing property of the GRAM domain is critical for GRAMD1b function.**

A   Lysates of wild-type (control) and GRAMD1 TKO HeLa cells that stably expressed either EGFP or EGFP-tagged GRAMD1b (EGFP-GRAMD1b) constructs as indicated [wild-type (WT), R189W mutant (R189W), R189W/R191A mutant (RW/RA)] were processed by SDS–PAGE and immunoblotted (IB) with anti-GFP, anti-GRAMD1b, and anti-actin antibodies.

B   Confocal images of live GRAMD1 TKO HeLa cells stably expressing EGFP-GRAMD1b constructs as indicated that were additionally transfected with an accessible PM cholesterol biosensor, mCherry-tagged GRAM domain of GRAMD1b (mCherry-GRAM$_{1b}$). Cells were treated with SMase (100 mU/ml for 1 h at 37°C) before imaging. Insets show at higher magnification the regions indicated by white dashed boxes. Note the very weak PM recruitment of mCherry-GRAM$_{1b}$ in cells that stably expressed EGFP-GRAMD1b WT, compared to the strong PM recruitment of mCherry-GRAM$_{1b}$ in cells that stably expressed EGFP-GRAMD1b R189W mutant (R189W) or EGFP-GRAMD1b R189W/R191A mutant (RW/RA). Scale bars, 10 μm.

C   Quantification of PM mCherry-GRAM$_{1b}$ signals to the cytosolic mCherry-GRAM$_{1b}$ signals, as assessed by confocal microscopy and line scan analysis from GRAMD1 TKO (TKO) HeLa cells expressing mCherry-GRAM$_{1b}$ with SMase treatment (100 mU/ml for 1 h at 37°C) as shown in (B) and Fig EV3B (mean ± SEM, $n = 20$ cells for each condition; data are pooled from two independent experiments; Dunnett's multiple comparisons test, **$P < 0.0001$. n.s. denotes not significant).

D   Left: Time course of normalized mCherry signal, as assessed by TIRF microscopy, from GRAMD1 TKO (TKO) HeLa cells stably expressing either EGFP or EGFP-GRAMD1b constructs as indicated that were additionally transfected with an accessible PM cholesterol biosensor mCherry-GRAM$_{1b}$. SMase treatment (100 mU/ml) is indicated. Right: Values of $\Delta F/F_0$ corresponding to the end of the experiment as indicated by the arrow [mean ± SEM, $n = 31$ cells (EGFP), $n = 28$ cells (EGFP-GRAMD1b WT), $n = 23$ cells (EGFP-GRAMD1b R189W), $n = 27$ cells [EGFP-GRAMD1b R189W/R191A (RW/RA)]; data are pooled from two independent experiments for each condition; Dunnett's multiple comparisons test, **$P < 0.0001$ (EGFP versus WT and WT versus R189W), **$P = 0.0002$ (WT versus RW/RA)].

E   Wild-type (control) and GRAMD1 TKO (TKO) HeLa cells that stably expressed either EGFP or EGFP-GRAMD1b constructs as indicated [wild-type (WT), R189W mutant (R189W), R189W/R191A mutant (RW/RA)], were cultured in the medium supplemented with 10% lipoprotein-deficient serum (LPDS) and mevastatin (50 μM) for 16 h and then treated with SMase (100 mU/ml) for 3 h at 37°C. Top: Lysates of the cells were processed for SDS–PAGE and IB with anti-SREBP-2 and anti-actin antibodies. Arrows indicate precursor (P) and cleaved (C) forms of SREBP-2. Bottom: The response rate was obtained by normalizing the ratio of the band intensity of the cleaved SREBP-2 over the total band intensity of cleaved and precursor forms of SREBP-2 from the cells with SMase treatment by the one from the cells without SMase treatment for each condition. Note that the suppression of SREBP-2 cleavage is attenuated in GRAMD1 TKO HeLa cells compared to wild-type control HeLa cells. Note also the rescue by expression of wild-type EGFP-GRAMD1b (WT) but not by mutant versions of EGFP-GRAMD1b [R189W, R189W/R191A (RW/RA)] [mean ± SEM, $n = 4$ lysates (independent experiments) for each condition; Dunnett's multiple comparisons test, **$P = 0.0035$ (Control + EGFP versus TKO + EGFP), **$P < 0.0001$ (TKO + EGFP versus TKO + WT), n.s. denotes not significant].

F   Amphotericin B resistance of SMase-treated wild-type (control) and GRAMD1 TKO (TKO) HeLa cells that stably expressed either EGFP or EGFP-GRAMD1b constructs as indicated [wild-type (WT), R189W mutant (R189W), R189W/R191A mutant (RW/RA)]. Left: Cells that had been pre-treated with SMase (100 mU/ml) for 3 h at 37°C were treated with indicated concentration of Amphotericin B for 20 min at 37°C. After overnight recovery in culture media, cell viability was measured by detecting ATP present in each well via luminescence (see Materials and Methods). The same number of cells were seeded in each well before SMase treatment. Note the reduced viability of cells with increasing amount of Amphotericin B. Right: Quantification of cell viability with increasing amount of Amphotericin B. Note the resistance of wild-type control and GRAMD1 TKO cells that stably expressed EGFP-GRAMD1b (WT) compared to GRAMD1 TKO cells or GRAMD1 TKO cells that stably expressed mutant versions of EGFP-GRAMD1b [R189W, R189W/R191A (RW/RA)] (mean ± SEM, $n = 3$ independent experiments for each condition).

Source data are available online for this figure.

serum (LPDS) and mevastatin, an HMG-CoA reductase inhibitor, for 16 h (a condition that leads to maximum SREBP-2 cleavage in both control and GRAMD1 TKO cells by cholesterol starvation). We then treated the cells with sphingomyelinase and used total cell lysates to monitor the suppression of SREBP-2 cleavage, which results from the PM to ER transport of accessible PM cholesterol. Cell lysates were collected before and 3 h after sphingomyelinase treatment and analyzed by SDS–PAGE followed by immunoblotting against SREBP-2 (Fig 5E). Suppression of SREBP-2 cleavage was observed in control cell lysates; however, this suppression was significantly attenuated (but not completely eliminated) in GRAMD1 TKO cells. Re-expression of EGFP-GRAMD1b in TKO cells robustly suppressed SREBP-2 cleavage, thereby rescuing the phenotype (Fig 5E). However, the expression of EGFP-GRAMD1b–R189W or EGFP-GRAMD1b–R189W/R191A in TKO cells failed to rescue the phenotype (Fig 5E). This is consistent with the inefficient GRAMD1-dependent transport of cholesterol observed with these mutations *in vitro* (Fig 4C).

Yeast mutants that lack GRAMD1 homologs (Lam/Ltc proteins) are highly susceptible to treatment with the polyene antibiotic, Amphotericin B, compared to WT cells (Gatta *et al*, 2015; Marek *et al*, 2020). Amphotericin B binds PM sterols and eventually causes cell death by various mechanisms, including sequestration of PM sterols and the formation of non-selective ion pores (Kinsky, 1970; Wang *et al*, 2018). We found that GRAMD1 TKO cells pre-treated with sphingomyelinase for 3 h were more efficiently killed by Amphotericin B compared to WT cells subjected to the same

treatment (Fig 5F). These results suggest that Amphotericin B preferentially targets accessible cholesterol in the PM of mammalian cells, further supporting exaggerated expansions in the accessible pool of PM cholesterol in GRAMD1 TKO cells following sphingomyelinase treatment. The sensitivity of GRAMD1 TKO cells to Amphotericin B was dramatically reduced by re-expressing EGFP-GRAMD1b, but not by re-expressing EGFP or EGFP-GRAMD1b mutants (Fig 5F), demonstrating that GRAMD1b plays a direct role in modulating sensitivity of the PM to Amphotericin B, and that this function requires a GRAM domain that can sense accessible PM cholesterol. Taken together, these data confirm the importance of GRAMD1 in the transport of accessible cholesterol from the PM to the ER in response to acute expansion of the accessible pool of PM cholesterol, thereby preventing the overaccumulation of cholesterol in the PM. Further, our data show that this function requires a GRAM domain that can sense accessible cholesterol, which is essential for the recruitment of GRAMD1 to ER-PM contact sites.

## Identification of a novel GRAM domain mutation in GRAMD1b (G187L) that enhances its ability to sense accessible PM cholesterol

To gain further insights into the mechanisms of ligand recognition by GRAM$_{1b}$, we mutated amino acids that were within ~4–14 Å of the R189 residue. The goal was to identify amino acid residues whose mutation increases sensitivity of GRAM$_{1b}$ to cholesterol. Selected amino acids (R166, Q173, F182, T184, F186, G187, and

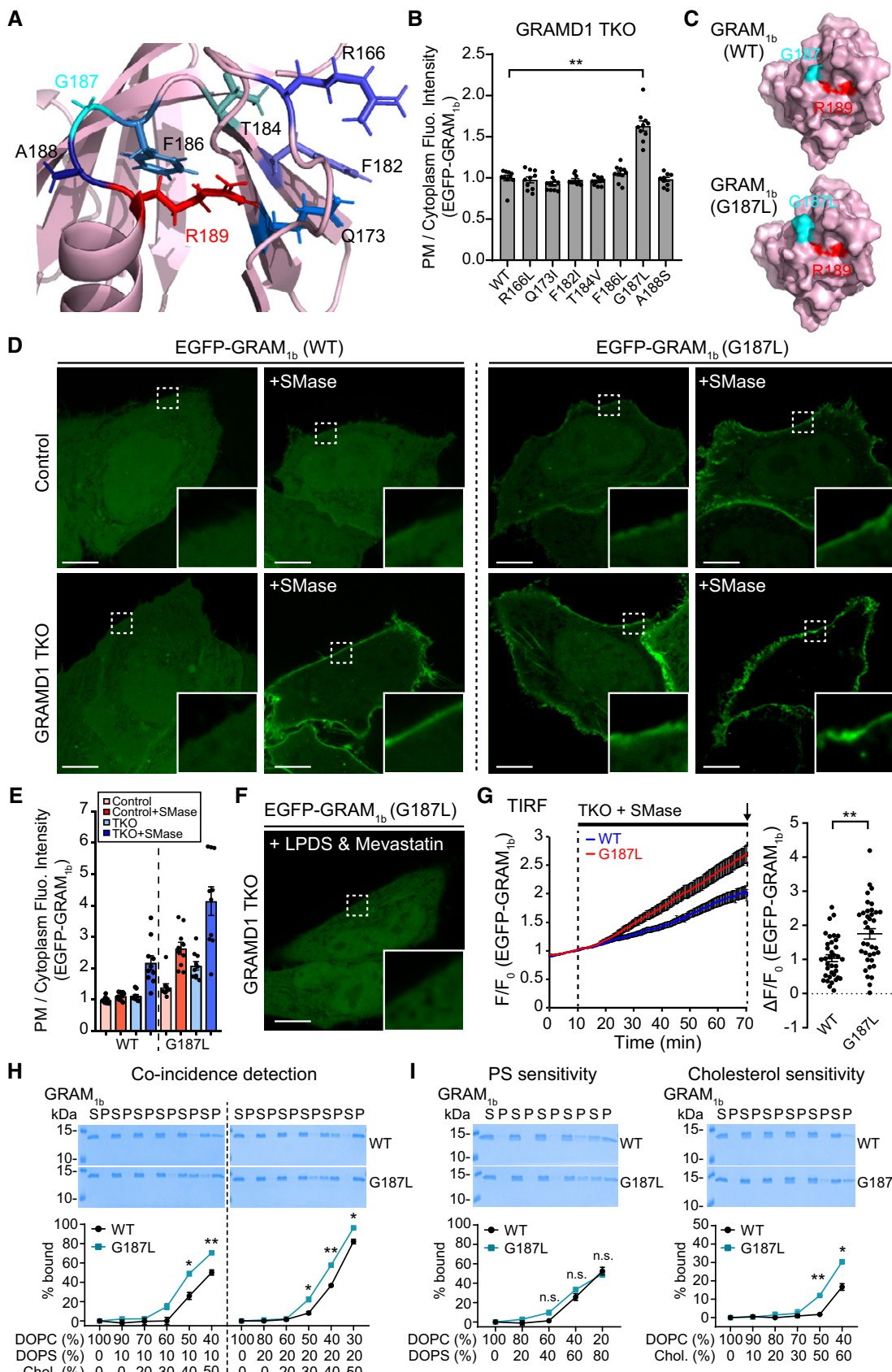

Figure 6.

◀

**Figure 6. G187L mutation specifically increases the cholesterol sensitivity of the GRAM domain of GRAMD1b.**

A   Close-up view of the ribbon diagram of the modeled GRAM domain of GRAMD1b (GRAM$_{1b}$) showing amino acid side chains of R189 and other amino acids, lying in the ~4–14 Å vicinity of R189, that were mutated in mini-mutagenesis screen as shown in (B).

B   Quantification of the ratio of PM signals to the cytosolic signals of wild-type EGFP-GRAM$_{1b}$ (WT) and mutant versions of EGFP-GRAM$_{1b}$ (R166L, Q173I, F182I, T184V, F186L, G187L, A188S), as assessed by confocal microscopy and line scan analysis from GRAMD1 TKO HeLa cells, expressing indicated constructs, as shown in (D) and Fig EV4A (mean ± SEM, n = 10 cells for each condition; data are pooled from one experiment; Dunnett's multiple comparisons test, **P < 0.0001).

C   Surface representations of the modeled wild-type GRAM$_{1b}$ (WT) and mutant GRAM$_{1b}$ carrying G187L mutation (G187L) showing the close proximity of G187 (or G187L) and R189 residues.

D   Confocal images of live wild-type (control) and GRAMD1 TKO HeLa cells, expressing either wild-type EGFP-GRAM$_{1b}$ (WT) or mutant EGFP-GRAM$_{1b}$ (G187L), with or without SMase treatment (100 mU/ml for 1 h at 37°C). Insets show at higher magnification the regions indicated by white dashed boxes. Note the recruitment of mutant EGFP-GRAM$_{1b}$ (G187L) to the PM even in wild-type HeLa cells at rest, which was further enhanced by SMase treatment. Scale bars, 10 μm.

E   Quantification of the ratio of PM signals to the cytosolic signals of wild-type EGFP-GRAM$_{1b}$ (WT) and mutant EGFP-GRAM$_{1b}$ (G187L), as assessed by confocal microscopy and line scan analysis from GRAMD1 TKO HeLa cells, expressing indicated constructs, with or without SMase treatment (100 mU/ml for 1 h at 37°C), as shown in (D) (mean ± SEM, n = 10 cells for each condition; data are pooled from one experiment).

F   A confocal image of live GRAMD1 TKO HeLa cells expressing mutant EGFP-GRAM$_{1b}$ (G187L). Cells were cultured in the medium supplemented with 10% lipoprotein-deficient serum (LPDS) and mevastatin (50 μM) for 16 h to deplete accessible cholesterol before imaging. An inset shows at higher magnification the region indicated by a white dashed box. Note the absence of PM recruitment (compare to (D)). Scale bars, 10 μm.

G   Time course of normalized EGFP signal, as assessed by TIRF microscopy, from GRAMD1 TKO (TKO) HeLa cells expressing either wild-type EGFP-GRAM$_{1b}$ (WT) or mutant EGFP-GRAM$_{1b}$ (G187L) as indicated. Cells were cultured in the medium supplemented with 10% lipoprotein-deficient serum (LPDS) and mevastatin (50 μM) for 16 h before imaging. SMase treatment (100 mU/ml) is indicated. Right: Values of $\Delta F/F_0$ corresponding to the end of the experiment as indicated by the arrow [mean ± SEM, n = 36 cells (WT), n = 37 cells (G187L), data are pooled from two independent experiments for each condition; two-tailed unpaired Student's t-test, **P = 0.0003]. Note the rapid PM recruitment of G187L mutant compared to WT. See also Movie EV3.

H, I   Liposome sedimentation assays of wild-type GRAM$_{1b}$ (WT) and mutant GRAM$_{1b}$ (G187L). Liposomes containing the indicated mole% lipids were incubated with purified GRAM$_{1b}$ proteins as shown. Bound proteins [pellet, (P)] were separated from the unbound proteins [supernatant, (S)], run on SDS–PAGE and visualized by colloidal blue staining [mean ± SEM, n = 3 independent experiments for all the conditions; (H) Left (10% PS): Holm–Sidak t-test for multiple comparisons, *P = 0.022679 at 40% cholesterol, **P = 0.009195 at 50% cholesterol. Right (20% PS): Holm–Sidak t-test for multiple comparisons, *P = 0.023163 at 30% cholesterol, **P = 0.000327 at 40% cholesterol, *P = 0.017194 at 50% cholesterol. (I) Holm–Sidak t-test for multiple comparisons, n.s. denotes not significant, **P < 0.000001 at 50% DOPC, 50% cholesterol, *P = 0.011271 at 40% DOPC, 60% cholesterol]. DOPC, phosphatidylcholine (1,2-dioleoyl-sn-glycero-3-phosphocholine); DOPS, phosphatidylserine (1,2-dioleoyl-sn-glycero-3-phospho-L-serine); Chol., cholesterol.

Source data are available online for this figure.

A188) (Fig 6A) were individually mutated to either small hydrophobic amino acids or serine. GRAMD1 TKO HeLa cells expressing these EGFP-GRAM$_{1b}$ mutants (R166L, Q173I, F182I, T184V, F186L, G187L, or A188S) were imaged using SDC microscopy to examine whether the mutant versions were more strongly recruited to the PM than WT EGFP-GRAM$_{1b}$ (Figs 6B and EV4A). Strikingly, EGFP-GRAM$_{1b}$–G187L exhibited enhanced PM recruitment compared to EGFP-GRAM$_{1b}$ in GRAMD1 TKO cells (Fig 6B–D). Notably, EGFP-GRAM$_{1b}$–G187L was recruited to the PM even in control HeLa cells, despite limited PM cholesterol accessibility at rest. PM recruitment of EGFP-GRAM$_{1b}$–G187L was further enhanced in both control and GRAMD1 TKO HeLa cells after sphingomyelinase treatment (Fig 6D and E). These results suggest that the newly identified GRAM$_{1b}$ mutation, G187L, enhances the ability of GRAM$_{1b}$ to sense accessible PM cholesterol.

The enhanced PM recruitment of EGFP-GRAM$_{1b}$–G187L was further analyzed using TIRF microscopy during sphingomyelinase treatment. GRAMD1 TKO cells expressing either EGFP-GRAM$_{1b}$ or EGFP-GRAM$_{1b}$–G187L were analyzed for comparison. Consistent with the results from SDC microscopy, more EGFP-GRAM$_{1b}$–G187L was bound to the PM at rest compared to EGFP-GRAM$_{1b}$ (Fig EV4B; compare with Fig 6D and E). Sphingomyelinase treatment induced even stronger recruitment of EGFP-GRAM$_{1b}$–G187L to the PM over the entire 1-h imaging session compared to EGFP-GRAM$_{1b}$ (Fig EV4B). Importantly, the additional treatment of GRAMD1 TKO cells with methyl-β-cyclodextrin, which extracts cholesterol from cellular membranes, resulted in acute loss of EGFP-GRAM$_{1b}$–G187L from the PM within 2 min (Fig EV4B), demonstrating that recruitment of EGFP-GRAM$_{1b}$–G187L to the PM is cholesterol-dependent. Given the enhanced PM recruitment

of EGFP-GRAM$_{1b}$–G187L, we sought to find out whether G187L mutation could rescue impaired recruitment of EGFP-GRAM$_{1b}$–R191A (an anionic lipid-sensing defective mutant) (Fig 3F–H) to the PM. To this end, GRAMD1 TKO cells expressing either EGFP-GRAM$_{1b}$, EGFP-GRAM$_{1b}$–R191A, or EGFP-GRAM$_{1b}$–G187L/R191A were analyzed for comparison via SDC and TIRF microscopy. Remarkably, sphingomyelinase treatment induced robust recruitment of EGFP-GRAM$_{1b}$–G187L/R191A to the PM at similar levels to EGFP-GRAM$_{1b}$, thereby restoring impaired PM binding of EGFP-GRAM$_{1b}$–R191A (Appendix Fig S4A–C).

To compare the kinetics by which EGFP-GRAM$_{1b}$–G187L and EGFP-GRAM$_{1b}$ are recruited to the PM during a transient expansion of the accessible pool of PM cholesterol, accessible cholesterol was first depleted from GRAMD1 TKO cells that expressed one of these two proteins using a combination of LPDS and mevastatin for 16 h. This resulted in an almost complete loss of EGFP-GRAM$_{1b}$–G187L from the PM, confirming the specificity of GRAM$_{1b}$–G187L to accessible cholesterol (Figs 6F and EV4C). Cells were then treated with sphingomyelinase to release accessible PM cholesterol from the sphingomyelin-sequestered pool of PM cholesterol. Sphingomyelinase treatment after pre-depletion resulted in a more rapid PM recruitment of EGFP-GRAM$_{1b}$–G187L compared to EGFP-GRAM$_{1b}$, further indicating that the G187L mutation increases the sensitivity of EGFP-GRAM$_{1b}$ to accessible PM cholesterol (Fig 6G, Movie EV3).

Finally, we compared the membrane binding properties of GRAM$_{1b}$–G187L and GRAM$_{1b}$ *in vitro* by incubating purified GRAM domains with liposomes containing fixed amount of PS (10 or 20%) and increasing amounts of cholesterol (0–50%), as in Figs 2D and 3D. Strikingly, the binding curve of GRAM$_{1b}$–G187L was shifted

toward lower cholesterol concentrations compared to that of GRAM$_{1b}$ in both conditions (Fig 6H). We also incubated purified GRAM domains with liposomes containing fixed amount of PS (10%) and sterol (60%) with different ratio of cholesterol and epicholesterol [that does not bind to GRAM$_{1b}$ (Fig 3B)]. GRAM$_{1b}$ bound strongly to liposomes that contained 40–60% of cholesterol (or 20–0% epicholesterol), and the binding curve of GRAM$_{1b}$–G187L was shifted again toward lower cholesterol concentrations (Fig EV4D). These results are consistent with an increased sensitivity of GRAM$_{1b}$–G187L to membranes that contain both PS and cholesterol.

### The G187L mutation increases sensitivity of the GRAM domain to cholesterol without altering its affinity for PS

Our results suggest that the G187L mutation enhances the ability of GRAM$_{1b}$ to recognize cholesterol, PS, or both. To disentangle these possibilities, liposomes containing increasing amount of only PS (0–80%) or only cholesterol (0–60%) were generated, and the binding efficiencies of purified GRAM$_{1b}$–G187L or GRAM$_{1b}$ to these liposomes were compared. Both proteins bound to liposomes containing only PS at similar levels (Fig 6I). However, GRAM$_{1b}$–G187L bound to liposomes containing only cholesterol more strongly than GRAM$_{1b}$ (Fig 6I). For liposomes that contained 60% cholesterol, ~30% of GRAM$_{1b}$–G187L bound to liposomes, whereas only ~15% of GRAM$_{1b}$ were bound (Fig 6I). We also generated liposomes containing only sterol (60%) with different ratio of cholesterol and epicholesterol. For liposomes that contained 60% of cholesterol (or 0% epicholesterol), ~35% of GRAM$_{1b}$–G187L bound to liposomes, whereas ~15% of GRAM$_{1b}$ were bound (Fig EV4E). Thus, the G187L mutation made the GRAM domain more sensitive to cholesterol without increasing its sensitivity to PS.

In addition to sphingomyelin, phospholipid acyl chain saturation has profound effects on the accessibility of cholesterol in membranes (Radhakrishnan & McConnell, 2000; Sokolov & Radhakrishnan, 2010; Lange *et al*, 2013; Gay *et al*, 2015; Chakrabarti *et al*, 2017). If GRAM$_{1b}$–G187L retains its ability to sense the accessibility of cholesterol, its binding to artificial membranes should be influenced by the acyl chain diversity of the phospholipids, as is the case for GRAM$_{1b}$ (Naito *et al*, 2019). To test this possibility, we generated liposomes containing fixed amounts of PS (10%) and tested two more types of PC that possess different acyl chain structures, namely 1,2-diphytanoyl-sn-glycero-3-phosphocholine (DPhyPC) and 1-palmitoyl-2-oleoyl-glycero-3-phosphocholine (POPC), in addition to DOPC (Fig EV4F). Branched (DPhyPC) and more unsaturated (DOPC) acyl chains lower the tendency to form ordered conformations in the membrane, and thus, POPC has the strongest cholesterol sequestration effect, followed by DOPC and DPhyPC (Sokolov & Radhakrishnan, 2010). The binding of both GRAM$_{1b}$–G187L and GRAM$_{1b}$ was significantly increased as the ordering tendency of PC was lowered (i.e., as the cholesterol sequestration effect was reduced) (Figs 6H and EV4F and G). GRAM$_{1b}$ bound 0, 0, and ~60% to POPC-containing, DOPC-containing, and DPhyPC-containing liposomes, respectively, when they contained 30% cholesterol. For GRAM$_{1b}$–G187L, results for these same liposomes were 0%, ~20%, and ~80% binding. These results demonstrate that GRAM$_{1b}$–G187L retains the ability to sense accessible cholesterol in membranes, and that it does so more efficiently than GRAM$_{1b}$.

### G187 is critical for determining the sensitivity of the GRAM domain to accessible PM cholesterol and regulating GRAMD1b-dependent cholesterol transport

To further understand the role of G187 in GRAM domain function, we mutated this residue to every amino acid and expressed these mutant EGFP-GRAM$_{1b}$ proteins in GRAMD1 TKO cells. We then used SDC microscopy to examine their recruitment to the PM with or without sphingomyelinase treatment (Figs 7A and EV5A). PM recruitment was mostly enhanced when G187 was replaced by more hydrophobic residues. Replacing glycine with phenylalanine, methionine, isoleucine, leucine, valine, cysteine, or tryptophan all enhanced PM recruitment, whereas replacement with alanine or threonine had no effect. Importantly, treatment with methyl-β-cyclodextrin resulted in acute loss of PM binding in all versions, consistent with their specific interaction with PM cholesterol (Figs 7A and EV5A). Interestingly, replacing glycine with less hydrophobic residues (serine, proline, or asparagine) reduced PM recruitment, whereas replacement with acidic amino acids (glutamic acid and aspartic acid) completely eliminated PM recruitment. These results support the critical importance of the G187 position in modulating the sensitivity of GRAM$_{1b}$ to accessible PM cholesterol. Further, a hydrophobic amino acid is preferred in this position for the effective recognition of accessible cholesterol.

Finally, we examined the effect of the G187L mutation on GRAMD1b-dependent cholesterol transfer using a FRET-based *in vitro* lipid transfer assay. Near full-length GRAMD1b proteins carrying the G187L mutation (GRAMD1b–G187L) were purified (Fig EV5B) and mixed with PM-like and ER-like liposomes, as in Fig 4A. Addition of GRAMD1b–G187L resulted in much stronger tethering, as judged by optical turbidity, compared to the addition of near full-length GRAMD1b–WT (Fig 7B). Both GRAMD1b-WT and GRAMD1b-G187L bound equally robustly to ER-like liposomes (Fig EV5E). Thus, enhanced tethering observed with GRAMD1b-G187L was due to stronger binding of GRAMD1b-G187L to PM-like liposomes compared to GRAMD1b-WT. Consistent with the enhanced tethering, addition of GRAMD1b–G187L resulted in more efficient DHE transfer from PM-like to ER-like liposomes over time in a concentration-dependent manner (~3.6 DHE molecules per min by GRAMD1b–G187L versus ~1.2 DHE molecules per min by GRAMD1b versus ~0 DHE molecules per min by GRAMD1b–R189W/R191A) (Figs 7C and EV5C and D). Thus, the G187L mutation resulted in both increased membrane tethering and enhanced cholesterol transfer by GRAMD1b.

The effect of the G187L mutation on GRAMD1b-mediated membrane tethering was further examined in cells. We generated GRAMD1 TKO cells that stably expressed EGFP-GRAMD1b–G187L via transduction of lentiviruses. Immunoblot analysis of total cell lysates confirmed that EGFP-GRAMD1b–G187L was expressed at levels similar to EGFP-GRAMD1b in another stable cell line (Fig EV5F; compare with Fig 5A). Cholesterol-dependent recruitment of stably expressed EGFP-GRAMD1b proteins to ER-PM contacts, which reflects tethering of ER membranes to the PM, was analyzed by TIRF microscopy upon cholesterol loading (i.e., treatment with the cholesterol/methyl-β-cyclodextrin complex). EGFP-GRAMD1b–G187L was more strongly recruited to ER-PM contacts compared to EGFP-GRAMD1b–WT following cholesterol loading, consistent with the critical role of G187 in determining sensitivity of

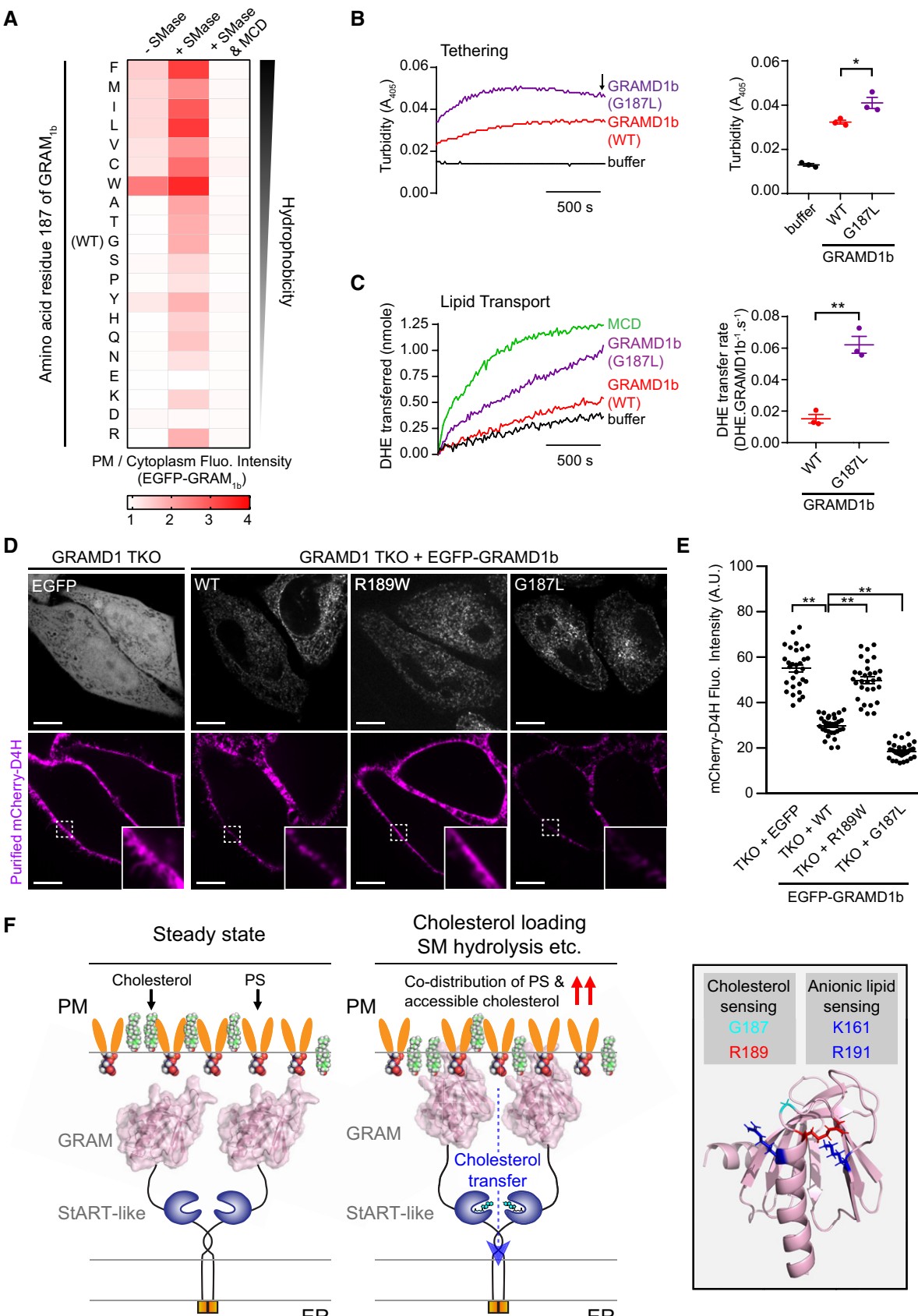

Figure 7.

Figure 7. G187 is critical for determining the sensitivity of the GRAM domain to accessible PM cholesterol and regulating GRAMD1b-dependent cholesterol transport.

A  Effects of mutation of G187 on the property of the GRAM domain of GRAMD1b (GRAM$_{1b}$) to sense transient expansion of the accessible pool of PM cholesterol. Indicated EGFP-tagged mutant versions of GRAM$_{1b}$ were expressed in GRAMD1 TKO HeLa cells and imaged under confocal microscopy with or without SMase treatment (100 mU/ml for 1 h at 37°C) or with SMase treatment followed by 5 min treatment with Methyl-β-cyclodextrin (MCD) (5 mM) to assess their PM recruitment. The mean values ($n = 20$ cells from two independent experiments for untreated and SMase-treated conditions; $n = 10$ cells from one experiment for SMase & MCD-treated conditions) of the ratio of PM signals to the cytosolic signals, as assessed by line scan analysis, are presented for each condition as a heatmap. Amino acids are ranked according to Goldman, Engelman and Steitz (GES) hydrophobicity scale with the most hydrophobic amino acid on the top. Individual values for each condition are shown in Fig EV5A.

B  Liposome tethering by purified GRAMD1b proteins. Left: Representative time course of liposome tethering, as assessed by turbidity of mixtures containing both L$_{PM}$ and L$_{ER}$ together with either wild-type GRAMD1b proteins (WT), mutant GRAMD1b G187L proteins (G187L), or buffer alone as indicated. Experiments and data analysis were performed as in Fig 4. Right: Values of turbidity corresponding to the end of the experiment as indicated by the arrow (mean ± SEM, $n = 3$ independent experiments; Dunnett's multiple comparisons test, *$P = 0.0143$).

C  DHE transfer between liposomes by purified GRAMD1b proteins. Left: Representative time course of DHE transfer from L$_{PM}$ to L$_{ER}$ mediated by either wild-type GRAMD1b proteins (WT), mutant GRAMD1b G187L proteins (G187L), or buffer alone as indicated. Methyl-β-cyclodextrin (MCD) at 1 mM was used to determine DHE equilibration. Experiments and data analysis were performed as in Fig 4. Right: Values of DHE transfer rate of wild-type GRAMD1b (WT) and mutant GRAMD1b (G187L), as estimated by the FRET-based lipid transfer assay (mean ± SEM, $n = 3$ independent experiments; two-tailed unpaired Student's $t$-test, **$P < 0.0014$).

D  Confocal images of live GRAMD1 TKO HeLa cells that stably expressed either EGFP control or EGFP-tagged GRAMD1b (EGFP-GRAMD1b) constructs as indicated. Cells were stained with recombinant mCherry–D4H proteins (10 mg/ml) (accessible PM cholesterol biosensor) for 15 min at room temperature before imaging. Insets show at higher magnification the regions indicated by white dashed boxes. Note that expression of EGFP-GRAMD1b–WT, but not EGFP-GRAMD1b–R189W, reduced binding of mCherry-D4H to the PM, in GRAMD1 TKO HeLa cells. Further reduction in mCherry-D4H binding to the PM was observed in GRAMD1 TKO HeLa cells that stably expressed EGFP-GRAMD1b–G187L compared to GRAMD1 TKO HeLa cells that stably expressed EGFP-GRAMD1b–WT. Scale bars, 10 μm.

E  Values of mCherry-D4H signals at the PM after background subtraction, as assessed by confocal microscopy and line scan analysis of GRAMD1 TKO HeLa cells that stably expressed either EGFP or indicated EGFP-GRAMD1b constructs as shown in (D) [mean ± SEM, $n = 30$ cells for all the conditions; data are pooled from three independent experiments; Dunnett's multiple comparisons test, **$P < 0.0001$].

F  A model of the molecular basis of accessible PM cholesterol recognition by the GRAM domain of GRAMD1b. At steady state, limited amount of accessible cholesterol (i.e. chemically active cholesterol) in the inner leaflet of the PM is not sufficient for the recruitment of the GRAM domain to the PM. Upon additional cholesterol loading to the PM or liberation of cholesterol from sphingomyelin (SM)-sequestered pool of inaccessible PM cholesterol (i.e. chemically inactive cholesterol), accessible PM cholesterol transiently increases in the inner leaflet of the PM beyond a certain threshold (as unsequestered cholesterol can spontaneously flip-flop between the outer and inner leaflets), resulting in increased codistribution of accessible cholesterol and anionic lipids, including phosphatidylserine (PS), in this leaflet, which serves as a platform for PM recruitment of the GRAM domain. The GRAM domain then binds to both accessible cholesterol and anionic lipids, using distinct but synergistic binding sites for cholesterol and anionic lipids, to tether the PM to the ER, and facilitates the StART-like domain-dependent extraction and transport of accessible PM cholesterol to the ER. Accessible cholesterol transport to the ER results in suppression of SREBP-2 cleavage, preventing overaccumulation of cholesterol in cells. Our study identified amino acid residues that are important for cholesterol sensing (G187, R189) and anionic lipid sensing (K161, R191) of the GRAM domain. G187 is critical for determining the sensitivity of the GRAM domain to accessible PM cholesterol, while K161 and R191 contribute to a basic patch of the GRAM domain that is required for sensing anionic lipids, including PS. An intellectual disability-associated R189W mutation, which specifically results in cholesterol-sensing defect, impairs the GRAM domain's ability to sense transient expansions of the accessible pool of PM cholesterol and abolishes GRAMD1b-mediated PM to ER cholesterol transport.

the GRAM domain to accessible PM cholesterol and in promoting tethering of the ER to the PM (Fig EV5G, Movie EV4).

We further examined the physiological impact of the G187L mutation on GRAMD1b-mediated accessible cholesterol transport in cells. To this end, we first monitored the time course of the suppression of SREBP-2 cleavage upon sphingomyelinase treatment in cells that had been pre-treated with a combination of LPDS and mevastatin for 16 h. Cell lysates were collected before and 30, 90 or 180 min after sphingomyelinase treatment and analyzed by SDS–PAGE followed by immunoblotting against SREBP-2 (Appendix Fig S5A). GRAMD1 TKO cells that stably expressed either EGFP-GRAMD1b–WT or EGFP-GRAMD1b–G187L were analyzed for comparison. Robust suppression of SREBP-2 cleavage over time was observed in both conditions (Appendix Fig S5A). Suppression of SREBP-2 cleavage could be reliably detected by immunoblotting only after ~90 min, thus making it difficult to effectively assess enhanced accessible cholesterol transport of EGFP-GRAMD1b–G187L (compared to EGFP-GRAMD1b–WT) that may occur on a shorter time scale, by this approach. Thus, we focused our subsequent analysis on the impact of the G187L mutation on PM cholesterol homeostasis.

To investigate whether the G187L mutation of GRAMD1b facilitated extraction of accessible cholesterol from the PM, we transiently expressed an accessible PM cholesterol biosensor, mCherry-GRAM$_{1b}$, in GRAMD1 TKO cells that stably expressed either EGFP-GRAMD1b–WT or EGFP-GRAMD1b–G187L and visualized accessible

cholesterol accumulation in the PM following 1 h of sphingomyelinase treatment by TIRF microscopy over time (Appendix Fig S5B). mCherry-GRAM$_{1b}$ was less recruited to the PM in GRAMD1 TKO cells that stably expressed EGFP-GRAMD1b–G187L compared to GRAMD1 TKO cells that stably expressed EGFP-GRAMD1b–WT (Appendix Fig S5B), indicating that accessible PM cholesterol was more efficiently extracted by EGFP-GRAMD1b–G187L compared to EGFP-GRAMD1b–WT. Next, we assessed the sensitivity of these cells against Amphotericin B. GRAMD1 TKO cells pre-treated with sphingomyelinase for 3 h were more efficiently killed by Amphotericin B compared to WT cells subjected to the same treatment, and such phenotype was rescued by expressing EGFP-GRAMD1b, but not by expressing EGFP (see above, Fig 5F). Remarkably, the sensitivity of GRAMD1 TKO cells to Amphotericin B was much more reduced by expressing EGFP-GRAMD1b–G187L compared to EGFP-GRAMD1b–WT, making these cells highly resistant to this drug (Fig EV5H). These results further support the notion that the G187L mutation enhances GRAMD1b-mediated accessible PM cholesterol extraction.

Finally, we directly measured the size of the accessible pool of PM cholesterol by incubating GRAMD1 TKO cells with purified mCherry-tagged D4H (mCherry-D4H) proteins (accessible cholesterol biosensor) at steady state and assessing these cells via SDC microscopy. GRAMD1 TKO cells that stably expressed either EGFP, EGFP-GRAMD1b–WT, EGFP-GRAMD1b–R189W, or EGFP-GRAMD1b–G187L were analyzed for comparison. mCherry-D4H

strongly bound to the PM of control GRAMD1 TKO cells that stably expressed EGFP, consistent with accumulation of accessible cholesterol in the PM in these cells (Naito *et al*, 2019) (Fig 7D and E). mCherry-D4H bound significantly less to the PM by expressing EGFP-GRAMD1b–WT, consistent with the direct role of GRAMD1b in extracting and transporting accessible cholesterol from the PM to the ER (Naito *et al*, 2019; Ferrari *et al*, 2020) (Fig 7D and E). In contrast, expression of EGFP-GRAMD1b–R189W failed to reduce the binding of mCherry-D4H to the PM (Fig 7D and E). Remarkably, the binding of mCherry-D4H to the PM was further reduced by expressing EGFP-GRAMD1b–G187L compared to EGFP-GRAMD1b–WT (Fig 7D and E). These results reveal the critical role of the proper sensitivity of the GRAM domain to accessible PM cholesterol in the regulation of GRAMD1-dependent cholesterol extraction and transport to the ER, thereby maintaining the size of the accessible pool of cholesterol in the PM.

Collectively, our results demonstrate an essential role for the cholesterol-sensing property of the GRAM domain in facilitating membrane tethering and in determining the rate of GRAMD1-mediated cholesterol transfer. Together with the analysis of mutants with reduced or enhanced abilities to sense cholesterol, our results highlight the critical importance of GRAM domain-mediated coincidence detection of both accessible cholesterol and anionic lipids, including PS, in fine-tuning the transport of accessible cholesterol from the PM to the ER and controlling the size of the accessible pool of cholesterol in the PM (Fig 7F).

# Discussion

We have identified the molecular basis by which the GRAM domain of GRAMD1b (GRAM$_{1b}$) detects accessible cholesterol in the presence of anionic lipids in the PM. We have also demonstrated that GRAM$_{1b}$ contributes to the regulation of GRAMD1-mediated PM to ER cholesterol transport at ER-PM contact sites. Key findings of the current study are the following:

(1) We found that a fraction of accessible cholesterol codistributes with PS in artificial membranes. Importantly, this codistribution is less pronounced in the presence of sphingomyelin, which forms a complex with cholesterol, thereby reducing its accessibility. Efficient PM recruitment of GRAM$_{1b}$ requires the presence of both PS and accessible cholesterol, strongly indicating that the GRAM domain detects the codistribution of anionic lipids, including PS, and accessible cholesterol in the inner leaflet of the PM when it becomes prominent.
(2) Using a mutagenesis approach, we found that GRAM$_{1b}$ possesses distinct, but closely apposed sites that synergistically sense anionic lipids, including PS, and accessible cholesterol. Recent human genetic studies identified a missense mutation in GRAM$_{1b}$ that is linked to intellectual disability. Strikingly, GRAM domains carrying this intellectual disability-associated missense mutation (R189W) are not able to sense cholesterol, but retain affinity for PS. Further, GRAM$_{1b}$–R189W interacts much less with the PM, even after expansion of the accessible pool of cholesterol in this bilayer.
(3) Our results demonstrate that the GRAM domain is essential for GRAMD1-dependent cholesterol transport at ER-PM contact sites. Mutant GRAMD1b proteins, carrying a GRAM domain that

is defective in cholesterol sensing, failed to tether membranes and transported cholesterol much less efficiently *in vitro* compared to wild-type GRAMD1b proteins. Accordingly, mutant GRAMD1b proteins failed to rescue the accumulation of accessible PM cholesterol, as well as the dysregulation of SREBP2 activity, in cells lacking GRAMD1s.
(4) Through a mini-mutagenesis screen, we found that a single amino acid substitution (G187L) made GRAM$_{1b}$ hypersensitive to cholesterol without altering its sensitivity to PS. While wild-type GRAM domains are recruited to the PM only after transient expansion of the accessible pool of PM cholesterol (induced by sphingomyelin hydrolysis), GRAM$_{1b}$–G187L was recruited to the PM even at steady state. Importantly, the G187L mutant retained its selectivity for accessible cholesterol.

We previously showed that the GRAM domain of GRAMD1s act as a coincidence detector for accessible cholesterol and anionic lipids, including PS, which is a major acidic lipid enriched in the inner leaflet of the PM. However, the molecular basis for the synergistic binding of GRAM domains to membranes that contain both cholesterol and anionic lipids remained unclear. Our FRET-based analysis of artificial membranes containing PC, PS, and cholesterol revealed the presence of cholesterol/PS codistribution, where cholesterol remained accessible. This suggests that some fraction of cholesterol can remain closely associated with PS (and possibly other anionic lipids) without being sequestered by this lipid. Addition of sphingomyelin into the membranes downregulated the codistribution of accessible cholesterol and PS, consistent with the ability of sphingomyelin to effectively sequester cholesterol (Endapally *et al*, 2019). In cells, masking/trapping accessible PM cholesterol via purified accessible cholesterol-binding proteins (D4H) or masking the head group of PS within the inner leaflet of the PM via overexpressed PS-selective C2 domain (Lact-C2) led to reduced PM recruitment of GRAM$_{1b}$ even when the accessible pool of PM cholesterol was expanded by hydrolysis of sphingomyelin. This indicates that 1) codistribution of cholesterol and anionic lipids, including PS, within the inner leaflet of the PM becomes prominent when levels of accessible PM cholesterol exceed a certain threshold, or when levels exceed cholesterol sequestering capacity (e.g., by sphingomyelin hydrolysis, cholesterol loading) and that 2) both accessible cholesterol and anionic lipids are required for the binding of the GRAM domain to the PM as coligands (Fig 7F).

Our results are consistent with the presence of two distinct ligand binding sites within GRAM$_{1b}$, one for binding anionic lipids and the other for binding cholesterol. Our mutagenesis study suggested that GRAM$_{1b}$ binds the acidic head group of anionic lipids, including PS, via its positively charged surface present at the β5-β6 and β7-α2 loops (i.e., the basic patch) and binds cholesterol through other amino acid(s) that are close to the basic patch (Fig 7F). Converting key amino acid residues within the GRAM$_{1b}$ basic patch (K161, R191) to alanine drastically reduced the ability of the GRAM domain to sense anionic lipids, without altering its affinity for cholesterol. In contrast, binding of the GRAM domain to cholesterol-containing membranes depends on R189 near the basic patch and the presence and stereospecificity of the hydroxyl head group and tetracyclic ring structure of cholesterol. Converting a single amino acid within GRAM$_{1b}$ (G187) to another amino acid either increased or decreased its sensitivity for accessible PM cholesterol, depending on its

hydrophobicity (Fig 7A). We speculate that one of the amino acids important for GRAM domains to recognize cholesterol might be G187. It is currently unclear how R189 plays a role in cholesterol recognition. Identification of the exact cholesterol-binding sites has been challenging for many cholesterol-sensing proteins, including the cholesterol-binding domains of cholesterol-dependent cytolysins from pathogenic bacteria, such as perfringolysin O (PFO) and anthrolysin O (ALO). Thus, elucidating exactly how GRAMD1 GRAM domains recognize cholesterol will provide great insights into general cholesterol-sensing mechanisms in the future.

The simultaneous binding of two lipids to an individual domain occurs synergistically when both lipids are present in the target membrane (Lemmon, 2008). Some domains, including PH domains, contain synergistic binding sites that each exhibit low affinity for the two lipids. Such domains bind with highest affinity to membranes that contain both the lipids they recognize (two lipid coincidence detection). For example, the PH domain of Arf GAP ASAP1 possesses two synergistic lipid-binding sites and binds most efficiently to $PI(4,5)P_2$-containing membranes in the presence of other anionic lipids, such as PS (Jian *et al*, 2015). The PX domain of p47$^{phox}$ also possesses synergistic lipid-binding sites, one for $PI(3,4)P_2$ and the other for anionic lipids such as phosphatidic acid and PS (Karathanassis *et al*, 2002). Similarly, the C2 domain of PKC$\alpha$ binds synergistically to $PI(4,5)P_2$ and PS (Guerrero-Valero *et al*, 2009). Coincidence detection of accessible cholesterol and anionic lipids via distinct ligand binding sites would ensure robust recruitment of the GRAM domain to the PM, whose inner leaflet is enriched with PS and other anionic lipids, only when levels of accessible PM cholesterol exceed a certain threshold. The switch-like property of the GRAM domain in binding the PM only when necessary prevents GRAMD1s from extracting too much cholesterol. Notably, a recent study by Trinh *et al* showed that PM PS is essential for the PM to ER transport of LDL-derived cholesterol (Trinh *et al*, 2020). This is in good agreement with our data, which revealed a critical role of anionic lipids, including PS, as coligands for PM recruitment of GRAMD1b for the PM to ER cholesterol transport.

Using a cell-free reconstitution approach, we showed that the ability of GRAM$_{1b}$ to bind accessible cholesterol fine-tuned both membrane tethering and cholesterol transport functions of GRAMD1b. Accordingly, GRAMD1b mutants that could not sense cholesterol (R189W or R189W/R191A) failed to tether membranes and transported cholesterol less efficiently. In contrast, GRAMD1b proteins that were hypersensitive to cholesterol (G187L) exhibited increased membrane tethering and cholesterol transport. Non-vesicular lipid transport between cellular membranes is most efficient when donor and target membranes are closely apposed (Lev, 2012; Wong *et al*, 2019). Our results confirm this notion and further demonstrate that the cholesterol-dependent binding of GRAMD1 GRAM domains to the PM *in trans* promotes tethering of ER membranes [where GRAMD1s, which form homo- and hetero-meric complexes (Naito *et al*, 2019), are located] to the PM and facilitates the non-vesicular cholesterol transport between these membranes by the StART-like domain (Fig 7F). A potential role of the GRAM domain to actively promote the codistribution of accessible cholesterol and anionic lipids within the PM to enhance StART-like domain-dependent extraction of cholesterol from this bilayer cannot be excluded. Notably, near full-length GRAMD1b proteins carrying anionic lipid-sensing defective GRAM domain mutations (i.e., basic patch mutations: R191A or K161A/R191A) tethered membranes and transported cholesterol as efficiently as wild-type near full-length GRAMD1b proteins *in vitro*, despite critical importance of the basic patch for the isolated GRAM$_{1b}$ to sense anionic lipids *in vitro* and to interact with the PM *in vivo*. Thus, there may be other amino acids outside of the GRAM$_{1b}$ that contribute to anionic lipid-sensing property of full-length GRAMD1b proteins.

Importantly, the R189W mutation is associated with intellectual disability in humans. We found that GRAM domains with the R189W mutation specifically impaired its binding to cholesterol without affecting its binding to PS and failed to bind to the PM even when levels of accessible cholesterol were elevated. Accordingly, a version of GRAMD1b with this mutation was unable to rescue cholesterol transport defects in cells lacking GRAMD1s. These results demonstrate that the R189W mutation in the GRAM domain results in severe GRAMD1b loss of function in humans. Mounting evidence suggests a link between GRAMD1b and various neurodevelopmental disorders, including schizophrenia and intellectual disability (Schizophrenia Working Group of the Psychiatric Genomics C, 2014; Reuter *et al*, 2017; Santos-Cortez *et al*, 2018; Thyme *et al*, 2019). Interestingly, knocking out GRAMD1b in zebrafish results in reduced brain activity (Thyme *et al*, 2019). In mice, GRAMD1b plays an important role in steroidogenesis within the adrenal glands (Sandhu *et al*, 2018). However, the function of GRAMD1b in mammalian brain currently remains unknown. Future studies are needed to elucidate the role of GRAMD1b and other GRAMD1s in mammalian brain development and neuronal function.

Finally, using various *in vitro* and *in vivo* assays, we showed that G187L mutation results in GRAM domains that are hypersensitive to accessible cholesterol, while maintaining affinity for PS. Other widely used toxin-based biosensors for accessible cholesterol, including D4 of ALO and PFO, are useful for measuring levels of accessible cholesterol in the PM, but they trap accessible cholesterol in the PM and induce dysregulation of cholesterol homeostasis (Infante & Radhakrishnan, 2017; Johnson *et al*, 2019). Thus, they are not suitable for long-term, live cell imaging of mammalian cells. We envision that the GRAM domain with the G187L mutation, which we name "GRAM-H", may serve as a powerful tool for characterizing the distribution of accessible cholesterol in the inner leaflet of the PM.

In summary, we have shown that cells monitor the codistribution of accessible cholesterol and anionic lipids within the inner leaflet of the PM and fine-tune the rate at which accessible PM cholesterol is extracted and transported to the ER via GRAMD1s. The GRAMD1 GRAM domain is the first example of an intracellular membrane-binding domain in eukaryotes that is specifically tuned to sense levels of accessible PM cholesterol. Because GRAMD1s are conserved from yeast to humans, our findings provide important insights into conserved mechanisms by which levels of intracellular sterols are sensed and monitored in all eukaryotes.

# Materials and Methods

### Antibodies and chemicals

Primary and secondary antibodies, chemicals, lipids, and other reagents used in this study are listed in Table EV1.

## DNA plasmids

DNA plasmids, the sequences of oligos, and primers used are listed in Table EV1.

### For recombinant protein purification

#### Cloning of His-GRAM$_{1b}$

cDNA corresponding to the GRAM domain of human GRAMD1b protein (NP_065767.1) (92-207) was cloned into the pNIC28-Bsa4 vector with an N-terminal His$_6$-tag and a TEV-protease cleavage site via the ligation-independent cloning method to generate pNIC28-Bsa4 His-GRAMD1b$_{92-207}$ (GRAM$_{1b}$).

#### Cloning of His-GRAM$_{1b}$ (K161A), His-GRAM$_{1b}$ (R189W), His-GRAM$_{1b}$ (R191A), His-GRAM$_{1b}$ (K161A/R191A), His-GRAM$_{1b}$ (R189W/R191A), and His-GRAM$_{1b}$ (G187L)

The amino acid residues (K161, R189, R191, and G187) present in the GRAM$_{1b}$ were mutated as indicated using site-directed mutagenesis in pNIC28-Bsa4 His-GRAMD1b$_{92-207}$ (GRAM$_{1b}$) with the following primer sets, (K161A: GRAMD1b_K161A_F and GRAMD1b_K161A_R; R189W: GRAMD1b_R189W_F and GRAMD1b_R189W_R; R191A: GRAMD1b_R191A_F and GRAMD1b_R191A_R; R189W/R191A: GRAMD1b_R189W R191A_F and GRAMD1b_R189W R191A_R; G187L: GRAMD1b_G187L_F and GRAMD1b_G187L_R), to generate pNIC28-Bsa4 His-GRAM$_{1b}$ (K161A), pNIC28-Bsa4 His-GRAM$_{1b}$ (R189W), pNIC28-Bsa4 His-GRAM$_{1b}$ (R191A), pNIC28-Bsa4 His-GRAM$_{1b}$ (R189W/R191A), and pNIC28-Bsa4 His-GRAM$_{1b}$ (G187L), respectively. pNIC28-Bsa4 His-GRAM$_{1b}$ (K161A) was then used as a template and mutated with the primer set GRAMD1b_R191A_F and GRAMD1b_R191A_R to generate pNIC28-Bsa4 His-GRAM$_{1b}$ (K161A/R191A).

#### Cloning of GRAMD1b-His (WT), GRAMD1b-His (R189W), GRAMD1b-His (R189W/R191A), GRAMD1b-His (R191A), GRAMD1b-His (K161A/R191A) and GRAMD1b-His (G187L)

cDNA corresponding to the cytosolic region of human GRAMD1b (NP_065767.1) (82-548), including both the GRAM domain and StART-like domain, was PCR-amplified using EGFP-GRAMD1b (Naito et al, 2019) as a template and the primer set 5'NcoI C-GRAMD1b (82-529) and 3'XhoI C-GRAMD1b (82-548). The PCR products were then ligated at NcoI and XhoI sites in pET28b(+) to generate pET28b(+) GRAMD1b$_{82-548}$-His.

The amino acid residues (K161, R189, R191, and G187) present in GRAMD1b were mutated as indicated using site-directed mutagenesis in pET28b(+) GRAMD1b$_{82-548}$-His with the following primer sets, (R189W: GRAMD1b_R189W_F and GRAMD1b_R189W_R; R189W/R191A: GRAMD1b_R189W R191A_F and GRAMD1b_R189W R191A_R; R191A: GRAMD1b_R191A_F and GRAMD1b_R191A_R; G187L: GRAMD1b_G187L_F and GRAMD1b_G187L_R), to generate pET28b(+) GRAMD1b$_{82-548}$-His (R189W), pET28b(+) GRAMD1b$_{82-548}$-His (R189W/R191A), pET28b(+) GRAMD1b$_{82-548}$-His (R191A), and pET28b(+) GRAMD1b$_{82-548}$-His (G187L), respectively. pET28b(+) GRAMD1b$_{82-548}$-His (R191A) was then used as a template and mutated with the primer set GRAMD1b_K161A_F and GRAMD1b_K161A_R to generate pET28b(+) GRAMD1b$_{82-548}$-His (K161A/R191A).

#### Cloning of ECFP-D4H and mVenus-Lact-C2

gBlocks (IDT) containing either ECFP-D4H or mVenus-Lact-C2 were synthesized and individually amplified by PCR, using the following primer sets (ECFP-D4H: 5'NcoI_ECFP-D4H and 3'BamHI_ECFP-D4H, mVenus-Lact-C2: 5'NcoI_mVenus-Lact-C2, and 3'BamHI_pNIC28_mVenus-Lact-C2). The PCR products were then ligated at NcoI and BamHI sites in pNIC28-Bsa4 to generate pNIC28-Bsa4 His-ECFP-D4H and pNIC28-Bsa4 His-mVenus-Lact-C2, respectively.

#### Cloning of His-D4H and His-Lact-C2

cDNA corresponding to the D4H of Perfringolysin O (PFO) or the C2 domain of Lactadherin was individually PCR-amplified using pNIC28-Bsa4 EGFP-D4H or gBlock (IDT) containing EGFP-Lact-C2 as a template and the following primer sets (D4H: 5'NcoI_pNIC28_D4H and 3'BamHI_pNIC28_D4H; C2: 5'NheI_C2 and 3'XhoI_EGFP-C2). The PCR products were then ligated at NcoI and BamHI sites in pNIC28-Bsa4 for D4H and at NheI and XhoI sites for Lact-C2 in pET28b(+) to generate pNIC28-Bsa4 His-D4H and pET28b(+) His-Lact-C2, respectively.

#### Cloning of mCherry-D4H

cDNA corresponding to the mCherry-D4H was PCR-amplified using mCherry-D4H vector and the primer set D4H_F and D4H_R. The PCR products were then ligated at NcoI and MfeI sites in pNIC28-Bsa4 to generate pNIC28-Bsa4 mCherry-D4H.

### For mammalian expression

#### Cloning of EGFP-GRAM$_{1b}$ (K161A), EGFP-GRAM$_{1b}$ (R189W), EGFP-GRAM$_{1b}$ (R191A), EGFP-GRAM$_{1b}$ (R189W/R191A), EGFP-GRAM$_{1b}$ (K161A/R191A), EGFP-GRAM$_{1b}$ (R166L), EGFP-GRAM$_{1b}$ (Q173I), EGFP-GRAM$_{1b}$ (F182I), EGFP-GRAM$_{1b}$ (T184V), EGFP-GRAM$_{1b}$ (F186L), and EGFP-GRAM$_{1b}$ (A188S)

The amino acid residues (K161, R189, R191, G187, R166, Q173, F182, T184, F186, and A188) present in the GRAM$_{1b}$ were mutated as indicated using site-directed mutagenesis in EGFP-GRAM$_{1b}$ (Naito et al, 2019) with the following primer sets, (K161A: GRAMD1b_K161A_F and GRAMD1b_K161A_R; R189W: GRAMD1b_R189W_F and GRAMD1b_R189W_R; R191A: GRAMD1b_R191A_F and GRAMD1b_R191A_R; R189W/R191A: GRAMD1b_R189W R191A_F and GRAMD1b_R189W R191A_R; R166L: GRAMD1b_R166L_F and GRAMD1b_R166L_R; Q173I: GRAMD1b_Q173I_F and GRAMD1b_Q173I_R; F182I: GRAMD1b_F182I_F and GRAMD1b_F182I_R; T184V: GRAMD1b_T184V_F and GRAMD1b_T184V_R; F186L: GRAMD1b_F186L_F and GRAMD1b_F186L_R; A188S: GRAMD1b_A188S_F2 and GRAMD1b_A188S_R), to generate EGFP-GRAM$_{1b}$ (K161A), EGFP-GRAM$_{1b}$ (R189W), EGFP-GRAM$_{1b}$ (R191A), EGFP-GRAM$_{1b}$ (R189W/R191A), EGFP-GRAM$_{1b}$ (R166L), EGFP-GRAM$_{1b}$ (Q173I), EGFP-GRAM$_{1b}$ (F182I), EGFP-GRAM$_{1b}$ (T184V), EGFP-GRAM$_{1b}$ (F186L), and EGFP-GRAM$_{1b}$ (A188S), respectively. EGFP-GRAM$_{1b}$ (R191A) was then used as a template and mutated with the primer set GRAMD1b_K161A_F and GRAMD1b_K161A_R to generate EGFP-GRAM$_{1b}$ (K161A/R191A).

#### Cloning of EGFP-GRAM$_{1b}$ G187 variants

The amino acid residue (G187) present in the GRAM$_{1b}$ was systematically mutated to every other amino acid using site-directed mutagenesis in EGFP-GRAM$_{1b}$ (Naito et al, 2019) with the following primer sets, (G187L: GRAMD1b_G187L_F and GRAMD1b_G187L_R; G187K: GRAMD1b_G187K_F and GRAMD1b_G187K_R; G187W: GRAMD1b_G187W_F and GRAMD1b_G187W_R; G187R: GRAMD1b_G187R_F and GRAMD1b_G187R_R; G187H: GRAMD1b_G187H_F and GRAMD1b_G187H_R; G187D: GRAMD1b_G187D_F and

GRAMD1b_G187D_R; G187E: GRAMD1b_G187E_F and GRAMD1b_G187E_R; G187S: GRAMD1b_G187S_F and GRAMD1b_G187S_R; G187T: GRAMD1b_G187T_F and GRAMD1b_G187T_R; G187N: GRAMD1b_G187N_F and GRAMD1b_G187N_R; G187Q: GRAMD1b_G187Q_F and GRAMD1b_G187Q_R; G187C: GRAMD1b_G187C_F and GRAMD1b_G187C_R; G187P: GRAMD1b_G187P_F and GRAMD1b_G187P_R; G187A: GRAMD1b_G187A_F and GRAMD1b_G187A_R; G187V: GRAMD1b_G187V_F and GRAMD1b_G187V_R; G187I: GRAMD1b_G187I_F and GRAMD1b_G187I_R; G187M: GRAMD1b_G187M_F and GRAMD1b_G187M_R; G187F: GRAMD1b_G187F_F and GRAMD1b_G187F_R; G187Y: GRAMD1b_G187Y_F and GRAMD1b_G187Y_R; G187L/R191A: GRAMD1b_G187L R191A_F and GRAMD1b_G187L R191A_R), to generate EGFP-GRAM$_{1b}$ G187 variants.

### Cloning of mCherry-GRAM$_{1b}$

cDNA corresponding to the GRAM domain of human GRAMD1b protein (NP_065767.1) (92-207) was digested from EGFP-GRAM$_{1b}$ (Naito *et al*, 2019) and ligated into pmCherry-C1 at XhoI and KpnI sites to generate mCherry-GRAM$_{1b}$.

### For lentivirus generation

The amino acid residues (R189, R191, and G187) present in GRAMD1b were mutated as indicated using site-directed mutagenesis in EGFP-GRAMD1b (Naito *et al*, 2019) with the following primer sets, (R189W: GRAMD1b_R189W_F and GRAMD1b_R189W_R; R189W/R191A: GRAMD1b_R189W R191A_F and GRAMD1b_R189W R191A_R; G187L: GRAMD1b_G187L_F and GRAMD1b_G187L_R), to generate EGFP-GRAMD1b (R189W), EGFP-GRAMD1b (R189W/R191A), and EGFP-GRAMD1b (G187L), respectively.

EGFP-GRAMD1b, EGFP-GRAMD1b (R189W), EGFP-GRAMD1b (R189W/R191A), and EGFP-GRAMD1b (G187L) were individually digested using NheI and MfeI and ligated into pLJM1-EGFP at NheI and EcoRI sites to generate pLJM1-EGFP-GRAMD1b, pLJM1-EGFP-GRAMD1b (R189W), pLJM1-EGFP-GRAMD1b (R189W/R191A), and pLJM1-EGFP-GRAMD1b (G187L), respectively.

### Generation of cell lines that stably expressed either EGFP, EGFP-GRAMD1b, or EGFP-GRAMD1b mutants

Lentiviral helper plasmids (pMD2.G, pRSV-REV, and pMDL/pRRE) (3.7 μg each) were transfected together with either one of the following plasmids [pLJM1-EGFP, pLJM1-EGFP-GRAMD1b, pLJM1-EGFP-GRAMD1b (R189W), pLJM1-EGFP-GRAMD1b (R189W/R191A), and pLJM1-EGFP-GRAMD1b (G187L)] (7.4 μg) into $4.4 \times 10^6$ HEK293T cells according to manufacturer's protocol. Supernatant was collected 48 h after transfection and filtered using a 0.45 μm filter unit to recover lentiviruses.

Wild-type control HeLa and GRAMD1 TKO HeLa cells were seeded at $1.5 \times 10^5$ cells and transduced with lentiviruses (LV-EGFP) to generate stable cell lines expressing EGFP.

GRAMD1 TKO HeLa cells (Naito *et al*, 2019) were seeded at $1.5 \times 10^5$ cells and transduced with either one of the following lentiviruses [LV-EGFP-GRAMD1b, LV-EGFP-GRAMD1b (R189W), LV-EGFP-GRAMD1b (R189W/R191A), and LV-EGFP-GRAMD1b (G187L)] to generate stable GRAMD1 TKO HeLa cell lines expressing indicated proteins.

After puromycin selection (5 μg/ml), BD FACSAria™ Fusion (BD Biosciences) was used to isolate cells that are positive for EGFP fluorescence. The sorted cells were subsequently maintained in 0.5 μg/ml puromycin, and protein expression was confirmed by microscopy and Western blotting.

### Cell culture and transfection

HeLa cells were cultured in Dulbecco's modified Eagle's medium (DMEM) containing 20% fetal bovine serum (FBS) and 1% penicillin/streptomycin at 37°C and 5% $CO_2$. Transfection of plasmids was carried out with Lipofectamine 2000 (Thermo Fisher Scientific). Wild-type and genome-edited HeLa cell lines were routinely verified as free of mycoplasma contamination at least every 2 months, using Myco-Guard Mycoplasma PCR Detection Kit (Genecopoeia). No cell lines used in this study were found in the database of commonly misidentified cell lines that is maintained by ICLAC and NCBI Biosample.

### Fluorescence microscopy

For imaging experiments, cells were plated onto 35 mm glass bottom dishes at low density (MatTek Corporation). All live cell imaging was carried out 1 day after transfection.

Spinning disk confocal (SDC) microscopy (Figs 3F and G, 5B and C, 6B and D–F, 7A and D and E, EV3B, EV4A and C, EV5A, Appendix Figs S1B and S4A and B) was performed on a setup built around a Nikon Ti2 inverted microscope equipped with a Yokogawa CSU-W1 confocal spinning head, a Plan-Apo objective (100× 1.45-NA), a back-illuminated sCMOS camera (Prime 95B; Photometrics). Excitation light was provided by 488-nm/150 mW (Coherent) (for GFP), 561-nm/100 mW (Coherent) (for mCherry) and 642-nm/110 mW (Vortran) (for iRFP) (power measured at optical fiber end) DPSS laser combiner (iLAS system; Gataca systems), and all image acquisition and processing was controlled by MetaMorph (Molecular Device) software. Images were acquired with exposure times in the 400-500 msec range.

Total internal reflection fluorescence (TIRF) microscopy (Figs 1D and E, 3H, 5D, 6G, EV3A, EV4B and EV5G, Appendix Figs S1C, S4C and S5B, and Movie EV1-EV4) was performed on a setup built around a Nikon Ti2 inverted microscope equipped with a HP Apo-TIRF objective (100X1.49-NA), and a back-illuminated sCMOS camera (Prime 95B; Photometrics). Excitation light was provided by 488-nm/70 mW (for GFP), 561-nm/70 mW (for mCherry), and 647-nm/125 mW (for iRFP) (power measured at optical fiber end) DPSS laser combiner (Nikon LU-NV laser unit), coupled to the motorized TIRF illuminator through an optical fiber cable. Critical angle was maintained at different wavelengths throughout the experiment from the motorized TIRF illuminator. Acquisition was controlled by Nikon NIS-Element software. For time-lapse imaging, images were sampled at 0.05 Hz with exposure times in the 500 ms range.

Cells were washed twice and incubated with $Ca^{2+}$ containing buffer (140 mM NaCl, 5 mM KCl, 1 mM $MgCl_2$, 10 mM HEPES, 10 mM glucose, and 2 mM $CaCl_2$, pH 7.4) before imaging with either an SDC microscope or a TIRF microscope. All types of microscopy were carried out at 37°C.

### Drug stimulation for time-lapse TIRF imaging

For all time-lapse TIRF imaging experiments with drug stimulation, drugs were added to the cells 5 min after the initiation of the imaging except for methyl-β-cyclodextrin (MCD) treatment, where 5 mM

MCD (Sigma-Aldrich/Merck) was added to the cells as indicated in Fig EV6B. Other drugs were used with the following concentration: 200 μM cholesterol/MCD complex generated as described previously (Brown *et al*, 2002); 100 mU/ml sphingomyelinase (SMase) (Sigma-Aldrich/Merck). For ECFP-D4H binding assay, cells were pre-incubated with 3 μM purified ECFP-D4H proteins for 30 min at 37°C before imaging; cells were maintained in the presence of 3 μM purified ECFP-D4H proteins throughout the imaging.

### Plasma membrane binding of the recombinant mCherry-D4H proteins

GRAMD1b TKO HeLa Cells that stably expressed either EGFP or EGFP-GRAMD1b variants (WT, G187L, and R189W) that were additionally transfected with iRFP-PH-PLCδ were washed once with imaging buffer and subsequently incubated with the same imaging buffer that contained purified mCherry-D4H proteins (10 μg/ml) for 15 min at room temperature. Cells were then washed twice with the same buffer without mCherry-D4H proteins and immediately imaged under SDC microscopy at room temperature.

### Image analysis

All images were analyzed offline using ImageJ (http://fiji.sc/wiki/index.php/Fiji). Quantification of fluorescence signals was performed using Excel (Microsoft) and Prism 7 or 8 (GraphPad Software). All data are presented as mean ± SEM. In dot plots, each dot represents value from a single cell with the bar as the mean.

For time-lapse imaging via TIRF microscopy, changes in PM fluorescence over time were analyzed by manually selecting regions of interest covering the largest possible area of the cell foot-print. Mean fluorescence intensity values of the selected regions were obtained and normalized to the average fluorescence intensity before stimulation after background subtraction.

For analysis of the recruitment of either EGFP-GRAM$_{1b}$ constructs or mCherry-GRAM$_{1b}$ to the PM via SDC microscopy, line scan analysis was performed. A line of 5 μm in length was manually drawn around the PM, and EGFP or mCherry fluorescence intensity along the manually drawn line was measured. The peak intensity around the PM region was normalized with the intensity of cytoplasmic region and then plotted for quantification.

For the analysis of the recombinant mCherry-D4H binding to the PM via SDC microscopy, line scan analysis was performed. D4H fluorescence intensity along the manually drawn line (5 μm in length) was measured. The fluorescence intensity corresponding to the extracellular region was subtracted from the maximum fluorescence intensity along the line, corresponding to the PM signals (defined by peak iRFP-PH-PLCδ signals), and plotted for quantification.

### Biochemical analyses

#### Protein purification

All proteins were overexpressed in *E. coli* BL21-DE3 Rosetta cells. A 750 ml culture was grown until OD$_{600}$ ~0.5–0.7 with appropriate antibiotics. 0.1 mM IPTG (Thermo Fisher Scientific) was then added, and the culture was further grown at 18°C for 18 h to allow protein expression. Cells were harvested by centrifugation at 4,700 *g*, at 4°C for 15 min, and re-suspended in 30 ml of lysis buffer (100 mM HEPES, 500 mM NaCl, 10 mM imidazole, 10% glycerol, 0.5 mM TCEP, pH 7.5), supplemented with protease inhibitors (Complete, EDTA-free; Roche) together with the cocktail of 100 μg/ml lysozyme (Sigma-Aldrich/Merck) and 50 μg/ml DNAse I (Sigma-Aldrich/Merck). Cells were lysed with sonication on ice in a Vibra Cell (Sonics and Materials, Inc) (70% power, 3 s pulse on, 3 s pulse off for 3 min for three to five rounds). The lysate was clarified by centrifugation at 47,000 *g*, at 4°C for 20 min. The supernatants were incubated at 4°C for 30 min with Ni-NTA resin (Thermo Fisher Scientific), which had been equilibrated with 2.5 ml of wash buffer 1 (20 mM HEPES, 500 mM NaCl, 10 mM imidazole, 10% glycerol, 0.5 mM TCEP, pH 7.5). The protein–resin mixtures were then loaded onto a column to be allowed to drain by gravity. The column was washed with 10 ml of wash buffer 1 once and 10 ml of wash buffer 2 (20 mM HEPES, 500 mM NaCl, 25 mM imidazole, 10% glycerol, 0.5 mM TCEP, pH 7.5) once, and then eluted with 1.25 ml of elution buffer 1 (20 mM HEPES, 500 mM NaCl, 500 mM imidazole, 10% glycerol, 0.5 mM TCEP, pH 7.5). The proteins were then concentrated using Vivaspin 20 MWCO 10 kDa or MWCO 30 kDa (GE Healthcare) and further purified by gel filtration (Superdex 200 increase 10/300 GL, GE Healthcare) with elution buffer 2 (20 mM HEPES, 300 mM NaCl, 10% glycerol, 0.5 mM TCEP, pH 7.5), using the AKTA Pure system (GE Healthcare). Relevant peaks were pooled, and the protein sample was concentrated.

### Liposome-based experiments
#### Liposome preparation
Liposomes were prepared as previously described (Naito *et al*, 2019). Lipids in chloroform were dried under a stream of N$_2$ gas, followed by further drying in the vacuum for 2 h. Mole% of lipids used for the acceptor and donor liposomes in FRET-based lipid transfer assays are shown in Table EV1. The dried lipid films were hydrated with HK buffer (50 mM HEPES, 120 mM potassium acetate, pH 7.5). Liposomes were then formed by five freeze-thaw cycles (liquid N$_2$ and 37°C water bath) followed by extrusion using Nanosizer with a pore size of 100 nm (T&T Scientific Corporation). Liposomes in Figs 2E and 3B (except cholesterol-containing liposomes), Figs EV4D and E, and EV5E, Appendix Figs S1A and S3E were not subjected to extrusion. Liposomes in cholesterol specificity assay in Fig 3B were subjected to sonication (1 min, twice) after the freeze-thaw step.

#### Liposome sedimentation assays
Liposome sedimentation assays were performed as previously described (Naito *et al*, 2019). Heavy liposomes were prepared by hydrating 1.6 mM dried lipid films in HK buffer containing 0.75 M sucrose and subjected to freeze-thaw cycles five times. Next, 160 μl of heavy liposomes were pelleted and washed with HK buffer without sucrose twice to remove unencapsulated sucrose. Pelleted heavy liposomes were re-suspended in 160 μl HK buffer and incubated with indicated GRAM$_{1b}$ proteins (6-7 μg) or near full-length GRAMD1b proteins (12 μg) for 1 h at room temperature. Unbound proteins (supernatant) were separated from liposome-bound proteins (pellet) by centrifugation at 21,000 *g* for 1 h at 25°C. After centrifugation, the supernatant was removed, and pellets were re-suspended in 160 μl HK buffer. 20 μl samples were taken from both fractions and run on SDS–PAGE followed by colloidal blue staining.

The protein bands were quantified using Fiji, and the % bound GRAM$_{1b}$ was plotted for each condition after background (% bound GRAM$_{1b}$ in DOPC only liposomes) subtraction.

### FRET-based DHE transfer and turbidity assays

Buffer of the purified proteins was replaced with HK buffer prior to the FRET-based DHE transfer assay. Reactions were performed in 50 µl volumes. The final lipid concentration in the reaction was 1 mM, with PM- and ER-like liposomes added at a 1:1 ratio. Reactions were initiated by the addition of protein to a final concentration of 0.2 µM in a 96-well plate (Corning). The fluorescence intensity of DNS-PE (i.e. FRET signals), resulting from FRET between DNS-PE and DHE (excited at 310 nm), was monitored at 525 nm every 15 s over 30 min at room temperature by using a Synergy H1 microplate reader (Biotek). The values of blank solution (buffer only without liposomes or proteins) were subtracted from all the values from each time point. Data were expressed as the number of DHE molecules transferred using the calibration curve. For the generation of the calibration curve, FRET signals were measured for the L$_{PM}$ containing 0%, 2.5% (0.625 nmole), 5% (1.25 nmole), 7.5% (1.875 nmole), and 10% (2.5 nmole) DHE and 2.5% DNS-PE (0.5 mM lipids in total). The mean of FRET signals at t = 0 from three replicates were plotted against the DHE mole number in liposomes (Appendix Fig S2D). Then, the mole number of the transferred DHE from the donor to acceptor liposomes in in vitro DHE transfer assay was obtained using the following formula: $y = 8905x + 6131$ (derived from the linear fit of the calibration curve). To obtain $x$ (the amount of DHE present in L$_{PM}$ in nmole), the FRET values from each time point of the in vitro lipid transfer assay were substituted for the $y$ of the equation. $x$ was normalized to the initial DHE amount in L$_{PM}$ (2.5 nmole), and the amount of DHE transferred from L$_{PM}$ to L$_{ER}$ in nmole ($\Delta$) was plotted for the $y$ axis of Figs 4C, 7C and EV5C, Appendix Figs S2E and F, and S3D. Transfer rates of GRAMD1b (WT), GRAMD1b mutants, and buffer alone were individually calculated from the slopes of the graphs using the one-phase association function with constraints [$y = 0$ and plateau = 1.25 (plateau = 1.25 nmole DHE in the equilibrium)] of Prism 7 (GraphPad); values for GRAMD1-dependent DHE transfer rates [for GRAMD1b (WT) and GRAMD1b mutants] were shown after background (buffer alone) subtraction (Figs 4C, 7C and EV5D, Appendix Fig S3D). During the assay, turbidity of liposome–protein mixtures was monitored as an estimate of liposome tethering by measuring absorbance at 405 nm (Figs 4B and 7B, Appendix Fig S3C). The initial increase in turbidity could not be captured due to the time lag (~30 s) between the time when the mixture of liposomes and purified proteins were aliquoted into individual wells of a 96-well plate and the time when the plate reader started to read the optical turbidity of individual wells.

### FRET-based lipid codistribution detection

Liposomes (20 µM total lipids) were mixed with ECFP-D4H (200 nM) and mVenus-Lact-C2 (200 nM) in elution buffer 2 (20 mM HEPES, 300 mM NaCl, 10% glycerol, 0.5 mM TCEP, pH 7.5). After incubation for 5 min, fluorescence spectra were recorded using a Synergy H1 microplate reader (Biotek). ECFP signal (ex/em 430 nm/477 nm), mVenus signal (ex/em 505 nm/525 nm), and FRET signal (ex/em 430 nm/525 nm) were measured.

### FRET-based competition assay with untagged Lact-C2 and untagged D4H proteins

Liposomes (20 µM total lipids) that were mixed with ECFP-D4H (200 nM) and mVenus-Lact-C2 (200 nM) were additionally mixed with either Lact-C2 or D4H at indicated concentrations (0-1000 nM) in elution buffer 2. After incubation for 5 min, fluorescence spectra were recorded using a Synergy H1 microplate reader (Biotek). ECFP signal (ex/em 430 nm/477 nm), mVenus signal (ex/em 505 nm/525 nm), and FRET signal (ex/em 430 nm/525 nm) were measured.

### Western blotting and immunoprecipitation

Western blotting was performed as previously described (Naito et al, 2019). HeLa cells were lysed in SDS lysis buffer (2% SDS, 150 mM NaCl, 10 mM Tris, pH 8.0) and incubated at 60°C for 20 min followed by incubation at 70°C for 10 min. The lysates were treated with Benzonase Nuclease (SantaCruz) for 30 min at room temperature. Cell lysates were processed for SDS–PAGE and immunoblotting with standard procedure. All immunoblots were developed by chemiluminescence using the SuperSignal West Dura reagents (Thermo Fisher Scientific).

### Amphotericin B resistance assay

1 x 10$^4$ HeLa cells that stably expressed EGFP and 1.5 x 10$^4$ GRAMD1 TKO HeLa cells that stably expressed either EGFP, EGFP-GRAMD1b (WT), EGFP-GRAMD1b (R189W), EGFP-GRAMD1b (R189W/R191A), or EGFP-GRAMD1b (G187L) were plated onto 96-well plates 1 day prior to drug treatment. On the day of drug treatment, each well was first treated with 100 µl of DMEM containing 20% FBS, 1% penicillin/streptomycin, and 100 mU/ml SMase for 3 h at 37°C. Wells were washed with PBS twice and further treated with 100 µl DMEM containing 20% FBS, 1% penicillin/streptomycin, and Amphotericin B (25 µg/ml, 50 µg/ml, 100 µg/ml, 150 µg/ml, 200 µg/ml, 300 µg/ml, or 400 µg/ml) for 20 min at 37°C. Cells were washed with PBS twice and then recovered in DMEM containing 20% FBS, 1% penicillin/streptomycin for 24 h at 37°C. CellTiter-Glo® Luminescent Cell Viability Assay Kit was then used to quantify cell viability according to manufacturer's protocol.

## Molecular modeling

The modeled structure of the GRAM domain of GRAMD1b was obtained by submitting the primary sequence of the GRAM domain (92–207) to I-TASSER server and using PH-GRAM domain of Lam6/Ltc1 (PDB: 5YQR) as a template.

## Bioinformatics

For Fig EV2B, the sequences of human GRAMD1s and their selected homologs were manually retrieved from UniProtKB. Sequence alignment was performed by Clustal Omega; ESPript 3.0 (Robert & Gouet, 2014) was used to annotate the secondary structure information based on Lam6/Ltc1 (PDB: 5YQR). For Fig EV2C and D, the sequences of human GRAMD1b and its homologs were extracted from UniProtKB by the search term "gene:gramd1b length:[500 TO 1500]". Ninety-nine non-redundant sequences from 99 species were selected, and sequences of LAM6, LAM5, LAM4, LAM2, Ltc1, VAD1, and ZC328.3 were manually retrieved from UniProtKB. Sequence alignment of the 106 sequences was performed by Clustal

Omega, and the sequence logo was generated by WebLogo 3.7.4 (Crooks *et al*, 2004).

### Statistical analysis

No statistical method was used to predetermine sample size, and the experiments were not randomized for live cell imaging. Sample size and information about replicates are described in the figure legends. The number of biological replicates for all cell-based experiments and the number of technical replicates for all other biochemical assays are shown as the number of independent experiments within figure legends for each figure. Comparisons of data were carried out by the two-tailed unpaired Student's *t*-test, Holm–Sidak *t*-test for multiple comparisons, or the one-way ANOVA followed by Tukey or Dunnett corrections for multiple comparisons as appropriate with Prism 7 or 8 (GraphPad software). Unless $P < 0.0001$, exact $P$ values are shown within figure legends for each figure. $P > 0.05$ was considered not significant.

## Data availability

The authors declare that there are no primary datasets and computer codes associated with this study.

**Expanded View** for this article is available online.

## Acknowledgements

We thank Darshini Jeyasimman, Jingbo Sun, Nur Raihanah Binte Mohd Harion, and Esther Tan Hui En for discussion. This work was supported in part by the Singapore Ministry of Education Academic Research Fund Tier 2 (MOE2017-T2-2-001), a Nanyang Assistant Professorship (NAP), and a Lee Kong Chian School of Medicine startup grant (LKCMedicine-SUG) to Y.S. T.N. was supported by a overseas research fellowship from the Japan Society for Promotion of Science.

## Author contributions

All authors participated in the design of experiments, data analysis, and interpretation. BE performed all the structural modeling and designed GRAM domain mutations. BE and YS participated in designing the liposome-based assays that were performed by BE. TN, DHZK, and YS participated in designing the imaging and cell-based biochemical assays that were performed by TN and DHZK. BE, TN, and DHZK performed all the genetic manipulations. BE designed and performed all the protein purification work with the help of DD. TN and DHZK designed and performed lentivirus production. DHZK performed viral transduction with the help of TN. YS wrote the manuscript with input from all the authors.

## Conflict of interest

The authors declare that they have no conflict of interest.

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
