## [Review Process File · The EMBO Journal]

Molecular basis of accessible plasma membrane cholesterol recognition by the GRAM domain of GRAMD1b

Bilge Ercan, Tomoki Naito, Dylan Hong Zheng Koh, Dennis Dharmawan, and Yasunori Saheki
DOI: [10.15252/embj.2020106524](https://doi.org/10.15252/embj.2020106524)

Corresponding author(s): Yasunori Saheki (yasunori.saheki@ntu.edu.sg)

Review Timeline:

Submission Date:	13th Aug 20
Editorial Decision:	7th Sep 20
Revision Received:	24th Nov 20
Editorial Decision:	11th Dec 20
Revision Received:	16th Dec 20
Accepted:	8th Jan 21

Editor: Elisabetta Argenzio

Transaction Report:

Thank you for submitting your manuscript entitled "The molecular basis of accessible plasma membrane cholesterol recognition by the GRAM domain of GRAMD1b" (EMBOJ-2020-106524) to The EMBO Journal. Your study has been sent to three referees for evaluation and we have now received reports from them, which are enclosed below for your information.

As you can see, while the referees find your work potentially interesting, they also raise several major points that need to be addressed before they can support publication in The EMBO Journal. In particular, referee #1 stresses that more mechanistic readouts are needed to understand the role and significance of the G187L and R189W mutants. In addition, reviewer #2 finds that the specificity of GRAMD1b as a sensor for PS and cholesterol remains unclear.

We agree with the referees that these are important points and addressing these as well as all the other reviewers' criticisms will be essential to pursue publication of this study in The EMBO Journal. Strong support from the referees would also be needed for publication here. Given the overall interest of your study, I would like to invite you to submit a revised version of the manuscript according to the referees' requests. I should add that it is The EMBO Journal policy to allow only a single round of revision, and acceptance of your manuscript will therefore depend on the completeness of your responses in this revised version.

REFEREE REPORTS

Referee #1:

In their manuscript, Ercan et al. describe the binding of GRAM1b to membranes with free cholesterol, generated after its release from sphingomyelin. Its binding can be interfered with by competitor proteins. In yeast, the Lam6/Ltc1 protein is its homolog. This homology is used to elucidate the role of specific residues in the recognition of PS that boosts cholesterol binding by GRAM1b. These residues, K161 and R191, were mutagenized, which compromised binding to liposomes containing both PS and cholesterol, but not only cholesterol. They investigate the importance of the head group and side chain of cholesterol for the binding of GRAM1b. An R189W mutation is described, which gives rise to intellectual disability, and the authors detect compromised binding to cholesterol-containing liposomes for this mutant. Further data shows this mutant is unable to rescue cholesterol transfer and GRAM1b localization to the plasma membrane. A G187L mutant was then identified that showed increased binding to the plasma membrane. While most of the findings are based on detailed analysis, the full significance of the findings is hard to grasp. More mechanistic readouts are needed to understand in particular the meaning of the G187L and R189W mutants. In that sense, it is unclear whether the paper aims to identify a disease-causing mutant (incompletely done) or whether it aims to characterize the cholesterol transfer mechanism in detail with multiple new, rationally designed mutants, again incompletely done.

Specific points

1. There are some issues with the description and interpretation of the results. The authors state that K161 and R191 play a role in cooperatively binding PS and cholesterol on page 9. However, the binding of these mutants to exclusively PS-containing liposomes is more compromised than to mixed liposomes. They later state on page 10 that these residues interfere with PS-sensing properties, which seems more accurate. Figure 3D shows identical binding of the single and double mutant, but the text states there was worse binding for the double mutant. This therefore excludes cooperativity between the two sites for binding to PS and cholesterol.
2. The desmosterol addition is only examined within the DOPC mix, which has a very low level of binding. What effects are seen with the more relevant, stronger bindings?
3. The exposure of the IF images in Figure 3F is too high, not allowing to unequivocally determine whether plasma membrane staining is lost upon mutation. The authors should show end-point TIRF images instead, which should show this better. Similarly, the images of Figure 6D are also over-exposed, thus making any statement difficult.
4. The R191A construct is missing from the data in Figure 4.
5. EGFP has a size of about 30kDa and yet, in Figure 5A the tagged and untagged proteins have the same size. This cannot be right and raises serious issues about the findings in this figure. Moreover, the functionality of these fusion constructs has not been verified.
6. Could a mutant of G187L that efficiently binds to cholesterol, but does not bind to PS (K161/R191) act as a dominant-negative?
7. The SREBP cleavage assay is possibly the most relevant in terms of significance of the mutations. What is the significance of the G187L mutant for SREBP cleavage? It should act to accelerate it. What are the consequences of such an activity?

Referee #2:

In this manuscript, the authors show that the GRAM domain is important for the function of an ER-anchored sterol transfer protein, GRAMD1b. They found the amino acid residues of GRAM domain are crucial for sensing PS (K161 and R191) or accessible cholesterol (G187 and R189) in the PM. Their finding indicates that the recognition of the co-distribution of accessible cholesterol and PS by GRAM domain fine tunes the transport of cholesterol from the PM to the ER. In general, the article is clearly written and the experimental results well controlled and presented. It is very clear that PS and cholesterol sensing is required for GRAMD1b function, but the specificity of sensing these two lipids is not clear. The following are suggested to improve this manuscript.

Major points:

- 1) The competition assay in Fig 1 show a decrease binding of GRAM proteins when liposomes were treated with ECFP-D4H (a cholesterol binder) and mVenus-Lact-C2 (a PS binder). If the liposomes were fully coated by ECFP-D4H and mVenus-Lact-C2 proteins, what is observed here might not be competition but a result of steric hindrance. If the liposomes contained, for instance, the same percentage of PIP or DAG, and a PIP- or DAG-binding protein was added instead, wouldn't the same effect be observed?
- 2) Why does the addition of cholesterol lead to a decrease FRET compared to pure DOPC vesicles in Fig 1D? In fact, some FRET is observed in liposomes containing only PC contrary to stated in the text.
- 3) According to the data in Fig 2, the authors found the amino acid residues that might sense PS. In the experiments, they increased the ratio of PS/PC showing that the mutated residues are less sensitive to PS. However, it is not clear whether it is purely electrostatics due to the removal of a basic patch or whether it is specific to PS. Instead of varying amounts of PS and PC, another negatively charged lipid like PI should be used, and the amount of PC kept constant.
- 4) The K161A and R191A mutations have only minor effects on the ability to bind PS-containing liposomes. Has a double mutation been attempted and does it have a stronger effect?
- 5) Throughout the manuscript, the authors refer to a cooperative binding of PS and cholesterol. This choice of word is maybe unwise as cooperative binding has a narrow meaning in enzymology. Cooperative binding would mean for instance that binding PS would change GRAM1b affinity for cholesterol. This is not likely the case. It is more that both PS and cholesterol participate to the binding of GRAM1b to the vesicle. Multivalent or synergistic might be better terms.
- 6) In Fig 3D and 3E, it is difficult to agree that "purified GRAM1b carrying the R189W/R191A double mutation (GRAM1b-R189W/R191A) bound even less efficiently to liposomes containing cholesterol and PS than GRAM1b- R189W or GRAM1b" because the difference is minimal. This requires softer wording.
- 7) In Figure 6H-I and Fig 7A, the authors clearly show that the lower ratio of PC/cholesterol, the higher sensing of cholesterol when G187 was substituted into hydrophobic residues. However, the result might be due to a lower proportion of PC (positive-charged head group) in the liposomes (see comment 3 above for PS). It would be much more informative if the percentages of PC and PS were constant and cholesterol was replaced by a non-binding sterol such as epicholesterol, which varies minimally from cholesterol and does not affect the electrostatics of the vesicles
- 8) Can a double mutant GRAM-G187L/R191A rescue the defect of the R191A mutant with regards to PM binding and cholesterol transport?

Minor points:

- 1) Fig 1F and EV1G are as important for the claims of the paper. Yet one is stashed in the supplement, probably because the result is not nearly as conclusive. Both panels should be presented side by side with an explanation for why one yields a spectacular result and the other a much less spectacular one.
- 2) The sentence "No further reduction in binding to these liposomes was observed when the R191A mutation was added (Fig 3E)." does not appear to make sense with previous sentence. Is there any mistake? Was it meant to be "No further reduction in binding to liposomes containing no PS was observed when the R191A mutation was added (Fig 3E)"?

Referee #3:

In this paper, the authors focus on the conserved cholesterol-transfer protein called GRAMD1b. This protein is known to be transiently enriched at membrane contact sites between ER and PM. In particular, GRAMD1b is able to sense a certain threshold of accessible cholesterol at the plasma membrane, via its GRAM domain, and then transports it to the ER via its StART-like domain. This transport informs ER cholesterol sensors (linked to SREBP2) which control cholesterol biosynthesis and uptake. It is crucial to have a deeper knowledge of this type of lipid transport mechanisms because they contribute directly to cellular cholesterol homeostasis. This study unravels the molecular functioning of the N-terminal GRAM domain of GRAMD1b, that is, its binding to the membrane and the consequences on GRAMD1b activity. The main results are the discovery of two different but proximal sites in the GRAM domain for the coincident detection of accessible cholesterol and PS at the plasma membrane; the discovery that a mutation associated with intellectual disability in humans reduces the ability of GRAM to bind to membrane cholesterol; and the fact that a single glycine (G187) is critical for GRAM sensitivity to accessible cholesterol. The authors suggest that due to these GRAM molecular determinants, GRAMD1b is perfectly adapted to control the levels of accessible cholesterol in a PS-rich environment (i.e., the inner leaflet of the PM) because its proper binding to both cholesterol and PS ensures tethering at contact sites and efficient cholesterol transfer to the ER.

The authors use numerous experimental approaches comprising cell-based assays and in vitro reconstituted systems to support their hypotheses. In particular, they purify near full length GRAMD1b protein and perform tethering and sterol transfer assays between vesicles in vitro. The manuscript is pleasant to read and clear in the writing. The study seems serious and sound. However, I have some remarks and suggestions to improve the paper.

1. PS compared to other anionic lipids. The authors mention GRAM domain specificity for PS, but there is no evidence that GRAM is not able to bind to other anionic lipids such as PA, PI, PIP or PIP2. It is important to know whether GRAM's basic patches recognize a specific charge or polar head.
2. FRET assays. In Figure 1D, it is not clear why the fluorescence of ECFP at 477nm fluctuates between control conditions: (DOPC:DOPC:Chol:SM) 100:0:0:0 (black curve), 50:0:50:0 (orange curve) and 80:20:0:0 (green curve). Under these three conditions, the ECFP and mVenus probes should be far apart, so the ECFP fluo emission at 477nm should be at maximum. In addition, when adding SM, the drop in FRET at 545nm is not mirrored by an expected increase in ECFP fluorescence at 477nm. How do the authors explain this?
3. Page 9: "In contrast to PS detection, WT and mutant GRAM1b bound to liposomes that

contained only cholesterol (60%) to the same extent (~15% binding), arguing against a general role for the basic patch in membrane interaction, and instead supporting a specific role for the basic patch in PS recognition (Fig 2C)." It can be objected that here the authors use purely neutral membranes (DOPC/chol), i.e. they do not use other lipids with an acidic polar head as found in the inner leaflet of the plasma membrane in addition to PS. So in my opinion this experiment is not necessarily an argument against the general role of the basic patch in membrane interaction (see also my comment #1).

4. Basic patch mutants. The effect of the K161A and R191A mutants is relatively modest. The problem is that there are still many areas colored in blue (basic) as shown in the structural models (Fig2B) with both these mutants. The authors might consider using a double mutant K161A/R191A. It would be interesting to see the synergy between these two basic patch mutations.

5. GRAM binding control on DHE-enriched liposomes (as done with other sterols in figure 3B) seems useful to do, as the experiments in figure 4 use 10% DHE. DHE could contribute to the membrane binding of the GRAM domain. In addition, the authors do not provide evidence that the different mutants bind identically to ER-like liposomes containing DGS-NTA lipids using their Histidine tag. A control sedimentation assay would be useful here.

6. Optical turbidity assay Fig 4B. The problem here is that the WT signal is already well above the other signals at the beginning of the reaction. How do the authors interpret this?

Minor

A. DHE transfer assay: the experimental conditions do not seem perfectly optimized to appreciate transfer rates. There may be too much membranes in the assay. If the authors wish, they could try the same type of experiment with at least 5 times less lipids.

B. Page 11: Because the FRET signal between DHE and Dansyl increases over time in the experiment, Dansyl-PE is certainly not in the PM-like donor, but in the ER-like acceptor liposomes. This is to be corrected in the text and in figure 4A.

C. Page 8, second paragraph: Here the authors probably mean mCherry-GRAM1b and not EGFP-GRAM1b?

D. The color-code in the figure EV2D is odd: the acidic residues should be in red (and the basic ones in blue), it is generally recommended so. Thus E162 and D190 should be colored red.

-Sentences from the reviewers' comments are *in italics*

-Our responses are in blue

Referee #1:

In their manuscript, Ercan et al. describe the binding of GRAM1b to membranes with free cholesterol, generated after its release from sphingomyelin. Its binding can be interfered with by competitor proteins. In yeast, the Lam6/Ltc1 protein is its homolog. This homology is used to elucidate the role of specific residues in the recognition of PS that boosts cholesterol binding by GRAM1b. These residues, K161 and R191, were mutagenized, which compromised binding to liposomes containing both PS and cholesterol, but not only cholesterol. They investigate the importance of the head group and side chain of cholesterol for the binding of GRAM1b. An R189W mutation is described, which gives rise to intellectual disability, and the authors detect compromised binding to cholesterol-containing liposomes for this mutant. Further data shows this mutant is unable to rescue cholesterol transfer and GRAM1b localization to the plasma membrane. A G187L mutant was then identified that showed increased binding to the plasma membrane.

While most of the findings are based on detailed analysis, the full significance of the findings is hard to grasp. More mechanistic readouts are needed to understand in particular the meaning of the G187L and R189W mutants. In that sense, it is unclear whether the paper aims to identify a disease-causing mutant (incompletely done) or whether it aims to characterize the cholesterol transfer mechanism in detail with multiple new, rationally designed mutants, again incompletely done.

We thank the reviewer for all the constructive comments to improve our manuscript.

Specific points

1. There are some issues with the description and interpretation of the results. The authors state that K161 and R191 play a role in cooperatively binding PS and cholesterol on page 9. However, the binding of these mutants to exclusively PS-containing liposomes is more compromised than to mixed liposomes. They later state on page 10 that these residues interfere with PS-sensing properties, which seems more accurate.

Reply: We thank the reviewer for these comments. In the original text on page 9, we stated "As major defects in the ability of mutant GRAM_{1b} (e.g. basic patch mutants) to sense PS did not alter their ability to sense cholesterol, GRAM_{1b} likely possesses distinct sites for recognizing PS vs. cholesterol. Thus, these results indicate the presence of a cooperative site for detecting accessible cholesterol." We meant by these sentences that K161 and R191 are sites for recognizing PS and that there should be another site dedicated for recognizing cholesterol. We agree with the reviewer that K161A and/or R191A mutation affect the binding of the GRAM_{1b} to exclusively PS-containing liposomes while the effects of these mutations are less pronounced toward PS/Cholesterol-containing liposomes (original Fig 2C and D). To avoid confusion, we changed the original sentences to "As major defects in the ability of mutant GRAM_{1b} (e.g. basic patch mutants) to sense PS did not alter their ability to sense cholesterol, GRAM_{1b} likely possesses distinct sites for recognizing anionic lipids, including PS, and cholesterol. Thus, these results indicate the presence of another dedicated site for detecting accessible cholesterol that may act independently from the basic patch." In addition, we indicated throughout the texts that both PS and cholesterol participate in the binding of the GRAM_{1b}, which possesses "independent" binding sites for anionic lipids, including PS, and cholesterol, to membranes.

Figure 3D shows identical binding of the single and double mutant, but the text states there was worse binding for the double mutant. This therefore excludes cooperativity between the two sites for binding to PS and cholesterol.

Reply: In our independent replicates, R189W/R191A double mutant (purple line) showed “slight” reduction of binding to liposomes containing both PS and cholesterol compared to R189W single mutant (green line). Such reduction in binding, however, was not as obvious as we stated in our original manuscript. Thus, we toned down and removed the “cooperativity” from the relevant sentences as shown below.

Original sentence: “To examine the cooperativity of these sites, we generated a mutant version of GRAM_{1b} that carried the R191A basic patch mutation (defective in PS sensing) and the R189W mutation (defective in cholesterol sensing). Purified GRAM_{1b} carrying the R189W/R191A double mutation (GRAM_{1b}–R189W/R191A) bound even less efficiently to liposomes containing cholesterol and PS than GRAM_{1b}–R189W or GRAM_{1b} (~17% binding when liposomes contained 20% PS and 50% cholesterol), thereby confirming the cooperativity of cholesterol-sensing and PS-sensing properties (Fig 3D and E).”

New sentence: “We generated a mutant version of GRAM_{1b} that carried the R191A basic patch mutation (defective in PS sensing) and the R189W mutation (defective in cholesterol sensing). Purified GRAM_{1b} carrying the R189W/R191A double mutation (GRAM_{1b}–R189W/R191A) bound slightly less efficiently to liposomes containing cholesterol and PS than GRAM_{1b}–R189W (~17% binding when liposomes contained 20% PS and 50% cholesterol), thereby suggesting that cholesterol-sensing and PS-sensing properties of the GRAM domain act independently (Fig 3D and E).”

2. The desmosterol addition is only examined within the DOPC mix, which has a very low level of binding. What effects are seen with the more relevant, stronger bindings?

Reply: We performed a new set of liposome sedimentation assay to address the reviewer’s comments. We generated liposomes that contained 60% desmosterol, 20% DOPS, and 20% DOPC as well as liposomes that contained 60% cholesterol, 20% DOPS, and 20% DOPC and compared the binding of the GRAM_{1b} to these liposomes. We detected the same level of GRAM_{1b} binding to these liposomes (both at ~95%). These results are now included in **new Appendix Fig S1A**.

3. The exposure of the IF images in Figure 3F is too high, not allowing to unequivocally determine whether plasma membrane staining is lost upon mutation. The authors should show end-point TIRF images instead, which should show this better. Similarly, the images of Figure 6D are also over-exposed, thus making any statement difficult.

Reply: We adjusted the brightness and contrast uniformly across all the relevant images to avoid oversaturation of fluorescence signals (**Figs 3F, 6D, and 6F, Appendix Fig S1B**). In addition, we included end-point TIRF images (before sphingomyelinase treatment, 30 min and 60 min after sphingomyelinase treatment) as suggested by the reviewer in **new Appendix Fig S1C**. We also included a movie for the TIRF experiments (**new Movie EV1**).

4. The R191A construct is missing from the data in Figure 4.

Reply: We thank the reviewer for this comment. We purified near full-length GRAMD1b proteins carrying the R191A mutation and performed *in vitro* cholesterol transport assay. R191A mutation did not affect tethering (hence no reduction in cholesterol transport). We suspected that R191 might act redundantly with K161. Thus, we additionally purified near full-length GRAMD1b proteins carrying both R191A and K161A mutations (K161A/R191A double mutation). However, we found that the K161A/R191A double mutation did not affect

tethering nor cholesterol transport. These new data are now included in **new Appendix Figure S3A-D**.

In parallel, we purified the GRAM domain of GRAMD1b (GRAM_{1b}) carrying K161A/R191A double mutation and examined its property to interact with liposomes. Compared to purified GRAM_{1b} carrying either K161A or R191A single mutation, purified GRAM_{1b} carrying K161A/R191A double mutation 1) showed further reduction in binding to liposomes containing only PS (but no effects on binding to liposomes containing only cholesterol) (**updated Fig 2B and C**); 2) interacted slightly less efficiently with liposomes containing both PS and cholesterol (at 30%-40% cholesterol range) (**updated Fig 2D**). We also expressed EGFP-tagged GRAM_{1b} carrying K161A/R191A double mutation in GRAMD1 TKO cells and monitored its PM recruitment following sphingomyelinase treatment via confocal and TIRF microscopy and found that K161A/R191A double mutant was less recruited to the PM compared to K161A or R191A single mutant (**updated Fig 3F-H, new Appendix Fig S1B and C, and new Movie EV1**).

Because we did not detect reduction in tethering or cholesterol transport with near full-length GRAMD1b proteins carrying the K161A/R191A double mutation (despite significant impact of the K161A/R191A double mutation on the property of the GRAM_{1b} to sense anionic lipids in liposomes and the PM), we think that there are other amino acids outside of the GRAM_{1b} that contribute to anionic lipid sensing property of GRAMD1b proteins. We now inserted the following sentences in the discussion and discussed such possibility.

“Despite significant impact of the basic patch mutations on the property of the GRAM_{1b} to sense anionic lipids, GRAMD1b proteins carrying basic patch mutations alone (R191A or K161A/R191A) tethered membranes and transported cholesterol as efficiently as wild-type GRAMD1b proteins. Thus, there may be other amino acids outside of the GRAM_{1b} that contribute to anionic lipids sensing property of GRAMD1b proteins.” (page 20)

As our study focuses on the property of the GRAM domain, finding such amino acid elements is beyond the scope of our current study.

5. EGFP has a size of about 30kDa and yet, in Figure 5A the tagged and untagged proteins have the same size. This cannot be right and raises serious issues about the findings in this figure. Moreover, the functionality of these fusion constructs has not been verified.

Reply: We see the point raised by the reviewer. We note, however, that there is difference in migration between the untagged endogenous proteins of control HeLa cells and exogenously expressed EGFP-tagged GRAMD1b (EGFP-GRAMD1b) proteins in GRAMD1 triple knock-out (TKO) HeLa cells. We also note that there are at least two major isoforms of GRAMD1b proteins that are expressed in control HeLa cells (longer isoform closer to 130 kDa and shorter isoform closer to 100 kDa). To make this point clearer, we used lower percentage of the gel (6% instead of 12% that we used in the original Figure 5A) for SDS-PAGE and repeated the immunoblot analysis. In the **updated Figure 5A**, we believe that the separation of untagged endogenous GRAMD1b proteins and exogenously expressed EGFP-GRAMD1b proteins is clearer than before; both the two isoforms of endogenous untagged proteins migrate faster than EGFP-GRAMD1b. All EGFP-GRAMD1b proteins are present as single bands slightly above 130 kDa. Furthermore, the functionality of the EGFP-GRAMD1b was extensively validated in our previous manuscript (Naito et al., 2019, PMID: 31724953). In this previous work, we showed 1) that EGFP-GRAMD1b maintains the ability of GRAMD1b to localize to the ER and 2) that the expression of EGFP-GRAMD1b in GRAMD1 TKO HeLa cells (the same GRAMD1 TKO HeLa cells that we used in the current study) rescues cholesterol homeostasis defects observed in these cells (Naito et al., 2019, PMID: 31724953).

6. Could a mutant of G187L that efficiently binds to cholesterol, but does not bind to PS (K161/R191) act as a dominant-negative?

Reply: Thank you for the suggestion. We overexpressed either EGFP or EGFP-tagged GRAM domain of GRAMD1b (GRAM_{1b}) (WT, G187L, K161A, or R191A) in wild-type control HeLa cells. We then starved the cells with LPDS and mevastatin for 16 hours and examined whether the suppression of SREBP-2 cleavage following sphingomyelinase treatment (3 hours) was inhibited. We did not detect any inhibition (i.e. dominant-negative effects on cholesterol transport); robust suppression of SREBP-2 cleavage was observed in all the conditions (please see below).

Robust suppression of SREBP-2 cleavage observed in HeLa cells expressing either EGFP or EGFP-tagged GRAM domain variants of GRAMD1b (EGFP-GRAM_{1b}) as indicated. Representative immuno-blotting (IB) images are shown; n=4 independent replicates.

7. The SREBP cleavage assay is possibly the most relevant in terms of significance of the mutations. What is the significance of the G187L mutant for SREBP cleavage? It should act to accelerate it. What are the consequences of such an activity?

Reply: We performed an additional assay to determine if the suppression of SREBP-2 cleavage was accelerated in GRAMD1 TKO HeLa cells that stably expressed EGFP-GRAMD1b carrying the G187L mutation (EGFP-GRAMD1b-G187L) compared to GRAMD1 TKO HeLa cells that stably expressed wild-type EGFP-GRAMD1b (EGFP-GRAMD1b-WT). We monitored the time-course of the suppression of SREBP-2 cleavage upon sphingomyelinase treatment after starving these cells with a combination of LPDS and mevastatin for 16 hours. Cell lysates were collected before and 30 min, 90 min, or 180 min after sphingomyelinase treatment and analyzed by SDS-PAGE followed by immuno-blotting against SREBP-2. Equally robust suppression of SREBP-2 cleavage was observed over time in both conditions (**new Appendix Fig S5A**). Suppression of SREBP-2 cleavage could be reliably detected by immuno-blotting only after ~90 min, thus making it difficult to effectively assess enhanced accessible cholesterol transport of EGFP-GRAMD1b-G187L (compared to EGFP-GRAMD1b-WT) that may occur in shorter time scale (or earlier time points), by this approach. Thus, to study the significance of the G187L mutation, we focused our subsequent analysis on PM cholesterol homeostasis.

Our study suggests that the sensitivity of wild-type GRAM_{1b} toward accessible PM cholesterol is precisely tuned to detect a “transient expansion” of the accessible pool of PM cholesterol, allowing GRAMD1b to extract and transport accessible PM cholesterol to the ER only when it is needed. Such property of GRAMD1b is critically important for cells to maintain the size of the accessible pool of PM cholesterol; in cells lacking GRAMD1s, the

accessible pool of cholesterol expands in the PM (Naito et al., 2019, PMID: 31724953; Ferrari et al., 2020, PMID: 32719109). Maintaining proper size of the accessible pool of PM cholesterol is important for cell physiology; for example, recent studies showed that mammalian cells actively reduce the size of the accessible pool of PM cholesterol upon viral and bacterial infections as one of their host defense mechanisms (Wang et al., 2020, PMID: 32944968; Abrams et al., 2020, Zhou et al., 2020, PMID: 32284563; PMID: 32514064). Thus, we examined whether G187L mutation affected the size of the accessible pool of cholesterol in the PM. If the GRAM_{1b} was hyper-sensitive for sensing accessible PM cholesterol (as in the case of the G187L mutation), GRAMD1b would extract too much accessible cholesterol from the PM, thereby reducing the size of the accessible pool of PM cholesterol. This is exactly what we observed in the GRAMD1 TKO cells that stably expressed EGFP-GRAMD1b-G187L. By reflecting the enhanced cholesterol transporting property of purified GRAMD1b proteins carrying the G187L mutation *in vitro* (original Fig 7C), we found that GRAMD1 TKO cells that stably expressed EGFP-GRAMD1b-G187L are much more resistant to Amphotericin B treatment compared to GRAMD1 TKO cells that stably expressed EGFP-GRAMD1b-WT, indicating reduced size of the accessible pool of PM cholesterol in cells expressing EGFP-GRAMD1b-G187L (**new Fig EV5H**). Furthermore, in TIRF-based experiments where we monitored PM recruitment of accessible cholesterol biosensor (mCherry-GRAM_{1b}) following sphingomyelinase treatment, GRAMD1 TKO cells that stably expressed EGFP-GRAMD1b-G187L showed very little PM recruitment of mCherry-GRAM_{1b} compared to GRAMD1 TKO cells that stably expressed EGFP-GRAMD1b-WT, suggesting that extraction of accessible PM cholesterol was accelerated in GRAMD1b-G187L compared to GRAMD1b-WT *in vivo* (**new Appendix Fig S5B**). Finally, we directly measured the size of the accessible pool of PM cholesterol by staining the PM of these cells with purified mCherry-tagged D4H proteins [a well-established accessible cholesterol biosensor (e.g. Gay et al., 2015, PMID: 25809258; Maekawa and Fairn, 2015, PMID: 25663704)]. We found that GRAMD1 TKO cells that stably expressed EGFP-GRAMD1b-G187L showed significantly reduced size of the accessible pool of PM cholesterol at steady state compared to GRAMD1 TKO cells that stably expressed EGFP-GRAMD1b-WT (**new Fig 7D and E**). In contrast, GRAMD1 TKO cells that stably expressed EGFP-GRAMD1b-R189W showed significantly increased size of the accessible pool of PM cholesterol at steady state compared to GRAMD1 TKO cells that stably expressed EGFP-GRAMD1b-WT (**new Fig 7D and E**). Collectively, these new results show that G187L mutation (or R189W mutation) of the GRAM_{1b} significantly alters the size of the accessible pool of PM cholesterol as the consequences of accelerated (or impaired) GRAMD1-dependent accessible cholesterol transport in cells.

Referee #2:

In this manuscript, the authors show that the GRAM domain is important for the function of an ER-anchored sterol transfer protein, GRAMD1b. They found the amino acid residues of GRAM domain are crucial for sensing PS (K161 and R191) or accessible cholesterol (G187 and R189) in the PM. Their finding indicates that the recognition of the co-distribution of accessible cholesterol and PS by GRAM domain fine tunes the transport of cholesterol from the PM to the ER. In general, the article is clearly written and the experimental results well controlled and presented. It is very clear that PS and cholesterol sensing is required for GRAMD1b function, but the specificity of sensing these two lipids is not clear.

We thank the reviewer for these very positive comments.

The following are suggested to improve this manuscript.

Major points:

1) The competition assay in Fig 1 show a decrease binding of GRAM proteins when liposomes were treated with ECFP-D4H (a cholesterol binder) and mVenus-Lact-C2 (a PS binder). If the liposomes were fully coated by ECFP-D4H and mVenus-Lact-C2 proteins, what is observed here might not be competition but a result of steric hindrance. If the liposomes contained, for instance, the same percentage of PIP or DAG, and a PIP- or DAG-binding protein was added instead, wouldn't the same effect be observed?

Reply: We thank the reviewer for these comments. We previously showed that the GRAM domain of GRAMD1 acts as a co-incidence detector of both accessible cholesterol and anionic lipids, including PS and PIP [PI(4)P and PI(4,5)P₂] (Naito et al., 2019, PMID: 31724953). Incorporation of PIP to liposomes induced enhanced binding of the GRAM domain of GRAMD1b (GRAM_{1b}) to cholesterol-containing liposomes as incorporation of PS did (Naito et al., 2019, PMID: 31724953) (please also see below our reply to the comment #3 of the reviewer). Thus, incorporating the same amount of PIP and PS in cholesterol-containing liposomes and performing a competition assay with PIP-binding proteins is expected to reduce the binding of GRAM_{1b} to the liposomes by masking PIP (still not excluding the possibility of steric hindrance). Alternatively, we considered using the same amount of DAG and PS in cholesterol-containing liposomes and performing a competition assay with DAG-binding proteins. We encountered two problems with this strategy: 1) we could not incorporate the same amount of DAG and PS in cholesterol-containing liposomes; 2) we realized that the C1 domain, a major DAG-binding module, has an ability to bind to PS in addition to DAG (the C1 domain also acts as a co-incidence detector of DAG and PS) (e.g. Stewart et al., 2014, PMID: 25124034; Kazanietz et al, 1995, PMID: 7782331; Bittova et al., 2001, PMID: 11029472) (please see below our data showing interaction of purified C1 domains of rat PKC δ to liposomes containing only PS).

Interaction of C1 domains of PKC δ with liposomes. (A) A SEC profile of purified GST-tagged C1 domains (C1A+C1B) of rat PKC δ (GST-PKC δ -C1). (B) A representative gel image of a liposome sedimentation assay as indicated. Quantification shows ~50% binding of GST-PKC δ -C1 to liposomes containing only PS (n=3 independent experiments).

These issues prevented us from using this well-characterized DAG-binding module as a competitor to specifically examine the possibility of steric hindrance. Thus, we toned down the relevant sentence in the result section as shown below to discuss the possibility of steric hindrance (page 7).

Original sentence: "Thus, even when the total amount of cholesterol within the membrane was held constant, masking either accessible cholesterol (via ECFP-D4H) or PS (via mVenus-Lact-C2) inhibited binding of the GRAM domain to membranes."

New sentence: "Thus, even when the total amount of cholesterol within the membrane was held constant, masking either accessible cholesterol (via ECFP-D4H) or PS (via mVenus-Lact-C2) inhibited binding of the GRAM domain to membranes, although possible steric hindrance mediated by these competitor proteins may have also contributed to reduction in binding."

2) *Why does the addition of cholesterol lead to a decrease FRET compared to pure DOPC vesicles in Fig 1D? In fact, some FRET is observed in liposomes containing only PC contrary to stated in the text.*

Reply: We thank the reviewer for these comments. We re-purified ECFP-D4H and mVenus-Lact-C2 proteins and repeated the experiments with a fresh batch of lipids. Our new results show that there is no decrease in FRET signals in cholesterol-added liposomes [(DOPC:DOPS:Chol:SM) 50:0:50:0] (orange curve) compared to pure DOPC liposomes [(DOPC:DOPS:Chol:SM) 100:0:0:0] (black curve). Our previous results were replaced by these new results (**new Fig 1D and E**).

3) *According to the data in Fig 2, the authors found the amino acid residues that might sense PS. In the experiments, they increased the ratio of PS/PC showing that the mutated residues are less sensitive to PS. However, it is not clear whether it is purely electrostatics due to the removal of a basic patch or whether it is specific to PS. Instead of varying amounts of PS and PC, another negatively charged lipid like PI should be used, and the amount of PC kept constant.*

Reply: We previously showed that the GRAM domain of GRAMD1 acts as a co-incidence detector of both accessible cholesterol and anionic lipids, including PS and PIP [PI(4)P and PI(4,5)P₂] (Naito et al., 2019, PMID: 31724953). To address whether the basic patch (that is formed by K161 and R191) of GRAM_{1b} has a specific role in PS sensing or a general role in anionic lipid sensing, we additionally examined the binding of the purified GRAM domains of GRAMD1b (GRAM_{1b}) carrying K161A and/or R191A mutation (we now include new data from GRAM_{1b} carrying K161A/R191A double mutation in the revised manuscript) to liposomes containing a fixed amount of anionic lipids [either PS, phosphatidic acid (PA), PI(4)P, PI(4,5)P₂, or phosphatidylinositol (PI)] (10%), DOPC (40%), and cholesterol (50%). Wild-type GRAM_{1b} bound liposomes that contained any one of these anionic lipids with some preference toward PS, PA, PI(4)P, and PI(4,5)P₂, consistent with our previous results (Naito et al., 2019, PMID: 31724953). We found that K161A, R191A, or K161A/R191A reduced (but not eliminated) the binding of the GRAM_{1b} to all these liposomes, with K161A/R191A showing the strongest impact (**new Fig 2E**). Based on these results, we think that the basic patch of the GRAM_{1b} contributes primarily to electrostatic interaction with anionic lipids in general. We inserted a following sentence "These results support a general role for the basic patch in anionic lipid sensing primarily through electrostatic interaction (Fig 2E)." in the section of the results where we described these new data (page 9). In the model in **updated Figure 7F**, we changed "PS sensing" to "anionic lipid sensing" to make it more general.

4) *The K161A and R191A mutations have only minor effects on the ability to bind PS-*

containing-liposomes. Has a double mutation been attempted and does it have a stronger effect?

Reply: Thank you for the suggestion. We additionally purified the GRAM domain of GRAMD1b (GRAM_{1b}) carrying K161A/R191A double mutation and examined its property to interact with liposomes. Compared to purified GRAM_{1b} carrying either K161A or R191A single mutation, purified GRAM_{1b} carrying K161A/R191A double mutation 1) showed further reduction in binding to liposomes containing only PS (but no effects on binding to liposomes containing only cholesterol) (**updated Fig 2B and C**); 2) interacted slightly less efficiently with liposomes containing both PS and cholesterol (at 30%-40% cholesterol range) (**updated Fig 2D**). We also expressed EGFP-tagged GRAM_{1b} carrying K161A/R191A double mutation in GRAMD1 TKO cells and monitored its PM recruitment following sphingomyelinase treatment via confocal and TIRF microscopy and found that K161A/R191A double mutant was less recruited to the PM compared to K161A or R191A single mutants (**updated Fig 3F-H, new Appendix Fig S1B and C, and new Movie EV1**). These results suggest that both K161 and R191 contribute to the formation of the basic patch of GRAM_{1b}. However, we note that even the K161A/R191A double mutation does not completely eliminate the binding of the GRAM_{1b} to liposomes containing both cholesterol and PS (or other anionic lipids) (**updated Fig 2D and new Fig 2E**). While K161 and R191 have major roles in anionic lipid sensing property of the GRAM_{1b}, we think there might be other amino acids that additionally contribute to anionic lipid sensing property of this domain.

5) Throughout the manuscript, the authors refer to a cooperative binding of PS and cholesterol. This choice of word is maybe unwise as cooperative binding has a narrow meaning in enzymology. Cooperative binding would mean for instance that binding PS would change GRAM1b affinity for cholesterol. This is not likely the case. It is more that both PS and cholesterol participate to the binding of GRAM1b to the vesicle. Multivalent or synergistic might be better terms.

Reply: We thank the reviewer for these comments. We replaced the word “cooperative” to “synergistic” throughout the texts. We also made it clear that both PS and cholesterol participate in the binding of the GRAM domain of GRAMD1b (GRAM_{1b}), which possesses “independent” binding sites for PS and cholesterol, to membranes.

6) In Fig 3D and 3E, it is difficult to agree that "purified GRAM1b carrying the R189W/R191A double mutation (GRAM1b-R189W/R191A) bound even less efficiently to liposomes containing cholesterol and PS than GRAM1b- R189W or GRAM1b" because the difference is minimal. This requires softer wording.

Reply: We agree with the reviewer. As R189W single mutant already showed much reduced binding to these liposomes (compared to wild-type control), such reduction in binding was not as obvious as we stated in our original manuscript. Thus, we toned down the relevant sentences as shown below.

Original sentence: “To examine the cooperativity of these sites, we generated a mutant version of GRAM_{1b} that carried the R191A basic patch mutation (defective in PS sensing) and the R189W mutation (defective in cholesterol sensing). Purified GRAM_{1b} carrying the R189W/R191A double mutation (GRAM_{1b}-R189W/R191A) bound even less efficiently to liposomes containing cholesterol and PS than GRAM_{1b}- R189W or GRAM_{1b} (~17% binding when liposomes contained 20% PS and 50% cholesterol), thereby confirming the cooperativity of cholesterol-sensing and PS-sensing properties (Fig 3D and E).”

New sentence: “We generated a mutant version of GRAM_{1b} that carried the R191A basic patch mutation (defective in PS sensing) and the R189W mutation (defective in cholesterol sensing). Purified GRAM_{1b} carrying the R189W/R191A double mutation (GRAM_{1b}-

R189W/R191A) bound slightly less efficiently to liposomes containing cholesterol and PS than GRAM_{1b}-R189W (~17% binding when liposomes contained 20% PS and 50% cholesterol), thereby suggesting that cholesterol-sensing and PS-sensing properties of the GRAM domain act independently (Fig 3D and E).”

7) In Figure 6H-I and Fig 7A, the authors clearly show that the lower ratio of PC/cholesterol, the higher sensing of cholesterol when G187 was substituted into hydrophobic residues. However, the result might be due to a lower proportion of PC (positive-charged head group) in the liposomes (see comment 3 above for PS). It would be much more informative if the percentages of PC and PS were constant and cholesterol was replaced by a non-binding sterol such as epicholesterol, which varies minimally from cholesterol and does not affect the electrostatics of the vesicles

Reply: We thank the reviewer for these comments. We performed additional liposome sedimentation assays as the reviewer suggested. In one set of experiments, we used fixed amount of PC (30%), PS (10%) and sterol (60%). In the other set of experiments, we used fixed amount of PC (40%) and sterol (60%). In both conditions, we used different ratio of cholesterol and epicholesterol (i.e. 20% vs. 40%, 30% vs. 30%, 40% vs. 20%, 50% vs. 10%, 60% vs. 0%) and examined the binding of purified GRAM_{1b} (wild-type and G187L mutant) to the liposomes. We observed that purified GRAM_{1b} carrying the G187L mutation showed stronger binding to liposomes compared to wild-type GRAM_{1b} as sterol in liposomes was gradually dominated by cholesterol. These new data are now included in **new Figure EV4D and E**. For clarification, experiments that were shown in the original Figure 7A were performed in cells expressing EGFP-tagged GRAM_{1b} via microscopy, in contrasts to the *in vitro* liposome sedimentation assays that are shown in the original Figure 6H and I.

8) Can a double mutant GRAM-G187L/R191A rescue the defect of the R191A mutant with regards to PM binding and cholesterol transport?

Reply: We expressed EGFP-tagged GRAM domain of GRAMD1b (GRAM_{1b}) carrying G187L/R191A double mutation (EGFP-GRAM_{1b} G187L/R191A) in GRAMD1 TKO cells and examined its PM recruitment upon sphingomyelinase treatment and compared it to EGFP-GRAM_{1b} carrying R191A single mutation (EGFP-GRAM_{1b} R191A) via confocal and TIRF microscopy. EGFP-GRAM_{1b} G187L/R191A showed much stronger PM recruitment compared to EGFP-GRAM_{1b} R191A, thereby rescuing the effect of R191A mutation with regards to PM binding (**new Appendix Fig S4A-C**).

To test whether G187L mutation rescues the possible defect of cholesterol transport in GRAMD1b carrying R191A mutation, we first examined the effect of the R191A mutation to GRAMD1b-dependent cholesterol transport *in vitro*. We purified near full-length GRAMD1b proteins carrying the R191A mutation and performed *in vitro* cholesterol transport assay. R191A mutation did not affect tethering (hence no reduction in cholesterol transport). We suspected that R191 might act redundantly with K161. Thus, we additionally purified near full-length GRAMD1b proteins carrying both R191A and K161A mutations (K161A/R191A double mutation). However, we found that the K161A/R191A double mutation did not affect tethering nor cholesterol transport. These new data are now included in **new Appendix Figure S3A-D**. Because we did not detect reduction in tethering or in cholesterol transport with near full-length GRAMD1b proteins carrying the K161A/R191A double mutation (despite significant impact of the K161A/R191A double mutation on the property of the GRAM_{1b} to sense anionic lipids in liposomes and the PM), we think that there are other amino acids outside of the GRAM_{1b} that contribute to anionic lipid sensing property of GRAMD1b proteins. We now inserted the following sentences in the discussion and discussed such possibility. As our study focuses on the property of the GRAM domain, finding such amino acid elements is beyond the scope of our current study.

“Despite significant impact of the basic patch mutations on the property of the GRAM_{1b} to sense anionic lipids, GRAMD1b proteins carrying basic patch mutations alone (R191A or K161A/R191A) tethered membranes and transported cholesterol as efficiently as wild-type GRAMD1b proteins. Thus, there may be other amino acids outside of the GRAM_{1b} that contribute to anionic lipids sensing property of GRAMD1b proteins.” (page 20)

As R191A mutant is normal in cholesterol transport as stated above, we were unable to examine whether G187L mutation could rescue R191A mutant in cholesterol transport.

Minor points:

1) *Fig 1F and EV1G are as important for the claims of the paper. Yet one is stashed in the supplement, probably because the result is not nearly as conclusive. Both panels should be presented side by side with an explanation for why one yields a spectacular result and the other a much less spectacular one.*

Reply: We moved the original Figure EV1G to **new Figure 1G**. We think that one of the reasons why we do not see strong reduction in PM recruitment of the EGFP-tagged GRAM domain of GRAMD1b (EGFP-GRAM_{1b}) by PS masking via mCherry-Lact-C2 overexpression (compared to accessible cholesterol masking via purified ECFP-D4H proteins) may be the presence of other anionic lipids (e.g. PI(4,5)P₂, PI4P etc.) in addition to PS in the PM. These other anionic lipids may contribute to the residual interaction of the GRAM_{1b} to the PM (our new results in **new Figure 2E** support such possibility). We added the following sentences in the section of results.

“The moderate effect of mCherry-Lact-C2 overexpression suggests that the binding of GRAM_{1b} to the PM may additionally depend on other anionic lipids within this bilayer.” (page 8)

2) *The sentence "No further reduction in binding to these liposomes was observed when the R191A mutation was added (Fig 3E)." does not appear to make sense with previous sentence. Is there any mistake? Was it meant to be "No further reduction in binding to liposomes containing no PS was observed when the R191A mutation was added (Fig 3E)"?*

Reply: Thank you for pointing out this error. Yes, this is what we meant. To avoid confusion, we replaced the sentence to “No further reduction in binding to liposomes containing only cholesterol was observed when the R191A mutation was added (Fig 3E).”

Referee #3:

In this paper, the authors focus on the conserved cholesterol-transfer protein called GRAMD1b. This protein is known to be transiently enriched at membrane contact sites between ER and PM. In particular, GRAMD1b is able to sense a certain threshold of accessible cholesterol at the plasma membrane, via its GRAM domain, and then transports it to the ER via its StART-like domain. This transport informs ER cholesterol sensors (linked to SREBP2) which control cholesterol biosynthesis and uptake. It is crucial to have a deeper knowledge of this type of lipid transport mechanisms because they contribute directly to cellular cholesterol homeostasis. This study unravels the molecular functioning of the N-terminal GRAM domain of GRAMD1b, that is, its binding to the membrane and the consequences on GRAMD1b activity. The main results are the discovery of two different but proximal sites in the GRAM domain for the coincident detection of accessible cholesterol and PS at the plasma membrane; the discovery that a mutation associated with intellectual disability in humans reduces the ability of GRAM to bind to membrane cholesterol; and the fact that a single glycine (G187) is critical for GRAM sensitivity to accessible cholesterol. The authors suggest that due to these GRAM molecular determinants, GRAMD1b is perfectly adapted to control the levels of accessible cholesterol in a PS-rich environment (i.e., the inner leaflet of the PM) because its proper binding to both cholesterol and PS ensures tethering at contact sites and efficient cholesterol transfer to the ER.

The authors use numerous experimental approaches comprising cell-based assays and in vitro reconstituted systems to support their hypotheses. In particular, they purify near full-length GRAMD1b protein and perform tethering and sterol transfer assays between vesicles in vitro. The manuscript is pleasant to read and clear in the writing. The study seems serious and sound.

We thank the reviewer for these very positive comments.

However, I have some remarks and suggestions to improve the paper.

1. PS compared to other anionic lipids. The authors mention GRAM domain specificity for PS, but there is no evidence that GRAM is not able to bind to other anionic lipids such as PA, PI, PIP or PIP2. It is important to know whether GRAM's basic patches recognize a specific charge or polar head.

Reply: We previously showed that the GRAM domain of GRAMD1 acts as a co-incidence detector of both accessible cholesterol and anionic lipids, including PS and PIP [PI(4)P and PI(4,5)P₂] (Naito et al., 2019, PMID: 31724953). To address whether the basic patch (that is formed by K161 and R191) of GRAM_{1b} has a specific role in PS sensing or a general role in anionic lipid sensing, we additionally examined the binding of the purified GRAM domains of GRAMD1b (GRAM_{1b}) carrying K161A and/or R191A mutation (we now include new data from GRAM_{1b} carrying K161A/R191A double mutation in the revised manuscript) to liposomes containing a fixed amount of anionic lipids [either PS, phosphatidic acid (PA), PI(4)P, PI(4,5)P₂, or phosphatidylinositol (PI)] (10%), DOPC (40%), and cholesterol (50%). Wild-type GRAM_{1b} bound liposomes that contained any one of these anionic lipids with some preference toward PS, PA, PI(4)P, and PI(4,5)P₂, consistent with our previous results (Naito et al., 2019, PMID: 31724953). We found that K161A, R191A, or K161A/R191A reduced (but not eliminated) the binding of the GRAM_{1b} to all these liposomes, with K161A/R191A showing the strongest impact (**new Fig 2E**). Based on these results, we think that the basic patch of the GRAM_{1b} contributes primarily to electrostatic interaction with anionic lipids in general. We inserted a following sentence “These results support a general role for the basic patch in anionic lipid sensing primarily through electrostatic interaction (Fig 2E).” in the section of the results where we described these new data (page 9). In the model in **updated Figure 7F**, we changed “PS sensing” to “anionic lipid sensing” to make it more general.

2. FRET assays. In Figure 1D, it is not clear why the fluorescence of ECFP at 477nm fluctuates between control conditions: (DOPC:DOPC:Chol:SM) 100:0:0:0 (black curve), 50:0:50:0 (orange curve) and 80:20:0:0 (green curve). Under these three conditions, the ECFP and mVenus probes should be far apart, so the ECFP fluo emission at 477nm should be at maximum. In addition, when adding SM, the drop in FRET at 545nm is not mirrored by an expected increase in ECFP fluorescence at 477nm. How do the authors explain this?

Reply: We thank the reviewer for these important comments. As the reviewer pointed out, ECFP-D4H and mVenus-Lact-C2 should be far apart in the three conditions [(DOPC:DOPS:Chol:SM) 100:0:0:0 (black curve), 50:0:50:0 (orange curve) and 80:20:0:0 (green curve)]. We re-purified ECFP-D4H and mVenus-Lact-C2 proteins and repeated the experiments with a fresh batch of lipids. Our new results show that ECFP fluorescence at 477nm is at maximum in these three conditions (**new Fig 1D**). However, we still noted that ECFP fluorescence of DOPC:DOPS:Chol:SM (50:0:50:0) (orange curve) was slightly reduced compared to DOPC:DOPS:Chol:SM (100:0:0:0) or DOPC:DOPS:Chol:SM (80:20:0:0). We think that this might be due to the self-quenching of ECFP in a locally crowded environment where ECFP-D4H proteins are strongly bound to the cholesterol-containing liposomes [DOPC:DOPS:Chol:SM (50:0:50:0)]. Similar self-quenching phenomenon of fluorescent proteins have been reported for mCherry (Kruitwagen et al., 2015, PMID: 26615018). With the fresh batch of proteins and lipids, the drop in FRET signals at 545nm by addition of SM was indeed mirrored by slight increase in ECFP fluorescence at 477nm. We replaced the old data with these new data sets (**new Fig 1D and E**). We removed “dramatically” from the original sentence to tone down the effect of SM and add a few words to describe the additional increase of ECFP fluorescence (page 7).

3. Page 9: "In contrast to PS detection, WT and mutant GRAM1b bound to liposomes that contained only cholesterol (60%) to the same extent (~15% binding), arguing against a general role for the basic patch in membrane interaction, and instead supporting a specific role for the basic patch in PS recognition (Fig 2C)." It can be objected that here the authors use purely neutral membranes (DOPC/chol), i.e. they do not use other lipids with an acidic polar head as found in the inner leaflet of the plasma membrane in addition to PS. So in my opinion this experiment is not necessarily an argument against the general role of the basic patch in membrane interaction (see also my comment #1).

Reply: We thank the reviewer for these comments. As we stated in the reply to the reviewer's comment #1 above, we think that the basic patch of the GRAM domain contributes to anionic lipid sensing in general primarily through electrostatic interaction. We removed the phrase “arguing against a general role for the basic patch in membrane interaction, and instead supporting a specific role for the basic patch in PS recognition (Fig 2C).” from the relevant sentences to avoid confusion.

4. Basic patch mutants. The effect of the K161A and R191A mutants is relatively modest. The problem is that there are still many areas colored in blue (basic) as shown in the structural models (Fig2B) with both these mutants. The authors might consider using a double mutant K161A/R191A. It would be interesting to see the synergy between these two basic patch mutations.

Reply: Thank you for the suggestion. We additionally purified the GRAM domain of GRAMD1b (GRAM_{1b}) carrying K161A/R191A double mutation and examined its property to interact with liposomes. Compared to purified GRAM_{1b} carrying either K161A or R191A single mutation, purified GRAM_{1b} carrying K161A/R191A double mutation 1) showed further reduction in binding to liposomes containing only PS (but no effects on binding to liposomes containing only cholesterol) (**updated Fig 2B and C**); 2) interacted slightly less efficiently with liposomes containing both PS and cholesterol (at 30%-40% cholesterol range)

(**updated Fig 2D**). We also expressed EGFP-tagged GRAM_{1b} carrying K161A/R191A double mutation in GRAMD1 TKO cells and monitored its PM recruitment following sphingomyelinase treatment via confocal and TIRF microscopy and found that K161A/R191A double mutant was less recruited to the PM compared to K161A or R191A single mutants (**updated Fig 3F-H, new Appendix Fig S1B and C, and new Movie EV1**). These results suggest that both K161 and R191 contribute to the formation of the basic patch of GRAM_{1b}. However, we note that even the K161A/R191A double mutation does not completely eliminate the binding of the GRAM_{1b} to liposomes containing both cholesterol and PS (or other anionic lipids) (**updated Fig 2D and new Fig 2E**). While K161 and R191 have major roles in anionic lipid sensing property of the GRAM_{1b}, we think there might be other amino acids that additionally contribute to anionic lipid sensing property of this domain.

5. *GRAM binding control on DHE-enriched liposomes (as done with other sterols in figure 3B) seems useful to do, as the experiments in figure 4 use 10% DHE. DHE could contribute to the membrane binding of the GRAM domain. In addition, the authors do not provide evidence that the different mutants bind identically to ER-like liposomes containing DGS-NTA lipids using their Histidine tag. A control sedimentation assay would be useful here.*

Reply: We performed additional liposome sedimentation assays to examine 1) binding of the purified GRAM domain of GRAMD1b (GRAM_{1b}) to DHE-enriched liposomes (60% DHE; 40% DOPC); 2) the efficiency of different variants of purified near full-length GRAMD1b proteins that we used in the *in vitro* lipid transport assay (wild-type, R189W, R189W/R191A, R191A, K161A/R191A, and G187L) to interact with ER-like liposomes containing DGS-NTA lipids. We found 1) that the purified GRAM_{1b} showed minimal binding with DHE-enriched liposomes compared to cholesterol-enriched liposomes (**updated Fig 3A and B**) and 2) that all the variants of near full-length GRAMD1b proteins bound equally and robustly to ER-like liposomes containing DGS-NTA lipids (**new Fig EV5E and new Appendix Fig S3E**).

6. *Optical turbidity assay Fig 4B. The problem here is that the WT signal is already well above the other signals at the beginning of the reaction. How do the authors interpret this?*

Reply: We see the point of the reviewer. In our assay system, there is some time lag (~30 seconds) between the time when we aliquot the mixture of liposomes and purified proteins into individual wells of a 96 well plate and the time when the plate reader starts to read the optical turbidity of individual wells. Our core facility does not currently possess a cuvette system, where we can inject proteins while the optical turbidity of liposomes (or light scattering) is constantly measured. We mention the technical limitation of our assay in the materials and methods (page 27).

Minor

A. *DHE transfer assay: the experimental conditions do not seem perfectly optimized to appreciate transfer rates. There may be too much membranes in the assay. If the authors wish, they could try the same type of experiment with at least 5 times less lipids.*

Reply: Thank you for the suggestion. We repeated the assay with 5 times less lipids (0.2 μ M liposomes in total). Although we could still detect transfer of DHE between donor and acceptor liposomes with purified near full-length GRAMD1b proteins, signals of the turbidity (absorbance at 405 nm), which reflect tethering of these vesicles, became too low to detect them reliably (please see below for the results of three independent replicates: 0.05 μ M near full-length wild-type GRAMD1b proteins). Thus, we decided to use the current condition, which allowed us to measure both DHE transport and turbidity reliably and simultaneously.

Representative results from a liposome tethering assay with addition of purified near full-length wild-type GRAMD1b proteins to PM-like and ER-like liposomes (5 times less lipids in the reaction mixture compared to the condition that was used in the assay shown in the manuscript) showing high variability in turbidity signals.

B. Page 11: Because the FRET signal between DHE and Dansyl increases over time in the experiment, Dansyl-PE is certainly not in the PM-like donor, but in the ER-like acceptor liposomes. This is to be corrected in the text and in figure 4A.

Reply: FRET signals between DHE and Dansyl-PE actually decrease over time in our experiments (please see below an example of our assays: 0.2 μ M near full-length GRAMD1b proteins). We then converted the decrease of FRET signals into the amount of DHE transferred from donor PM-like liposomes to acceptor ER-like liposomes (y axis of Figure 4C) (please see materials and methods in the manuscript). Similar method was previously used by Karin Reinisch's group for measuring StART-like domain-dependent DHE transport from donor to acceptor liposomes (Horenkamp et al., 2018, PMID: 29467216).

A representative graph showing the decrease of FRET signals over time upon addition of purified near full-length wild-type GRAMD1b proteins to PM-like and ER-like liposomes.

C. Page 8, second paragraph: Here the authors probably mean mCherry-GRAM1b and not EGFP-GRAM1b?

Reply: Thank you for pointing it out. We fixed it to mCherry-GRAM_{1b}.

D. The color-code in the figure EV2D is odd: the acidic residues should be in red (and the basic ones in blue), it is generally recommended so. Thus E162 and D190 should be colored red.

Reply: We changed the color of E162 and D190 to red in **new Figure EV2C and D**.

Thank you for submitting a revised version of your study. The manuscript has now been sent back to the original referees, whose comments are appended below.

As you will see, reviewer #1 finds that his/her criticisms have been sufficiently addressed and recommends the study for publication. However, referee #2 and #3 have some remaining points that need to be solved before we can officially accept the manuscript. In addition, there are a few editorial issues concerning the text and the figures that I need you to address.

REFeree REPORTS

Referee #1:

The authors have nicely addressed all concerns that I had. I would only like to request a rephrasing of the description of Figures 5B/C, as I find it hard to understand the significance of transient versus stable expression of the constructs for PM recruitment and cholesterol accumulation.

Referee #2:

In their revised manuscript Ercan et al., have added a considerable amount of new experiments that

come a long way to address the reviewers comments.

In particular, they have now expressed and purified mutant versions of GRAMD1b harboring the two mutations claimed to impair PS binding as part of a near full-length protein. Surprisingly, they found that the mutations had no detectable effects *in vitro* despite showing promise in the context of isolated GRAM domains, as well as *in vivo*. This complicates the interpretation of the data, and instills questions about the validity of the model. By summing the facts that the GRAM domain does not recognize PS but rather any anionic lipid, that PS-binding mutants affect membrane-recruitment of an isolated domain but not of the whole protein *in vitro*, but affect membrane recruitment of the whole protein *in vivo*, it becomes difficult to exclude that the mutations have an effect different than the one put forward here. Particularly concerning is the discrepancy between *in vivo* and *in vitro* binding of the full-length protein. The authors try to reconcile the absence of binding deficiency of their mutant by arguing that "there may be other amino acids outside of the GRAM1b that contribute to anionic lipid sensing property of GRAMD1b proteins." However this explanation is inconsistent with the fact that the full-length mutant protein is defective for recruitment *in vivo*.

The discrepancy between the behaviour of the protein *in vitro* and *in vivo* casting important doubt on the validity of the conclusions, it is important to, first emphasize this discrepancy, and second acknowledge the possibility that, *in vivo*, the two PS-binding mutants might affect the protein by other mechanisms than preventing PS binding.

Additional points:

-Competitive lipid-binding versus steric hindrance is not addressed as the authors could not find a suitable lipid, outside of PS, to perform binding assays (DAG and PIP would not work for reasons well explained). It is nevertheless a pity that the authors did not experiment further on this point. They could for instance, dope their liposomes with Ni-NTA lipids and try to compete with His-tagged GFP (provided their recombinant protein is not itself His-tagged), or repeat the competition assays with mutant GRAM1b domains that cannot bind cholesterol or PS. The expectation would be that a cholesterol-binding deficient mutant will be competed by the C2 domain competitor but not the D4H-competitor, and vice versa for a PS-binding-deficient mutant. If not, then this would mean that steric hindrance is indeed the main cause of competition and not competitive lipid binding. Without addressing this point, the data of competitive binding have very little significance. It might shorten the manuscript to remove them entirely.

the manuscript is focused on PS binding, yet the authors show that GRAMD1b appears to bind other anionic lipids just as well, including PI and PIP₂, both of which are abundant in the PM. Why put such an emphasis on PS? Why not replace all mentions of PS-binding by anionic lipid binding?

-Fig 5B shows an assay where a cholesterol sensor is recruited to the PM if no WT GRAMD1b is present. This is used to claim that GRAMD1b transports free sterol away from the PM. An alternative explanation is that WT GRAMD1b binds to the same free cholesterol as the sensor and competes it away. The good control for that experiment would be to express a mutant of GRAMD1b that can bind the PM, but cannot transport sterols (a mutant of the STAR domain for instance). This control is not performed. Was it made in a previous paper? If yes a reference for this assay would be welcome. If not, what allows to conclude that competition rather than transport is at play? My apologies for not spotting this in the 1st round of review.

Referee #3:

Most of the experiments I requested have been carried out successfully.

1) I had suggested to the authors to try to optimize the DHE-transfer assay between liposomes, because in my opinion the sterol-transfer activity measured for GRAMD1b might be underestimated. For comparison, Fig EV5D indicates a DHE transfer rate (for GRAMD1b WT) of ~ 0.02 DHE/s/GRAMD1b, i.e., 1.2 DHE/min/molecule GRAMD1b, whereas in the paper Horenkamp et al. 2018 EMBOj, Fig2D, the rate is 40 DHE/min/molecule GRAMD1b. Horenkamp et al used a GRAMD1b construct that is also capable of tethering membranes (via Lact-C2 domain at N-ter). The >30 -fold difference in activity between the two studies is puzzling, even if the assay conditions are different. Can the authors comment on that?

2) In the manuscript the unit describing the DHE transfer rate is $\text{nmole.GRAMD1b}^{-1}.\text{s}^{-1}$ (in Fig 4C, Fig 7C, Appendix Fig S3D), but I would suggest $\text{DHE.GRAMD1b}^{-1}.\text{s}^{-1}$, where DHE is 'molecule' not 'nmole'. This better fits with the data presented. Can the authors check this?

-Sentences from the reviewers' comments are *in italics*

-Our responses are in blue

Referee #1:

The authors have nicely addressed all concerns that I had. I would only like to request a rephrasing of the description of Figures 5B/C, as I find it hard to understand the significance of transient versus stable expression of the constructs for PM recruitment and cholesterol accumulation.

Reply: We appreciate the reviewer for the very nice comment. We rephrased the description of Figures 5B/C in the manuscript text to clarify the significance of transient versus stable expression as shown below (page 12).

Original sentences: Stably expressed EGFP-GRAMD1b proteins localized to the ER as discrete patches, similar to previously reported localization of transiently expressed and functional EGFP-GRAMD1b (Fig 5B, (Naito et al., 2019)).

New sentences: "Stably expressed EGFP-GRAMD1b proteins localized to the ER as discrete patches, similar to previously reported localization of transiently transfected and functional EGFP-GRAMD1b (Naito et al., 2019), allowing us to compare the activities of various versions of EGFP-GRAMD1b proteins at similar levels of expression in GRAMD1 TKO cells (Fig 5B)."

Referee #2:

In their revised manuscript Ercan et al., have added a considerable amount of new experiments that come a long way to address the reviewers comments.

Thank you for the very nice comments. We appreciate the reviewer for all the constructive comments and suggestions to improve our manuscript.

*In particular, they have now expressed and purified mutant versions of GRAMD1b harboring **the two mutations claimed to impair PS binding as part of a near full-length protein**. Surprisingly, they found that the mutations had no detectable effects in vitro despite showing promise in the context of isolated GRAM domains, as well as in vivo.*

Reply: We think the reviewer is specifically concerned about the two PS/anionic lipid-binding mutations (i.e. the R191A single mutation and the K161A/R191A double mutation), which we further characterized during the previous round of revision.

*This complicates the interpretation of the data, and instills questions about the validity of the model. By summing the facts that the GRAM domain does not recognize PS but rather any anionic lipid, that PS-binding mutants affect membrane-recruitment of an isolated domain but not of the whole protein in vitro, but **affect membrane recruitment of the whole protein in vivo**, it becomes difficult to exclude that the mutations have an effect different than the one put forward here.*

Reply: To be clear, we did not provide any data/evidence showing that the R191A single mutation or the K161A/R191A double mutation affect membrane (i.e. plasma membrane: PM) recruitment of the whole protein (i.e. full-length GRAMD1 protein) *in vivo* in our manuscript (in contrast to the reviewer's comments highlighted above and below in bold).

Particularly concerning is the discrepancy between *in vivo* and *in vitro* binding of the full-length protein. The authors try to reconcile the absence of binding deficiency of their mutant by arguing that "there may be other amino acids outside of the GRAM1b that

contribute to anionic lipid sensing property of GRAMD1b proteins." However this explanation is inconsistent with the fact that **the full-length mutant protein is defective for recruitment in vivo**.

Reply: We would like to clarify our *in vivo* and *in vitro* results that we obtained with the two PS/anionic lipid-binding defective GRAM domain mutations (R191A and K161A/R191A) to this reviewer. As we showed in Figure 3F-H, PM recruitment of the "EGFP-tagged isolated GRAM domain of GRAMD1b (GRAM domain only construct)" was impaired *in vivo* when R191A single mutation or K161A/R191A double mutation was introduced to the GRAM domain. This is consistent with the results from our *in vitro* liposome binding assays that showed impaired interaction of purified GRAM domains of GRAMD1b with these mutations (R191A or K161A/R191A) to liposomes containing anionic lipids, including PS (Fig. 2C-E). However, we did not show, in any part of our manuscript, that these mutations (R191A and K161A/R191A) alone affect PM recruitment of "full-length GRAMD1b proteins" *in vivo*.

To further address the reviewer's concern, we additionally examined the impact of these mutations (R191A and K161A/R191A) to PM recruitment of EGFP-tagged full-length GRAMD1b *in vivo*. We found that PM recruitment of the EGFP-tagged full-length GRAMD1b was not affected when these mutations (R191A and K161A/R191A) have been introduced (see below). These results are consistent with our *in vitro* results, where purified near-full length GRAMD1b proteins that carry these PS/anionic lipid-binding defective GRAM domain mutations (R191A and K161A/R191A) could tether liposomes as efficiently as wild-type near-full length GRAMD1b proteins (Appendix Fig. S3C).

PM recruitment of EGFP-tagged full length GRAMD1b is not affected by basic patch mutations (R191A single or K161A/R191A double mutation) of the GRAM domain *in vivo*. Left: Time course of normalized EGFP signal, as assessed by TIRF microscopy, from GRAMD1 TKO (TKO) HeLa cells that stably expressed EGFP-tagged full-length GRAMD1b (EGFP-GRAMD1b) constructs as indicated. Cholesterol loading [the treatment with cholesterol/MCD complex (200 μM)] is indicated. Right: Values of ΔF/F₀ corresponding to 15 min time point as indicated by the arrow [mean ± SEM, n = 35 cells (EGFP-GRAMD1b), n = 36 cells [EGFP-GRAMD1b (R191A)], n = 44 cells [EGFP-GRAMD1b (K161A/R191A)], data are pooled from two independent experiments for EGFP-GRAMD1b and EGFP-GRAMD1b (R191A) and three independent experiments for EGFP-GRAMD1b (K161A/R191A); Dunnett's multiple comparisons test, n.s. denotes not significant].

The discrepancy between the behaviour of the protein in vitro and in vivo casting important doubt on the validity of the conclusions, it is important to, first emphasize this discrepancy, and second acknowledge the possibility that, in vivo, the two PS-binding mutants might affect the protein by other mechanisms than preventing PS binding.

Reply: As we stated above, we did not show in our manuscript that the two PS/anionic lipid-binding defective GRAM domain mutations (R191A and K161A/R191A) alone impair PM

recruitment of full-length GRAMD1b *in vivo*. We think our *in vitro* and *in vivo* results are consistent with the possibility that additional amino acid residues outside of the GRAM_{1b} (the GRAM domain of GRAMD1b) may contribute to anionic lipid sensing property of GRAMD1b proteins in the context of full-length proteins (despite the critical importance of the basic patch of the GRAM domain for the property of this domain to interact with anionic lipids in membranes in *in vitro* and *in vivo*). We edited the text throughout the manuscript to better clarify our *in vitro* and *in vivo* results and removed the word “PS-sensing” where it is appropriate to avoid confusion.

Additional points:

-Competitive lipid-binding versus steric hindrance is not addressed as the authors could not find a suitable lipid, outside of PS, to perform binding assays (DAG and PIP would not work for reasons well explained). It is nevertheless a pity that the authors did not experiment further on this point. They could for instance, dope their liposomes with Ni-NTA lipids and try to compete with His-tagged GFP (provided their recombinant protein is not itself His-tagged), or repeat the competition assays with mutant GRAM1b domains that cannot bind cholesterol or PS. The expectation would be that a cholesterol-binding deficient mutant will be competed by the C2 domain competitor but not the D4H-competitor, and vice versa for a PS-binding-deficient mutant. If not, then this would mean that steric hindrance is indeed the main cause of competition and not competitive lipid binding. Without addressing this point, the data of competitive binding have very little significance. It might shorten the manuscript to remove them entirely.

Reply: We appreciate the reviewer for these comments. We now removed the entire data set of the biochemical competitive binding assay from Figure 1 (previous Fig. 1A-C was removed, and a part of the cartoon model previously shown in Fig. EV1B is shown in **updated Figure 1A**) and shortened the manuscript (page 7).

the manuscript is focused on PS binding, yet the authors show that GRAMD1b appears to bind other anionic lipids just as well, including PI and PIP2, both of which are abundant in the PM. Why put such an emphasis on PS? Why not replace all mentions of PS-binding by anionic lipid binding?

Reply: We appreciate the reviewer for this constructive suggestion. We now replaced PS-binding by anionic lipid-binding as much as possible throughout the manuscript and figures (**updated Fig. 3F, Appendix Fig S1B-C**). In some parts, where we did experiments specifically with PS (e.g. liposome sedimentation assays etc.), we kept the original sentences and phrases for clarity.

-Fig 5B shows an assay where a cholesterol sensor is recruited to the PM if no WT GRAMD1b is present. This is used to claim that GRAMD1b transports free sterol away from the PM. An alternative explanation is that WT GRAMD1b binds to the same free cholesterol as the sensor and competes it away. The good control for that experiment would be to express a mutant of GRAMD1b that can bind the PM, but cannot transport sterols (a mutant of the STAR domain for instance). This control is not performed. Was it made in a previous paper? If yes a reference for this assay would be welcome. If not, what allows to conclude that competition rather than transport is at play? My apologies for not spotting this in the 1st round of review.

Reply: In our previous paper (Naito et al., 2019, PMID: 31724953), we expressed GRAMD1b that carries a cholesterol transport defective mutation within the StART-like domain (5P mutation) in GRAMD1 triple knockout cells (GRAMD1 TKO cells) and tested its ability to rescue the phenotype of these cells. Despite robust recruitment of GRAMD1b (5P) mutant to the PM, the 5P mutant could not induce dissociation of the GRAM domain of

GRAMD1b (GRAM_{1b} cholesterol biosensor) from the PM, thereby failing to rescue the phenotype of GRAMD1 TKO cells (Figure 5 of Naito et al., 2019). We now mention this result from our previous study by inserting the following sentence in the relevant part of the manuscript (page 12).

“We previously showed that PM recruitment of the GRAM_{1b} cholesterol biosensor following the treatment of GRAMD1 TKO cells with sphingomyelinase was suppressed by the re-expression of GRAMD1b, but not by the re-expression of GRAMD1b carrying mutations in the StART-like domain that cannot transport cholesterol (despite robust recruitment of this mutant protein to the PM upon sphingomyelinase treatment) (Naito et al., 2019), showing the specificity of this assay in examining cholesterol transporting property of GRAMD1b.”

Referee #3:

Most of the experiments I requested have been carried out successfully.

Thank you for the very nice comments. We appreciate the reviewer for all the constructive comments and suggestions to improve our manuscript.

*1) I had suggested to the authors to try to optimize the DHE-transfer assay between liposomes, because in my opinion the sterol-transfer activity measured for GRAMD1b **might be underestimated**. For comparison, Fig EV5D indicates a DHE transfer rate (for GRAMD1b WT) of ~0.02 DHE/s/GRAMD1b, i.e., 1.2 DHE/min/molecule GRAMD1b, whereas in the paper Horenkamp et al. 2018 EMBOj, Fig2D, the rate is 40 DHE/min/molecule GRAMD1b. Horenkamp et al used a GRAMD1b construct that is also capable of tethering membranes (via Lact-C2 domain at N-ter). The >30-fold difference in activity between the two studies is puzzling, even if the assay conditions are different. Can the authors comment on that?*

We appreciate the reviewer for this comment. In the experiment that Horenkamp et al. showed in Figure 2D of their manuscript, they used 1) different compositions of PM-like liposomes (e.g. they did not include cholesterol while we included cholesterol to promote GRAM domain-dependent tethering of PM-like and ER-like liposomes); 2) they used a chimeric GRAMD1b protein that replaced the GRAM domain of GRAMD1b with the Lact-C2 domain to induce tethering of PM-like liposomes (containing PS but not cholesterol) to ER-like liposomes. Because of these two major differences in the design of lipid transfer assays, we think it is difficult to compare the results of Horenkamp et al. to the ones of ours. However, it is indeed puzzling why Horenkamp et al. observed >30-fold higher DHE transport activity compared to our results. One possibility of such a difference is that the cholesterol transporting property of the StART-like domain within the near full-length GRAMD1b proteins that we used in our study is somehow autoinhibited by yet to be identified elements in the near full-length proteins (e.g. the N-terminal region containing the GRAM domain etc.). Another possibility (or technical limitation) is that addition of cholesterol in PM-like liposomes may have resulted in StART-like domain of GRAMD1b to extract cholesterol from these membranes more favorably than DHE (thus, reducing the rate of DHE transport). Further characterization of the regulatory mechanisms of GRAMD1-mediated sterol transport, in the context of near full-length proteins, requires future studies.

2) In the manuscript the unit describing the DHE transfer rate is nmole.GRAMD1b⁻¹.s⁻¹ (in Fig 4C, Fig 7C, Appendix Fig S3D), but I would suggest DHE.GRAMD1b⁻¹.s⁻¹, where DHE is 'molecule' not 'nmole'. This better fits with the data presented. Can the authors check this?

Thank you for spotting this. We checked the unit, and we agree with the reviewer that DHE.GRAMD1b⁻¹.s⁻¹ is the correct unit. We updated the unit in relevant graphs (Fig. 4C, Fig. 7C, Appendix Fig S3D).

I am pleased to inform you that your manuscript has been accepted for publication in The EMBO Journal.

Corresponding Author Name: Yasunori Saheki

Journal Submitted to: EMBO J

Manuscript Number: Manuscript EMBOJ-2020-106524